# LEARN LOW-DIMENSIONAL SHORTEST-PATH REPRESENTATION OF LARGE-SCALE AND COMPLEX GRAPHS

## ABSTRACT

Estimation of shortest-path (SP) distance lies at the heart of network analysis tasks. Along with the rapid emergence of large-scale and complex graphs, approximate SP-representing algorithms that transform a graph into compact and low-dimensional representations are critical for fast and scalable online analysis. Among different approaches, learning-based representation methods have made a breakthrough both in response time and accuracy. Several competitive works in learning-based methods heuristically leverage truncated random walk and optimization on the arbitrary linkage for SP representation learning. However, they have limitations on both exploration range and distance preservation. We propose in this paper an efficient and interpretable SP representation method called Betweenness Centrality-based Distance Resampling (BCDR). First, we prove that betweenness centrality-based random walk can occupy a wider exploration range of distance due to its awareness of high-order path structures. Second, we leverage distance resampling to simulate random shortest paths from original paths and prove that the optimization on such shortest paths preserves distance relations via implicitly decomposing SP distance-based similarity matrix. BCDR yields an average improvement of 25% accuracy and 25-30% query speed, compared to all existing approximate methods when evaluated on a broad class of real-world and synthetic graphs with diverse sizes and structures.

## 1 INTRODUCTION

Estimation of shortest-path (SP) distance lies at the heart of many network analysis tasks, such as centrality computation (Schönfeld & Pfeffer, 2021), node separation (Houidi et al., 2020), community detection (Zhang et al., 2020; Asif et al., 2022), which also directly contributes to enormous downstream applications, including point of interest (POI) search (Qi et al., 2020; Chen et al., 2021a) social relationship analysis (Carlton, 2020; Melkonian et al., 2021), biomedical structure prediction (Yue et al., 2019; Sokolowski & Wasserman, 2021), learning theory (Yang et al., 2021; Yuan et al., 2021), optimization (Rahmad Syah et al., 2021; Jiang et al., 2021b), etc. Nowadays, a key challenge of computing SP distance is the prohibitive complexity in very large and complex graphs. e.g., for a sparse undirected graph with $N$ nodes and $k$ queries, the time complexity of A* (Hart et al., 1968) and Dijkstra algorithm (Thorup & Zwick, 2004) are up to $O(kN)$ and $O(kN \log N)$ for unweighted and weighted graph, respectively.

Regarding this issue, various methods (Cohen et al., 2003; Fu et al., 2013; Akiba et al., 2013; Delling et al., 2014; Farhan et al., 2019; Liu et al., 2021) attempt answering exact distance in microseconds online via indexing or compressing techniques, which suffer huge storage costs on all pair SP distance representations and fail to reflect latent sub-structures in graphs for scalable queries (see Figure 1). Highly concise SP representation for large-scale and complex graphs remains to be studied yet. Regarding this, a surging number of approximate SP-representing algorithms that transform a graph into compact and low-dimensional representations are thus critical for fast and scalable online analysis. They can be categorized into oracle-based (Thorup & Zwick, 2004; Baswana & Kavitha, 2006), landmark-based (Potamias et al., 2009; Sarma et al., 2010; Gubichev et al., 2010) and learning-based (Rizi et al., 2018; Schlötterer et al., 2019; Qi et al., 2020; Jiang et al., 2021a) SP representation methods. Among these categories, learning-based methods are of high accuracy and short response time (see Table 1), owing much to flexible node embeddings in a metric space.

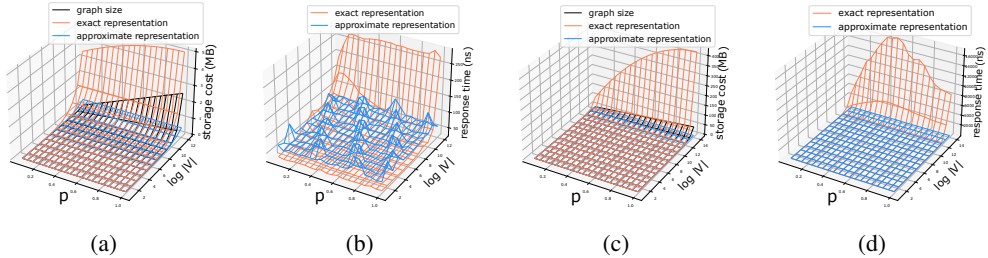

|     |     |     |     |
| :-: | :-: | :-: | :-: |
| (a) | (b) | (c) | (d) |

Figure 1: Differences between approximate (**ours.**) and exact (PLL (Akiba et al., 2013), efficient implementation of the hub-labeling method) SP representation methods regarding storage cost (megabytes, MB) and response time (nanoseconds, ns). We simulate a group of Bernoulli random graphs with $|V|$ nodes, and each edge is filled independently with probability $p$. **(a)** and **(c)** show the storage cost of exact representations increases dramatically relative to the graph size. **(b)** and **(d)** reflect longer response time of exact methods, induced by random access to massive information.

Table 1: Overall comparison of approaches to SP representation on DBLP dataset (A.8.4). PTC: preprocessing time complexity, PSC: preprocessing space complexity, RTC: response time complexity, TSC: the total storage cost for answering online distance queries, RT: real response time, AL: accuracy loss which is measured by mRE (see Equation 1). $N$: the number of nodes in the graph, $\bar{L}(N)$: the average label size of each node which increases along with $N$, $\bar{D}$: the amortized degree on each node. $\alpha_0$, $L$, $n$, $d$, $w$, $l$, $c$ and $\beta$ are hyperparameters in corresponded models.

| Categories | Method | PTC | PSC | RTC | TSC | RT | AL |
| --- | --- | --- | --- | --- | --- | --- | --- |
| Hub-labeling | PLL (Akiba et al., 2013) | $O(N^{1+\log \bar{L}(N)})$ | $O(N\bar{L}(N))$ | $O(\bar{L}(N))$ | 611.2 MB | 2104.4 ns | - |
| Oracle-based | ADO (Thorup & Zwick, 2004) | $O(\alpha_0 N^{1+\frac{1}{\alpha_0}})$ | $O(\alpha_0 N^{1+\frac{1}{\alpha_0}})$ | $O(\alpha_0)$ | 5,980 MB | 8,598 ns | 0.4985 |
| Landmark-based | LS (Potamias et al., 2009) | $O(lN)$ | $O(lN)$ | $O(l)$ | 334.6 MB | 12,094 ns | 0.3939 |
| Learning-based | Orion (Xiaohan et al., 2010) | $O(n^2 + nN)$ | $O(dN)$ | $O(d)$ | **19.35 MB** | 82.25 ns | 1.1897 |
| | Rigel (Xiaohan et al., 2011) | $O(n^2 + nN)$ | $O(dN)$ | $O(d + \bar{D})$ | 35.37 MB | 5,657 ns | 1.0662 |
| | DADL (Rizi et al., 2018) | $O((|L| + wl)N)$ | $O(dN + c)$ | $O(d + \bar{D})$ | 35.37 MB | 7,562 ns | 0.2016 |
| | Path2Vec (Kutuzov et al., 2019) | $O((|L| + wl)N)$ | $O(dN + c)$ | $O(d + \bar{D})$ | 35.37 MB | 7,700 ns | 0.6097 |
| | HALK (Schlötterer et al., 2019) | $O((|L| + wl)N)$ | $O(dN + c)$ | $O(d + \bar{D})$ | 35.37 MB | 7,704 ns | 0.3077 |
| | CatBoost-SDP (Jiang et al., 2021a) | $O((|L|N)$ | $O(|L|N + c)$ | $O(|L|c)$ | 44.16 MB | 9,270 ns | 0.0890 |
| | **BCDR (ours.)** | $O((|L| + \frac{wl}{\beta})N)$ | $O(|L|N + c)$ | $O(|L|c + \bar{D})$ | 39.19 MB | 7,247 ns | **0.0798** |
| | **BCDR-FQ (ours.)** | $O((|L| + \frac{wl}{\beta})N)$ | $O(dN + c)$ | $O(d)$ | **19.35 MB** | **58.82 ns** | 0.1840 |

Several competitive works in learning-based methods (Rizi et al., 2018; Schlötterer et al., 2019) heuristically leverage truncated random walk and optimization of node-cooccurrence likelihood on the arbitrary linkage to learn SP representations, which once achieved the state-of-the-art performance on approximation quality. However, they are not without limitations on efficiency and interpretability. On one side, a random walk is an *unstrained* node sequence from the root, possessing a limited exploration range of distance, thus resulting in uncaught distance relations with remote nodes. This is because each transition on nodes is not implied for a specific direction to move towards or beyond the root, especially after several walk steps, which restricts it from visiting remote nodes under limited walk steps (see Figure 2a). On the other side, the optimization on arbitrary linkage reflects excessively versatile local similarity among nodes, which preserves inaccurate distance relations from original graphs to the embedding space. In fact, it exerts a too-general metric over nodes' correlation, wherein the more edges or paths exist between two nodes, the stronger correlation they share. That means there are many ways to simulate a strong correlation for two nodes (e.g., add mutual edges, delete an edge to other nodes) even if some of the operations do not influence their actual SP distance (see Figures 2c and 2d). A detailed statement of related works on SP representation and motivation for estimating accurate SP distance can be found in Appendix A.1.

In this paper, we address the above shortcomings by proposing an efficient and interpretable SP representation method called Betweenness Centrality-based Distance Resampling (BCDR). It improves the approximation quality of SP representations with two components. The first is *betweenness centrality (BC)-based random walk* which explores a wider range of distance correlation on the graph due to its awareness of high-order path structures. **To our best knowledge, there is no existing method that combines betweenness centrality and random walk to learn SP representations.** We prove that BC-based transition is prone to jump out of local neighborhoods compared to random

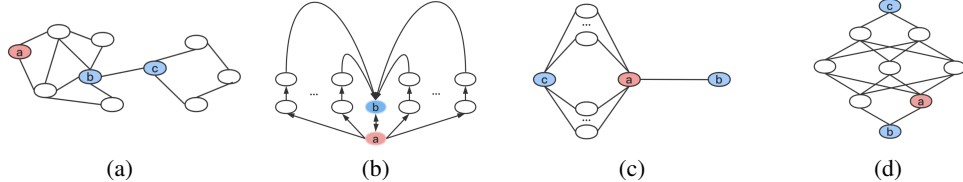

(a)        (b)        (c)        (d)

Figure 2: Distance confusion in previous SP representation learning. **(a)** random walk from $v_a$ has much difficulty in exploring beyond current community to $v_c$. **(b):** node similarity on random paths misleads the measurement of SP distance since the walk from $v_a$ is prone to steer clear of $v_b$ for starters and back to $v_b$ as the end, causing an extremely weak correlation between $v_a$ and $v_b$ even though they have an immediate edge. **(c):** a sufficient number of 2-hop links between $v_c$ and $v_a$ induce a shorter distance in embedding space than that of $v_b$ and $v_a$. **(d):** $v_b$ and $v_c$ sharing substantial connection are mapped closed to each other even if they have a large SP distance gap, while the divergence of distance between $v_b, v_c$ and $v_a$ is also plagued with extraction.

transition. The second is *distance resampling* which preserves accurate SP distance relations via implicitly decomposing an SP distance-based similarity matrix. In essence, it simulates the observation of random SPs from original walk paths and exerts desirable constraints on node representations to preserve distance relations over the graph.

We summarize the major contributions as follows: i) We propose *BC-based random walk* as an efficient strategy for exploring a wider range of SP distance within limited walk steps (see Section 3.1). ii) We propose *distance resampling* to preserve accurate distance relations among nodes to learn an interpretable SP representation (see Section 3.2). iii) We evaluate BCDR with a broad class of real-world and synthetic graphs, and it yields an average improvement of 25% accuracy and 25-30% query speed compared to all existing methods (see Section 4).

## 2   PRELIMINARY

**Notation:** $G = (V, E)$ denotes an undirected graph, with $V = \{v_i\}$ being the set of nodes and $E = \{(v_i, v_j)\}$ being the set of undirected edges, and $N = |V|$, $M = |E|$. We use $Z_{N \times d}$ to represent a matrix comprising embedded vectors of nodes, where $d$ is the embedding size, and the $i$-th row of $Z$ is corresponded with $v_i$. A path $p_{ij}$ of length $l \in \mathbb{N}_+$ on graph $G$ is an ordered sequence of nodes $(v_i, v_{a_1}, \cdots, v_{a_{l-1}}, v_j)$, where each node except the last one has an edge with the subsequent node. The shortest path $\mathring{p}_{ij}$ is one of the paths with the minimum length $D_{ij}$ between $v_i$ and $v_j$. Also, the SP distance matrix $D$ comprises $\{D_{ij}\}$. A node $v_i$'s neighborhood $\mathcal{N}_i$ is a set of nodes with an edge with $v_i$, i.e., $\mathcal{N}_i = \{v_j | (v_i, v_j) \in E\}$. For high-order neighborhoods of $v_i$, $\mathcal{N}_i^{(h)}$ is defined as a set of nodes $h$-hop away from $v_i$, i.e., $\{v_j | D_{ij} = h\}$. To avoid confusion with the symbol of paths, we use $\tilde{P}(\cdot)$ to represent a probability distribution in this paper. A truncated random walk $\mathcal{W}_i$ rooted at node $v_i$ of length $l$ is a random vector of $\langle \mathcal{W}_i^1, \mathcal{W}_i^2, \cdots, \mathcal{W}_i^l \rangle$, where $\mathcal{W}_i^k$ is a node chosen from the neighborhood of node $\mathcal{W}_i^{k-1}$ for $k = 1, ..., l$, with the initial probability $\tilde{P}(\mathcal{W}_i^0 = v_i) \equiv 1$. $W_i$ is a categorical distribution of nodes on $\mathcal{W}_i$, and the probability of each node in $W_i$ represents the frequency of occurrence on the sampled paths.

**Problem Definition & Metrics:** The evaluation of approximate SP representation methods is divided into two stages. For the offline stage, the processing time and memory usage when constructing SP representations are evaluated, and the storage size of such representations is considered. For the online stage, the query speed, memory usage, and approximation quality are evaluated. Thereinto, to evaluate query speed and memory usage, a million times of query requests for arbitrary node pairs are performed, then the memory and average response time for each node pair are recorded. For approximation quality, the commonly used metrics are mean of relative error (mRE) and mean of absolute error (mAE). For a group of SP distance queries $Q = \{(v_i, v_j)\}$, mRE is defined as the relative loss of the estimated value $\tilde{D}_{ij}$ with respect to the real value $D_{ij}$, while mAE measures the

absolute gap between them:

$$\text{mRE} := \frac{1}{|Q|} \sum_{(v_i,v_j) \in Q} \frac{|\tilde{D}_{ij} - D_{ij}|}{D_{ij}} \qquad \text{mAE} := \frac{1}{|Q|} \sum_{(v_i,v_j) \in Q} |\tilde{D}_{ij} - D_{ij}| \qquad (1)$$

## 3 METHOD

Although random walk (RW) is universally accepted as an efficient serialization strategy of similarity measurement on graphs (Grover & Leskovec, 2016; Zhuang & Ma, 2018), we argue that the intuitive practice of RW in representing SP structures has several limitations. Consider a walk path $p = (v_a, v_{a_1}, v_{a_2}, \cdots, v_{a_l}) \in P_a$ sampled by stochastic selection on neighborhoods from root node $v_a$. Distance measured along $p$ (i.e., the order on the walk) is not consistent with that on the graph (see Figure 2b) since the node sequence is *unstrained*, i.e., for $v_{a_i}, v_{a_j} \in p$, $i \leq j \not\Leftrightarrow D_{aa_i} \leq D_{aa_j}$, where $i$ and $j$ are indices of node $v_{a_i}$ and $v_{a_j}$ on $p$, and $1 \leq i, j \leq l$. Therefore, optimizing node co-occurrence likelihood on such walk paths incurs two problems.

1. **Problem 1: Limited exploration range of distance**. The exploration range of rooted random walk is not in proportion to its length since each transition on the walk has an agnostic tendency to move towards or beyond the root after a few steps (see Figure 2a).

2. **Problem 2: Intractability of distance relations on paths**. The distance measured on walk paths may not actually reflect the SP distance on the graph because of the unbalanced number of edges between different nodes (see Figure 2c and 2d).

In this section, we describe in detail our method as a decent way of representing SP structures. We discuss two techniques named *BC-based random walk* and *distance resampling* to address the above problems, respectively, and present the corresponding theoretical analysis for their interpretability. A time and space-efficient implementation of BCDR to integrate these techniques is available in Appendix A.2.

### 3.1 BC-BASED RANDOM WALK FOR WIDER EXPLORATION RANGE OF DISTANCE

**Definition 1.** *(Betweenness Centrality) Define $G = (V, E)$ as undirected graph. $v_i, v_s, v_t$ are arbitrary nodes in $V$. $\sigma_{st}(v_i)$ represents the number of shortest paths between $v_s$ and $v_t$ that pass $v_i$, and $\sigma_{st}$ is the total number of shortest paths between $v_s$ and $v_t$. Then we say that BC of $v_i$ is*

$$\text{BC}(v_i) = \sum_{s \neq i \neq t} \frac{\sigma_{st}(v_i)}{\sigma_{st}} \qquad (2)$$

To address Problem 1, we propose *BC-based random walk*. As defined in Definition 1, $\text{BC}(v_i)$ determines the probability of $v_i$ located on SPs of arbitrary node pairs. Thus, we consider a node with a large BC value vitally significant to drive the walk to move away from the root node, since it reveals an easy way of traveling to some other nodes with minimal steps. And to leverage this property, in BC-based random walk $\mathcal{W}_a = \langle \mathcal{W}_a^1, \mathcal{W}_a^2, ... \mathcal{W}_a^j, ... \rangle$ on node $v_a$, we prefer choosing nodes with the largest BC values among their neighborhoods when simulating walk paths, i.e.,

$$\tilde{P}(\mathcal{W}_a^j = v_m | \mathcal{W}_a^{j-1} = v_n) = \frac{\text{BC}(v_m)}{\sum_{v_k \in \mathcal{N}_n} \text{BC}(v_k)}, \ v_m \in \mathcal{N}_n \qquad (3)$$

Theorem 2, proved in Appendix A.3, indicates that *BC-based random walk* is prone to transit from $\mathcal{N}_a^{(h)}$ to $\mathcal{N}_a^{(h+1)}$, leading to a wider exploration range measured by the intrinsic graph's SP distance. Specifically, for each node $v_a$, $\mathcal{N}_a^{(h)}$ comprises two components, i.e., *final nodes* $f_h(v_a)$ and *connective nodes* $e_h(v_a)$. Thereinto, nodes in $f_h(v_a)$ have no edge with $\mathcal{N}_a^{(h+1)}$, while nodes in $e_h(v_a)$ have several edges with $\mathcal{N}_a^{(h+1)}$, as illustrated in Figure 3a. Our method significantly improves the performance when the number of *final nodes* $|f_h(v_a)|$ is larger or there are more edges from $f_h(v_a)$ to $e_h(v_a)$ (see analysis in Remark 1).

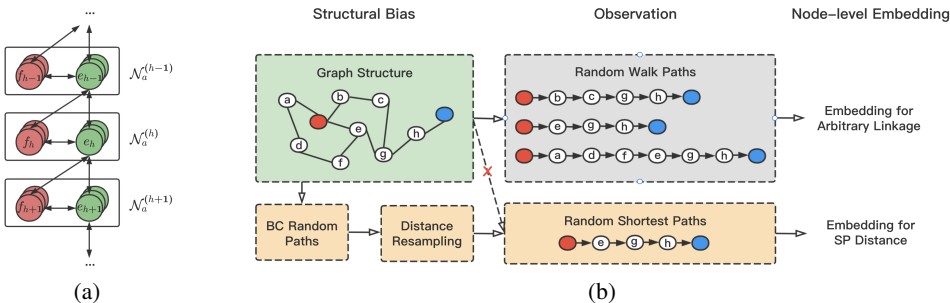

Figure 3: **(a):** high-order neighborhoods' structure of $v_a$. *BC-based random walk* enhances the transition from $f_h$ to $e_h$ and $e_h$ to $e_{h+1}$, prone to jump out of local neighborhoods. **(b):** comparison between general RW-based graph learning and BCDR. *Distance resampling* transforms the observation into random shortest paths, which exerts desirable constraints on learning SP representations.

Practically, pre-computing BC value on each node is time-consuming, which takes at least $O(NM)$ time. To address this problem, we estimate BC by leveraging breadth-first search (BFS) from a fixed number of landmarks, since nodes with large BC values tend to be visited first by BFS from any landmark. We show in Algorithm 1 and Appendix A.2.1 that this process could be involved in the simulation of distance triplets without introducing extra time complexity.

Finally, we conclude that *BC-based random walk* is a competitive walking pattern regarding exploration range of SP distance, since it possesses a strong tendency to jump out of local neighborhoods. We further verify our conclusion by comparing it with existing RW techniques in Section 4.2.

### 3.2 DISTANCE RESAMPLING FOR SP DISTANCE PRESERVATION

To address Problem 2, we propose *distance resampling*. We first illustrate a general RW-based graph learning paradigm in Figure 3b and clarify the differences between ours and other approaches. The basic idea of RW-based methods is to learn node-level embeddings $Z$ from pieces of *observation* (i.e., walk paths), and $Z$ thus reflects the structural bias on graphs. Specifically, for the naive RW strategy and its variants utilized in other approaches, the *observation* is a set of stochastic paths reflecting the property of arbitrary linkage between nodes, which asks $Z$ to preserve point-wise mutual information (PMI) similarity (proved in Levy & Goldberg (2014); Shaosheng et al. (2015)). Unfortunately, the PMI similarity shares no direct connection with SP distance and causes the problems depicted in Figure 2c and 2d. To fit $Z$ with correct information about SP structures, we intend to observe random shortest paths instead. This practice is feasible since the SP problem always has *optimal substructures*, i.e., the subpath between two nodes on any SP could also be extracted as an SP between these nodes. However, the prohibitive complexity of computing all pairs of SPs forbids us from performing such sufficient observation (see both technical and empirical comparisons between utilizing BCDR and directly sampling SPs for optimization in Appendix A.7). By way of an alternative, we propose a resampling strategy to transform BC random paths into approximate random SPs with efficient linear processing time and better performance.

Initially, we formulate the SP representation problem from the RW-based learning perspective. we refer to *random SP walk* $\mathring{W}_i$ as an ideal walking pattern whose transition reflects the probability of each shortest path passing through $v_i$. It means paths sampled from $\mathring{W}_i$ are prone to be an SP rooted at $v_i$. For sufficient observation on SPs, we thus have an optimization objective on $\mathring{W}_i$, i.e., $\mathcal{L}(Z) = \mathbb{E}_{v_i \sim \tilde{P}(V)} \left[ \log \tilde{P}_{\mathring{W}_i | \mathbf{Z}_i}(\mathring{W}_i | \mathbf{Z}_i) \right]$. To reduce optimization complexity, we replace the intrinsic probability normalization by negative sampling, according to Mikolov et al. (2013a;b), i.e.,

$$\mathcal{L}_n(Z) = \sum_{v_i \in V} \tilde{P}(v_i) \left\{ \mathbb{E}_{v_j \sim \tilde{P}_{\mathring{W}_i}(V)} [\log \hat{\sigma}(\mathbf{Z}_i \mathbf{Z}_j^T)] + \lambda \mathbb{E}_{v_k \sim \tilde{P}_n(V)} [\log \hat{\sigma}(-\mathbf{Z}_i \mathbf{Z}_k^T)] \right\} \quad (4)$$

, since we prefer an informative $Z$ instead of the accurate probability. Thereinto, $\tilde{P}_n$ is the distribution of negative sampling over the graph, $\lambda$ denotes the number of negative samples, and

$\hat{\sigma}(x) = (1 + e^{-x})^{-1}$. It is notable that $\mathring{W}_i$ on each node $v_i$ is backbreaking to extract, since it requires a traversal on all SPs passing through $v_i$. To address this, we revisit the node distribution $\tilde{P}_{W_i}$ on *BC-based random walk* $W_i$ and construct a distribution $\tilde{Q}_{W_i}$ resampled from $\tilde{P}_{W_i}$, as an efficient approximation to $\tilde{P}_{\mathring{W}_i}$, i.e.,

$$\tilde{Q}_{W_i}(v_j) = \frac{\alpha^{D_{ij}} \mathrm{BC}(v_j)}{\sum_{v_k \in W_i} \alpha^{D_{ik}} \mathrm{BC}(v_k)} \tag{5}$$

where $\alpha$ is a hyper-parameter controlling the weight decay by the distance, $0 < \alpha < 1$.

Finally, we maximize the following approximate objective $\hat{\mathcal{L}}_n$ on $W_i$ (instead of $\mathcal{L}_n$ on $\mathring{W}_i$), i.e.,

$$\hat{\mathcal{L}}_n(Z) = \sum_{v_i \in V} \tilde{P}(v_i) \left\{ \mathbb{E}_{v_j \sim \tilde{Q}_{W_i}(V)} [\log \hat{\sigma}(\mathbf{Z}_i \mathbf{Z}_j^T)] + \lambda \mathbb{E}_{v_k \sim \tilde{P}_n(V)} [\log \hat{\sigma}(-\mathbf{Z}_i \mathbf{Z}_k^T)] \right\} \tag{6}$$

We show in Proposition 1 that optimization of Equation 6 conforms to implicitly decompose an SP distance-based similarity matrix where for any $v_a$ and $v_b$ located far away from each other under the SP metric (i.e., a small $D_{ab}$) should be mapped with low similarity in the embedding space (i.e., a large $|\hat{D}_{ab}|$). Also, further discussion in Remark 2 shows that such similarity matrix $\hat{D}$ shares strong connections with the real SP distance matrix $D$ on graphs.

**Proposition 1.** *Let $G$ be an undirected graph, $W_i$ be the categorical distribution of nodes on the paths sampled by* BC-based random walk*. Negative sampling on each node $v_i$ takes a uniform distribution on $W_i$. Then, for sufficient observation of $W_1, \cdots, W_N$, maximizing $\hat{\mathcal{L}}_n$ defined by Equation 6 with embeddings $Z$ is equivalent to decomposing an SP distance-based similarity matrix $\hat{D} = ZZ^T$, where for any $v_a$ and $v_b$, the distance between them in the embedding space varies linearly with respect to distance $D_{ab}$, namely,*

$$\hat{D}_{ab} = Z_a Z_b^T = -\log \epsilon + D_{ab} \log \alpha \tag{7}$$

*where $\epsilon$ is a small constant related to the negative samples, which is independent of $v_b$, and $\alpha$ is the hyper-parameter defined in Equation 5, where $0 < \alpha < 1$.*

Proposition 1 is proved in Appendix A.4 by deriving the extreme point of $\hat{\mathcal{L}}_n$ regarding $Z$.

Then, we consider the preservation of SP distance relations. Some studies on metric learning (Hermans et al., 2017; Zeng et al., 2020) have revealed that a triplet of samples $(v_a, v_b, v_c)$ being easy to learn means if $v_b$ shares strong correlation with $v_a$, the distance between $v_b$ and $v_a$ in the embedding space should be shorter than that of $v_c$ and $v_a$. With this property, we have the following theorem (proved in A.6 by directly applying Proprosition 1), which indicates that our method is consistent with distance relations under the intrinsic SP metric.

**Theorem 1.** *Each symbol here follows the definition in Proposition 1. Let $D$ be a global distance matrix defined on graph $G$ and $D_{ab}$ be graph's SP distance between node $v_a$ and $v_b$. Then for any nodes $v_a, v_b, v_c \in G$,*

$$(D_{ab} - D_{ac})(\hat{D}_{ab} - \hat{D}_{ac}) \leq 0 \tag{8}$$

In conclusion, we discuss here the significance of *distance resampling* for preserving accurate distance relations. It exerts two implicit constraints on $Z$ to learn an interpretable SP representation. First, as stated in Proposition 1, the distance measured in the embedding space shares a strong negative correlation with that measured on the graph. Second, for any node triplet, the distance relation between any two of them is preserved according to Theorem 1. The two constraints are further verified in Section 4.3 against existing techniques.

### 3.3 Efficient Implementation of BCDR Algorithm

We also provide a time and space-efficient implementation of BCDR to integrate the above techniques in Algorithm 1. Like previous learning-based SP representation methods (Rizi et al., 2018), we first transform the graph into low-dimensional embeddings (i.e., $Z_{N \times d}$) and learn a distance predictor $g_\phi : (\mathbb{R}^d, \mathbb{R}^d) \to \mathbb{R}$ by observed distance triplets $\{(\mathbf{Z}_a, \mathbf{Z}_b, D_{ab})\}$. Then, the predictor $g_\phi$ will be involved in answering online distance queries. In addition, similar to Jiang et al. (2021a), we also improve the prediction results via gradient boosting techniques. The detailed designs of these procedures are described in Appendix A.2, and an ablation study to evaluate their impact on performance is provided in Appendix A.11.

# 4 EXPERIMENTAL EVALUATION

In this section, we show the comprehensive performance of BCDR with 5 real-world graphs of different sizes and 6 synthetic graphs of different structures. Specifically, we evaluate BCDR on 3 small graphs (i.e., Cora, Facebook, and GrQc) and 2 large graphs (i.e., DBLP and Youtube) for its scalability (see Section 4.1), and evaluate it on 6 synthetic graphs for its representational capacity of complex structures (see Section 4.2 and 4.3). Our method is compared with strong baselines from both approximate SP representation and general graph representation learning (GRL). In the experiments, we also provide two variants of BCDR, i.e., BCDR-FC and BCDR-FQ, for accelerating the construction and querying process, respectively. A detailed description of the datasets, including statistics and visualization, is thoroughly provided in Appendix A.8.

## 4.1 PERFORMANCE OF APPROXIMATE SP DISTANCE QUERY

We compare BCDR with other learning-based SP representation methods (i.e., Orion (Xiaohan et al., 2010), Rigel (Xiaohan et al., 2011), DADL (Rizi et al., 2018), Path2Vec (Kutuzov et al., 2019), HALK (Schlötterer et al., 2019)) and CatBoost-SDP (Jiang et al., 2021a) as well as other approximate methods, including landmark-based (i.e., LS (Potamias et al., 2009) and oracle-based (i.e., ADO (Thorup & Zwick, 2004)) techniques. All of the above models are run with six 3.50GHz Intel Xeon(R) CPUs and 128GB memory, and the precomputed representations of each model are serialized by Pickle. Each baseline generally follows the default parameters discussed in its paper with some trivial changes, so that its performance can be evaluated in a unified way. The detailed parameter setups of each model are provided in Appendix A.9. Like previous works, we initially compute all pairs of SP distance on each graph by BFS and take a uniform sampling to select $1,000,000$ distance triplets $\{(v_a, v_b, D_{ab})\}$ as test samples. All of the baselines, including ours, are purely implemented in Python 3.9 and evaluated under the same environment. Since only unweighted graphs are considered, the outputs of each model are quantized to integer when evaluating accuracy loss. Some of the experimental results are shown in Table 2 (see Appendix A.10 for extended comparisons with GRL models). We can see from the table that our model not only outperforms previous models regarding accuracy loss for all graphs but also shares competitive results on other metrics.

In detail, for *accuracy loss* (mAE and mRE), BCDR answers arbitrary queries with the minimum error due to a wider exploration range of distance and distance-preserved optimization. Notably, the variants of BCDR without boosting module (i.e., BCDR-FQ and BCDR-FC) also achieve the highest accuracy against other RW-based learning approaches (i.e., DADL and HALK) within almost the least storage cost. For *offline processing time* (PT), *memory usage* (PMU), and *storage cost* (SC), the results show BCDR possesses powerful scalability against the growth of graph scale. Even for a graph with millions of nodes, the offline processing could be completed within several hours, and the memory usage is close to the graph size. This is because we perform *BC-based random walks* with a fixed length on each graph, and the size of walk data is further reduced by *distance resampling*. In addition, although CatBoost-SDP seems to achieve strong scalability on these metrics, we need to point out that this method does not learn any representation of nodes and completely optimizes all pairs of distance in a boosting way, which subsequently suffers higher time and space cost for online queries. For *response time* (RT) and *memory usage* (QMU) in querying, we see BCDR-FC and most other learning-based models share similar low memory overhead since each distance query could be answered by checking the node embeddings and graph adjacency matrices. Furthermore, BCDR-FQ and Orion could answer such a query within tens of nanoseconds due to the absence of double-checking on adjacency matrices.

Besides, to evaluate the impact of critical components and hyper-parameters in BCDR, we further conduct an ablation study in Appendix A.11 where we discuss 6 different modifications to BCDR as well as an investigation on 9 critical parameters to show their impacts on different metrics.

## 4.2 EXPLORATION RANGE OF DISTANCE

As stated in Section 3.1, the exploration range of distance could be widened by *BC-based random walk* (BC-RW), since the latter helps to jump out of local neighborhoods. Here, we compare BC-RW with existing renowned walk strategies, including naive random walk (NRW) (Perozzi et al., 2014; Zhuang & Ma, 2018), second-order random walk (SORW) (Grover & Leskovec, 2016), and

Table 2: Performance comparison of approximate methods on SP distance queries. PT: processing time when constructing SP representations, PMU: processing memory usage, SC: space cost on storing SP representations, RT: average response time of answering a distance query, QMU: querying memory usage. mAE and mRE are the accuracy metrics. DNF means it did not finish in one day. We bold the top three performances, and highlight the top one with an underline.

| Dataset | Model | PT | PMU | SC | RT | QMU | mAE | mRE |
|---|---|---|---|---|---|---|---|---|
| Cora | ADO | **0.3179 s** | 6.205 MB | 0.9905 MB | 9,813 ns | 11.89 MB | 2.1070 | 0.4266 |
| | LS | **0.6355 s** | 2.952 MB | 0.6791 MB | 11,537 ns | 10.92 MB | 1.0599 | 0.2068 |
| | Orion | 61.34 s | 15.06 MB | **0.1655 MB** | **63.87 ns** | **0.1654 MB** | 3.0542 | 0.5242 |
| | Rigel | 61.36 s | 15.06 MB | **0.1655 MB** | **4,587 ns** | 0.2482 MB | 3.0464 | 0.5164 |
| | DADL | 68.22 s | 27.36 MB | **0.1696 MB** | 6,798 ns | 0.2486 MB | 1.0822 | 0.1862 |
| | Path2Vec | 172.9 s | 2.870 MB | **0.1696 MB** | 6,820 ns | 0.2486 MB | 3.2020 | 0.6066 |
| | HALK | 30.44s | 6.715 MB | **0.1696 MB** | 6,802 ns | 0.2486 MB | 1.7702 | 0.3293 |
| | CatBoost-SDP | **9.582 s** | **0.4444 MB** | 0.4711 MB | 8,987 ns | 4.285 MB | 0.8907 | 0.1585 |
| | BCDR (ours.) | 40.85 s | 2.979 MB | 0.4079 MB | 6,376 ns | 4.0623 MB | **0.8046** | **0.1411** |
| | BCDR-FQ (ours.) | 39.51 s | **2.728 MB** | **0.1696 MB** | **50.85 ns** | **0.1657 MB** | **0.7249** | **0.1247** |
| | BCDR-FC (ours.) | 35.60 s | **2.728 MB** | **0.1696 MB** | 6,830 ns | 0.2486 MB | **0.8243** | **0.1384** |
| Facebook | ADO | **1.991 s** | 13.10 MB | 1.783 MB | 8,105 ns | 16.94 MB | 1.1842 | 0.5080 |
| | LS | **3.382 s** | **4.390 MB** | 0.9929 MB | 11,664 ns | 12.28 MB | 0.9566 | 0.3924 |
| | Orion | 80.53 s | 21.00 MB | **0.2419 MB** | **60.81 ns** | **0.2418 MB** | 1.7770 | 0.6864 |
| | Rigel | 80.31 s | 21.00 MB | **0.2419 MB** | **4,976 ns** | 1.588 MB | 1.7531 | 0.6625 |
| | DADL | 176.3 s | 42.44 MB | **0.2459 MB** | 8,601 ns | 1.588 MB | 0.2250 | 0.0792 |
| | Path2Vec | 284.5 s | 5.459 MB | **0.2459 MB** | 8,582 ns | 1.588 MB | 1.4263 | 0.5489 |
| | HALK | 75.95 s | 21.28 MB | **0.2459 MB** | 8,777 ns | 1.588 MB | 0.9004 | 0.3517 |
| | CatBoost-SDP | **12.84 s** | **1.874 MB** | 0.6896 MB | 8,955 ns | 4.504 MB | **0.0203** | **0.0159** |
| | BCDR (ours.) | 142.7 s | 6.804 MB | 1.210 MB | 8,298 ns | 5.402 MB | **0.0106** | **0.0044** |
| | BCDR-FQ (ours.) | 160.6 s | **5.213 MB** | 0.2459 MB | **53.78 ns** | 0.2421 MB | 0.0978 | 0.0385 |
| | BCDR-FC(ours.) | 91.71 s | **5.213 MB** | 0.2459 MB | 8,713 ns | 1.588 MB | 0.1463 | 0.0478 |
| GrQc | ADO | **1.322 s** | 17.75 MB | 2.922 MB | 13,555 ns | 19.19 MB | 1.8747 | 0.3582 |
| | LS | **2.098 s** | 5.722 MB | 1.315 MB | 11,825 ns | 13.60 MB | 1.1538 | 0.2097 |
| | Orion | 98.14 s | 21.33 MB | **0.3202 MB** | **62.80 ns** | **0.3201 MB** | 3.0532 | 0.4849 |
| | Rigel | 98.00 s | 21.33 MB | **0.3202 MB** | **4,872 ns** | 0.5519 MB | 3.0493 | 0.4810 |
| | DADL | 103.0 s | 53.09 MB | **0.3242 MB** | 6,912 ns | 0.5523 MB | 0.9812 | 0.1659 |
| | Path2Vec | 333.0 s | 5.632 MB | **0.3242 MB** | 6,938 ns | 0.5523 MB | 4.1239 | 0.7231 |
| | HALK | 48.30 s | 13.80 MB | **0.3242 MB** | 6,955 ns | 0.5523 MB | 1.1695 | 0.2063 |
| | CatBoost-SDP | **11.03 s** | **0.9305 MB** | 0.7846 MB | 9,015 ns | 4.599 MB | 0.7815 | **0.1416** |
| | BCDR (ours.) | 69.61 s | 5.907 MB | 0.7916 MB | 6,452 ns | 4.367 MB | **0.7043** | **0.1274** |
| | BCDR-FQ (ours.) | 67.98 s | **5.351 MB** | **0.3242 MB** | **51.90 ns** | 0.3204 MB | 0.8743 | 0.1501 |
| | BCDR-FC (ours.) | 58.30 s | **5.351 MB** | **0.3242 MB** | 6,965 ns | 0.5523 MB | 0.8776 | **0.1442** |
| DBLP | ADO | 37,029 s | 8,899 MB | 199.8 MB | 8,598 ns | 5,980 MB | 3.0691 | 0.4985 |
| | LS | 5,838 s | 349.0 MB | 80.02 MB | 12,094 ns | 344.6 MB | 2.5060 | 0.3939 |
| | Orion | 5,531 s | 321.8 MB | **19.36 MB** | **82.25 ns** | **19.35 MB** | 3.5044 | 0.5165 |
| | Rigel | 5,523 s | 321.8 MB | **19.36 MB** | **5,657 ns** | 35.37 MB | 3.5043 | 0.5164 |
| | DADL | 2,650 s | 1,958 MB | **19.36 MB** | 7,562 ns | 35.37 MB | 1.2753 | 0.2016 |
| | Path2Vec | 27,453 s | 140.9 MB | **19.36 MB** | 7,700 ns | 35.37 MB | 3.9474 | 0.6097 |
| | HALK | 1,270 s | 649.6 MB | **19.36 MB** | 7,704 ns | 35.37 MB | 1.8477 | 0.3077 |
| | CatBoost-SDP | **244.4 s** | 56.47 MB | 40.34 MB | 9,270 ns | 44.16 MB | 0.5492 | 0.0890 |
| | BCDR (ours.) | 1,099 s | 89.04 MB | 41.82 MB | 7,247 ns | 39.19 MB | **0.4923** | **0.0798** |
| | BCDR-FQ (ours.) | **1,026 s** | **53.52 MB** | **19.36 MB** | **58.82 ns** | **19.35 MB** | 1.3018 | 0.1840 |
| | BCDR-FC (ours.) | **999.8 s** | **53.52 MB** | **19.36 MB** | 7,617 ns | 35.37 MB | 1.1014 | 0.1580 |
| Youtube | ADO | DNF | – | – | – | – | – | – |
| | LS | 87,902 s | 1,258 MB | 286.7 MB | 16,672 ns | 1,217 MB | 2.0159 | 0.4091 |
| | Orion | 50,484 s | 1,251 MB | **69.26 MB** | **163.9 ns** | **69.26 MB** | 2.8642 | 0.5473 |
| | Rigel | 51,246 s | 1,251 MB | **69.26 MB** | 13,808 ns | 114.9 MB | 2.8642 | 0.5473 |
| | DADL | 28,351 s | 6,983 MB | **69.27 MB** | 7,708 ns | 114.9 MB | 1.1144 | 0.2163 |
| | Path2Vec | DNF | – | – | – | – | – | – |
| | HALK | **5,971 s** | 2,187 MB | **69.27 MB** | 7,764 ns | 114.9 MB | 1.9035 | 0.3860 |
| | CatBoost-SDP | **3,030 s** | **190.4 MB** | 141.7 MB | 9,921 ns | 145.5 MB | **0.4022** | **0.0724** |
| | BCDR (ours.) | 9,247 s | 295.2 MB | 141.6 MB | 7,393 ns | 118.7 MB | **0.3297** | **0.0676** |
| | BCDR-FQ (ours.) | 7,893 s | **179.8 MB** | **69.27 MB** | **63.78 ns** | 69.27 MB | 0.9004 | 0.1704 |
| | BCDR-FC (ours.) | **6,000 s** | **179.8 MB** | **69.27 MB** | 6,156 ns | 114.9 MB | 0.9083 | 0.1745 |

random surfing (RS) (Cao et al., 2016). We also consider a DFS-like random walk (DFS-RW) as a strong baseline by setting a very small $q$ in Node2Vec for deep exploration. The methods are tested on 6 synthetic graphs with divergent structures. We randomly sample 20 root nodes and, for each root, simulate 10 walks to show how many nodes with different distance are explored at each step of the walk. The ideal situation for rooted walk paths with a fixed length $l$ is to cover up to nodes $l$-hop away from the current root. The results on *circle graphs* are shown in Figure 4. We can see from the results that our proposed BC-RW is much more competitive in exploring a wider range of SP distance. Further results and analysis on different graph structures are presented in Appendix A.12.

## 4.3   PRESERVATION OF DISTANCE RELATIONS

As discussed in Section 3.2, *distance resampling* is proposed to preserve accurate distance relations via implicitly decomposing an SP distance-based similarity matrix. Here, we show its interpretability for SP representations by visualizing the properties of embedded vectors $Z$ when compared with

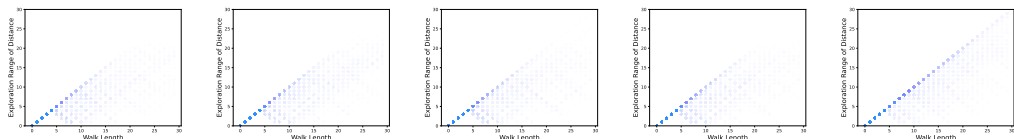

Figure 4: Exploration range of distance of different walk strategies on *circle graphs*. **Column from left to right:** different walk strategies, i.e, NRW, SORW, RS, DFS-RW, BC-RW (ours.).

the maximum likelihood optimization on other biased random walks. Environment configuration follows the previous section.

First, we evaluate the relation of distance measured on graphs and embedding spaces for each node pair. Specifically, distance on the embedding space is measured by inner product $\mathbf{Z}_i \mathbf{Z}_j^T$ for given nodes $v_i$ and $v_j$, and that on the graph is measured by SP distance. Initially, we learn embedded vectors from walk paths simulated by each walk strategy and randomly sample 100 source nodes with 100 destinations for each source. The results on *circle graphs* are shown in Figure 5 (refer to Appendix A.13 for extended results on other graphs), which indicates that embeddings enhanced by *distance resampling* have a better tendency to maintain a linear relationship on the distance metric between the original graph and embedding space. The results also verified our analysis in Remark 2 regarding relations between the SP distance-based similarity matrix $\hat{D}$ and the distance matrix $D$.

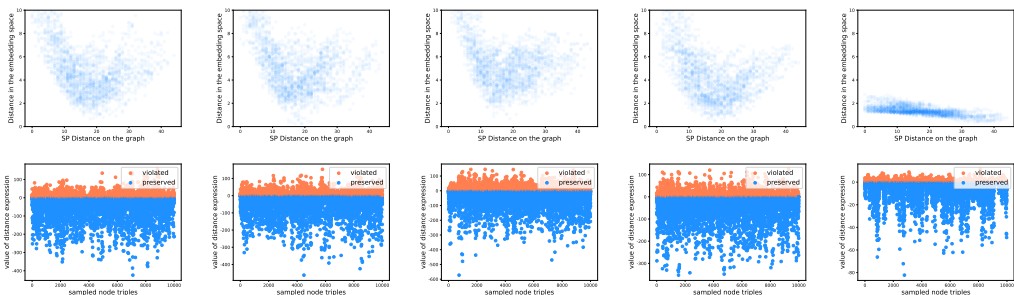

Figure 5: **Row 1:** measured distance from the embedding space and the original graph. **Row 2:** whether distance relations are violated in the embedding space. **Columns from left to right:** embeddings learned by different walk strategies,i.e., NRW, SORW, RS, DFS-RW, and BC-RW (ours.), respectively. For ours, walk paths are further simulated by *distance resampling*.

Second, we try to find out how much the probability distance relation is violated in the embedding space, i.e., whether a pair of nodes with larger SP distance is corresponded with less embedded similarity as described in Theorem 1. We randomly take $10,000$ node triplets $\{(v_a, v_b, v_c)\}$, and record if Equation 8 is satisfied. The results on *circle graphs* are shown in Figure 5 (refer to Appendix A.13 for extended results on other graphs). The figure confirms that our model is much more satisfactory in preserving distance relation than existing methods. This is because BC-RW provides sufficient observation on each node by locating many remote nodes with a sequence of centralized nodes on a graph, and thus *distance resampling* based on such observation could preserve distance relations of each node within its exploration range.

## 5 CONCLUSION

In this paper, we propose a novel graph SP representation method called Betweenness Centrality-based Distance Resampling (BCDR) and discuss two significant techniques for an efficient and interpretable SP representation. The experimental evaluation indicates that BCDR improves the approximation quality with a shorter response time for SP distance queries and possesses strong scalability to large-scale and complex graphs. Notably, the produced node representations by our method also reflect the highly-efficient paths for high-order message passing in GNNs, which appears to be helpful for structural graph pooling and inference. We leave it for our future work.

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

## A APPENDIX

### A.1 RELATED WORK

#### A.1.1 ESTIMATION OF ACCURATE SP DISTANCE

As an important global measurement on graphs, SP distance reflects the minimum travelling cost from node to node, similar to the geodesic distance on manifolds. Along the rapid emergence of large-scale graphs in many areas, space- and time-efficient estimation of accurate SP distance is urgently required in many downstream applications. In this part, we investigate the direct impact of SP distance estimation in different fields by discussing several real-world scenerios.

**Case 1: find nearest points of interest in road and social networks** Points of interest (POIs) (Chen et al., 2021a) are specific point locations that someone may find useful or interesting, e.g., hotels, campsites, fuel stations, etc. A real road network may contain millions of nodes, while thousands of users may issue SP distance queries simultaneously for searching the nearest POI from their location, like 'finding restaurants within 5 km distance' or 'ranking restaurant search results by distance'. To achieve such demands, learning to accurately and fast answer SP distance with limited computing resources is of high significance. Specifically, utilizing limited computing resources means the algorithm should be space- and time-efficient. Thereinto, less storage overhead enables the representations to be stored in users' mobile devices instead of centrally computing SP on the server. And less query time ensures that the computation of SP distance can be processed in real-time (since some POIs may change their positions frequently over time).

**Case 2: construct skeleton graph from mesh for 3D animation** In the literature of 3D animation, animating an articulated character requires constructing a skeleton graph to control the movement of the surface, i.e., place the skeleton joints inside the character and specify which parts of the surface are attached to which bone. A critical technique (Aujay et al., 2007; Poirier & Paquette, 2009) to automatically embed a skeleton into a character relies on computing a harmonic function under the SP metric on mesh graphs. This requires finding a group of nodes that locally maximize SP distance with the user-defined node. Since the mesh of a delicate-described character may have tens or hundreds of vertices, estimating and finding such nodes with the longest SP distance accurately and fast are also well-motivated.

**Case 3: estimate latencies in communication networks** In large-scale communication networks, the latencies between Internet hosts are defined as a round-trip measurement from one to another (i.e., SP distance), which is utilized for performance optimization in many network applications such as content distribution networks (Ratnasamy et al., 2002), multicast systems (Nogueira, 2014), distributed file system (Rhea et al., 2003), etc.

### A.1.2 APPROXIMATE SP REPRESENTATION

**Hard-coding Perspective** Compared with exact SP representations that improve query speed at the expense of huge storage costs, approximate methods are designed to find a compact and scalable representation of high performance both in time and space. The basic idea of these methods is to reduce the complexity of SP distance matrices. Thorup and Zwick (Thorup & Zwick, 2004) initially observe that a hierarchical sparse sampling of nodes could significantly reduce the number of elements in the distance matrix, and all pairs of SP distance are thus approximately represented by the distance relations on those nodes with a bounded error. They also provide a time-efficient algorithm to compute the pruned distance relations. Several later extensions (Baswana & Kavitha, 2006; Enachescu et al., 2008; Chen et al., 2009) are proposed to improve the processing time and space on specific graphs. However, these methods still have limitations on space complexity and accuracy. First, the sampled distance relations take $O(\alpha_0 N^{1+\alpha_0})$ space which is not linear to the number of nodes $N$, thus inducing scalable problems on large graphs. Second, the bounded error is often unacceptable on graphs with smaller diameters since even the most accurate model (with $\alpha_0 = 2$) allows three times the error of real distance.

Addressing these issues, landmark-based distance estimation methods (Potamias et al., 2009; Gubichev et al., 2010; Sarma et al., 2010) are proposed. Instead of sampling hierarchical sets of nodes, landmark-based methods only preserve distance relations between a fixed number of nodes (called landmarks) to others on the graph, and all pairs of SP distance could be bounded by their distance related to landmarks according to triangle inequality (Zheng et al., 2005; Lee et al., 2006; Mao et al., 2006), i.e., for any nodes $v_a$ and $v_b$,

$$\max_{v_i \in L} |D_{ai} - D_{ib}| \leq D_{ab} \leq \min_{v_j \in L} |D_{ai} + D_{ib}| \tag{9}$$

where $D_{ab}$ denotes the SP distance between $v_a$ and $v_b$, $L$ denotes the set of landmarks. The average accuracy could be optimized by selecting proper landmarks that covers as many SPs as possible. Unfortunately, finding the optimal finite set of landmarks with the minimal size has been proved to be NP-hard, which is mapping to a set-cover problem (Balas, 1982). Therefore, several heuristic selection strategies are discussed to tight Equation 9 by leading $D_{ab}$ almost near to its upper bound (Potamias et al., 2009). Other efforts (Gubichev et al., 2010; Tretyakov et al., 2011) are made to store SP trees for each landmark instead of distances at the cost of extra storage and response time. Nevertheless, the approximation performance in these models relies highly on graph structures, since less-centralized graphs (e.g., a grid-like graph) and graphs of large diameters (e.g., a large planar graph) require a large number of distributed landmarks to cover remote pairs of nodes.

**Learning Perspective** Instead of the hard-coding techniques mentioned above, our work steps forward from a learning perspective of SP distance estimation, which constructs general and scalable representations for arbitrary graphs. Under the low-rank assumption of SP distance matrices, the basic idea of learning-based methods is to transform the graph into a metric space while preserving the distance between pairs of nodes. As the embedding space is low-dimensional and continuous, extracting distance from learning-based SP representations is fast and scalable. However, directly optimizing the distance between all pairs is time-consuming, which takes at least $O(N^2)$ time for computing distance and subsequent optimization. Towards this, many graph coordinate systems (Ng & Zhang, 2002; Tang & Crovella, 2003; Costa et al., 2004; Dabek et al., 2004; Ng & Zhang, 2004) have been studied in the past years. To reduce processing complexity, a feasible learning procedure for very large graphs later proposed in Orion (Xiaohan et al., 2010) contains three steps. First, perform breadth-first search (BFS) from a small landmark set $L$ and record node pairs as well as their distance as training triplets $\{\langle v_l, v_a, D_{la} \rangle\}$ where $v_l \in L$, $v_a \in V$. Second, create a graph coordinate system $M$ by preserving distance relations among nodes in $L$, i.e.,

$$\arg \min_{L^M = \{v_i^M | v_i \in L\}} \sum_{v_i, v_j \in L} |D_{ij}^M - D_{ij}| \tag{10}$$

where $v_i^M$ denotes the embedded vector corresponding to the node $v_i$, and $D_{ij}^M$ denotes the geodesic distance between $v_i^M$ and $v_j^M$ measured on $M$. Third, fix $L^M$ and calibrate distance between other nodes and landmarks iteratively. Among these steps, the metric tensor defined on $M$ significantly affects the accuracy of distance estimation, and models regarding embedding in euclidean space (Rao, 1999; Lee, 2009; Xiaohan et al., 2010) and hyperbolic space (Shavitt & Tankel, 2008; Xiaohan et al., 2011) are well studied respectively.

Inspired by the great success in graph representation learning (GRL), further work (Rizi et al., 2018; Schlötterer et al., 2019) including ours treats $M$ as an agnostic but definite manifold learned by GRL techniques and estimates distance based on learnable metric criteria (usually a multi-layer perceptron). Therefore, the learning task here is converted from "calibrate the position of each node" to "learn powerful metric criteria to extract distance everywhere." This novel paradigm achieves higher accuracy with reduced training time despite the fact that we are unsure about whether general GRL models could embed sufficient information to infer all pairs of SPs. In this paper, we thus discuss an interpretable SP representation learning method and improve the comprehensive performance of SP distance estimation.

### A.1.3 Graph Representation Learning

Graph representation learning (GRL) organizes symbolic objects (such as nodes, edges, and clusters) in a way such that their similarities on the graph are well-preserved in the low-dimensional embedding space. Currently, most of these methods focus on preserving arbitrary linkage on graphs by considering high-order adjacency matrices, and are categorized into matrix factorization (MF) and random walk (RW) approaches. Thereinto, our work shares strong connections with general RW approaches, which embed remote nodes' correlation within linear complexity compared with MF methods. The basic idea of RW-based learning methods proposed in Deepwalk (Perozzi et al., 2014) is to dump complicated linkage structure on graphs into a few fixed-length node sequences in a statistical view and learn node embeddings to reflect their co-occurrence on walk paths using a skip-gram algorithm (Mikolov et al., 2013a;b). The learning process is to solve a maximum likelihood optimization problem based on observed sequences, i.e., for any nodes $v_a$ and $v_b$,

$$\arg\max_{\mathbf{Z}_a, \mathbf{Z}_b} \log \hat{\sigma}(\mathbf{Z}_a \mathbf{Z}_b^T) + \lambda \mathbb{E}_{v_k \sim \tilde{P}_n(V)}[\log \hat{\sigma}(-\mathbf{Z}_a \mathbf{Z}_k^T)])] \tag{11}$$

where $\mathbf{Z}_a$ denotes the embedded vector of $v_a$, $\lambda$ denotes the number of negative samples, $\hat{\sigma}(\cdot)$ denotes the sigmoid function where $\hat{\sigma}(x) = (1 + e^{-x})^{-1}$, and $\tilde{P}_n(\cdot)$ denotes a probability distribution of the negative sampling. Several practical strategies are proposed in the past few years to simulate structure-aware traversal on graphs in RW-based methods (Grover & Leskovec, 2016; Cao et al., 2016; Perozzi et al., 2017; Chen et al., 2018). In detail, to enhance sensibility on divergent structures, Node2Vec (Grover & Leskovec, 2016) exploits a biased random walk strategy to perform combinatorial traversal on graphs, including breadth-first search (BFS) and depth-first search (DFS), which explores both local-neighborhood linkage and correlation with remote nodes simultaneously. To reflect the locality around each node, a random walk with restarting mechanism (called random surfing) is applied in learning point-wise mutual information (PMI) representations (Cao et al., 2016). For capturing multi-scale representations of different-order neighborhoods, hierarchical random walks by skipping some of the nodes on paths are also proposed (Perozzi et al., 2017; Chen et al., 2018). Recently, Schlotterer et al. (Schlötterer et al., 2019) observed that RW-based methods perform better than others in exploring a wide range of distance and evaluated these methods as being helpful for SP distance estimation. However, a specific and insightful investigation of RW-based SP representation remains to be studied. In this paper, we discuss a novel biased random walk strategy toward high-order SP exploration and provide an explicit optimization algorithm for distance-preserved representation.

### A.2 Efficient Implementation of BCDR Algorithm

We discuss here an efficient implementation to integrate the techniques mentioned in Section 3.1 and 3.2. Our algorithm is presented in Algorithm 1, including constructing SP representations and answering online distance queries. The description and analysis of these procedures are provided as follows.

---

**Algorithm 1:** construct SP representations & answer distance queries

**Input:** input graph $G = (V, E)$, set of landmarks $L$, distance queries of node pairs $Q$, dimension of embeddings $d$, number of walk paths on each node $w_{in}$, length of walk paths $l_{in}$, BC decay coefficient $\zeta$, number of resampled walk paths on each node $w_{out}$, length of resampled walk paths $l_{out}$, distance decay coefficient $\alpha$, training epochs $m$, learning rate $\eta_r$, usage of fast query $\tau$, usage of boosting $\chi$.

**Output:** SP representation $Z$, predictor $g_\phi$, estimated distance $D$. (optional: regressors $b_1$, $b_2$, global representation $Z'$)

```
1  Def sim_DT_with_BC(G, L):
2      distance triplets T := list[]
3      approximate BCs γ := dict{v_i : 0, ∀v_i ∈ V}
4      for each landmark v_i ∈ L do
5          γ[v_i] ← 1
6          for each node v_j ∈ V reached by BFS from v_i do
7              append (v_i, v_j, D_ij) to T
8              γ[v_j] ← γ[v_j] + 1/D_ij
9          end
10     end
11     return T, γ
12 Def sim_BC_Walk(G, v_i, w_in, l_in, γ):
13     distance map D_i := dict{v_i : 0}
14     visit counter B := dict{v_j : 0, ∀v_j ∈ V}
15     for walk k from 0 to w_in do
16         visit sign set S_i := {v_i}
17         current node v_c := v_i
18         length c_i := 0
19         while c_i < l_in do
20             probabilities of the next candidate nodes
                   P̃_c^k := dict{}
21             for v_j ∈ N_c ∧ v_j ∉ S_i do
22                 P̃_c^k[v_j] ← γ[v_i] × (2 − tanh(ζ − B[v_j]))
23                 if v_j ∈ D_i then
24                     D_i[v_j] = min{D_i[v_j], D_i[v_c] + 1}
25                 else
26                     D_i[v_n] = D_i[v_c] + 1
27                 end
28             end
29             sample next v_n from normalized P̃_c^k
30             v_c ← v_n, S_i ← v_n
31             c_i ← c_i + 1
32             B[v_n] ← B[v_n] + 1
33         end
34     end
35     return D_i
```

```
36 Def cons_BCDR(G, d, L, w_in, l_in, ζ, w_out, l_out, α, m, η):
37     T, γ ← sim_DT_with_BC(G, L), P = list[]
38     for v_i ∈ V do
39         D_i ← sim_BC_Walk(G, v_i, w_in, l_in, γ)
40         probabilities of candidates nodes P̃_i = dict{}
41         for v_j ∈ D_i.keys do
42             P̃_i[v_j] := α^{D_i[v_j]} · γ[v_j]
43         end
44         for walk k from 0 to w_out do
45             sample walk path p_i of length l_out by normalized P̃_i
46             append p_i to P
47         end
48     end
49     maximize Equation 6 with Z_{N×d} and P by skip-gram.
50     define learnable distance predictor g_φ := (ℝ^d, ℝ^d) → ℝ
51     for each epoch from 0 to m do
52         L_MSE(v_a, v_b) :=
               [g_φ(Z_a, Z_b) − D_ab]^2 , ∀(v_a, v_b, D_ab) ∈ T
53         minimize L_MSE with φ at the learning rate η_r by SGD.
54     end
55     if χ then
56         define CatBoostRegressor b_1, b_2.
57         Z' := {Z'_i | Z'_i := list[D_ij]|_{j=0}^{|L|}, ∀v_j ∈ L}
58         Train b_1 by (Z'_a, Z'_b) → D_ab
59         Train b_2 by (Z'_a, Z'_b, b_1(Z'_a, Z'_b), g_φ(Z_a, Z_b)) → D_ab
60     end
61     return Z, g_φ [, Z', b_1, b_2]
62 Def query_BCDR(Q, Z, g_φ, τ [, E, Z']):
63     estimated distance D[v_a, v_b] = g_φ(Z_a, Z_b), ∀(v_a, v_b) ∈ Q
64     if χ then
65         D[v_a, v_b] = b_2(Z'_a, Z'_b, b_1(Z'_a, Z'_b), D[v_a, v_b])
66     end
67     if not τ then
68         D[v_a, v_b] ← 1, ∀(v_a, v_b) ∈ Q ∩ E
69     end
70     return D
```

---

### A.2.1 SIMULATION OF DISTANCE TRIPLETS & BC WALK

To simulate distance triplets (line 1 to 11), we perform BFS from a fixed number of landmarks $L$ and record their distance to each node on the graph (line 6 and 7). $L$ comprises nodes that are selected by heuristic strategies (e.g., by their degrees in descending order or randomly), and the simulated triplets with linear complexity reflect a sufficient group of distance relations among $V \times V$ for metric learning. Before simulating BC walk paths, pre-computed BC of each node is also required at first, which takes at least $O(NM)$ time on unweighted graphs for an exact solution. To reduce the time complexity, we consider a time-efficient approximation by integrating this process into the above simulation of distance triplets (line 8). Intuitively, $BC(v_a)$ measures some kind of relationship between $v_a$ and the centers of the graph, and nodes with larger BC values possess a shorter average SP distance to any node on the graph. Since BFS visits each node just in ascending order of distance, we thus estimate BC on each node by the average distance to all landmarks in $L$ without introducing extra time complexity.

Then, in the simulation of BC walk paths (line 12 to 35), we are interested in "nodes on these walk paths" instead of the full paths themselves, and the former with their distance to the root $v_i$ (i.e., $D_i$) will be passed to feed subsequent construction (line 39). Therefore, we enlarge the node coverage by a decay mechanism on BC to diverge different walks. Note that *BC-based random walk* possessing the ability to explore remote nodes tends to choose the paths that are prone to travel further, which causes ignorance of nodes on some dead ends. Addressing this, as stated in line 22, the probability of transiting to a node with large BC will be saturated after a sufficient number of walks passing through it, which means some rare paths are getting much easier to be visited later.

### A.2.2 Learning Embeddings & Distance Predictor

To learn node embeddings $Z$ on SP structures (line 37 to 49), we initially sample a group of BC walk paths as distance maps $\{D_i\}$ (line 39), and then resample from them to feed the skip-gram algorithm by considering distance decay and BC (line 42 and line 45), which preserves the property of distance relations discussed in Theorem 1. Besides, the resampling process also provides a shape transform of observed node sequences from $w_{in} \times l_{in}$ to $w_{out} \times l_{out}$. Let $\beta = (w_{in}l_{in})/(w_{out}l_{out}) > 1$, and this property leads the actual time cost of learning embeddings to be reduced to its $1/\beta$, compared with learning directly on the original paths.

For the learning of the distance predictor (line 37, and line 50 to 54), we utilize a two-layer fully connected neural network as the predictor, which takes the concatenation of two nodes' embeddings as input, and outputs a scalar to indicate the distance (line 50). The predictor model is learned from distance triplets $T$ by minimizing a mean squared error (MSE) between the predicted value and the real distance using the stochastic gradient descent (SGD) technique. Finally, the parameter of neural network $\phi$ and node embeddings $Z$ are stored as graph SP representations.

To improve the prediction results, we further integrate the distance predictor with CatBoost techniques (Dorogush et al., 2018). Initially, we treat the feature of nodes as a combination of *global features* and *local features*. Thereinto, *local features* are already represented by $Z$, since we have constructed SP representations on each node locally. For *the global feature* of any node $v_i$, we directly leverage the distance to each landmark as its global embedding $\mathbf{Z}'_i$. Then, we train two Cat-Boost regressors (i.e., $b_1$, $b_2$) in turn. The first regressor $b_1$ takes global features of two nodes as input and predicts their distance as output (line 58), while the second regressor $b_2$ takes as input not only such global features, but the distance predicted from both global (i.e., $b_1(\mathbf{Z}'_a, \mathbf{Z}'_b)$) and local (i.e., $g_\phi(\mathbf{Z}_a, \mathbf{Z}_b)$) features. Finally, the outputs of $b_2$ are regarded as the final prediction results of SP distance.

### A.2.3 Answering Distance Queries

For distance queries, We provide two versions with different properties based on the same SP representations. As revealed in previous studies of learning-based SP representation methods, the distance relations in local neighborhoods are really hard to converge, which causes a decline in average accuracy (Rizi et al., 2018; Kutuzov et al., 2019). For an input query pair $(v_a, v_b)$, it is helpful to alleviate such decline if we perform an extra search among first-order neighbors (i.e., search in the adjacency matrix) to judge if $(v_a, v_b)$ is an edge in the graph. Since this practice prolongs the total response time, we also preserve a fast version (BCDR-FQ) without neighbors searching for some potential applications.

### A.2.4 Parallelism

The implementation of BCDR is easy to be highly parallelized. In detail, the construction of SP representations could be divided into three parts, including simulating distance triplets, performing *BC-based random walk*, training embeddings, as well as the distance predictor. First, the BFS from each landmark could be parallelized at a thread level up to the size of the landmark set. Second, for the simulation of BC walk paths, the paths from different roots could also be simulated simultaneously. Third, the training process in the skip-gram algorithm and neural network could be locally parallelized by matrix computations.

### A.3 Theorem for Clarifying the Significance of BC-based Random Walk

**Theorem 2.** *Define $\mathcal{N}_a^{(h)} = \{v_j | D_{aj} = h\}$ as a set of nodes that are $h$-hops away from $v_a$ and $N_a^{(h)} = |\mathcal{N}_a^{(h)}|$ as the number of nodes in the set. Nodes in $\mathcal{N}_a^{(h)}$ could be divided into two sets, i.e., $e_h(v_a)$ and $f_h(v_a)$. Thereinto, $e_h(v_a)$ comprises nodes that have an edge with the nodes in $\mathcal{N}_a^{(h+1)}$ (called connective nodes), and $f_h(v_a)$ comprises the other nodes (called final nodes). The BC value of each node is approximated by considering only the shortest path of nodes within a range of $k$-hops locally. Let $\tilde{P}_R(\mathcal{N}_a^{(h)} \rightarrow \mathcal{N}_a^{(h+1)})$ represent the probability to transit from $\mathcal{N}_a^{(h)}$ to $\mathcal{N}_a^{(h+1)}$ by a naive random walk, and $\tilde{P}_B(\mathcal{N}_a^{(h)} \rightarrow \mathcal{N}_a^{(h+1)})$ represent that by a BC-based random walk. Let $\tilde{P}_R(f_h(v_a) \rightarrow e_h(v_a))$ be the probability of transition from $f_h(v_a)$ to $e_h(v_a)$. $\mathcal{E}(v_a)$ is the*

*eccentricity of $v_a$, $\mathcal{E}(v_a) = \max_{v_b \in G} D_{ab}$. Then, for sufficient large $\mathcal{E}(v_a)$, any node $v_a \in G$ and any $h > 1$,*

$$\frac{\tilde{P}_B(\mathcal{N}_a^{(h)} \to \mathcal{N}_a^{(h+1)})}{\tilde{P}_R(\mathcal{N}_a^{(h)} \to \mathcal{N}_a^{(h+1)})} = 1 + B(k) + C \tag{12}$$

$$\lim_{k \to \mathcal{E}(v_a)-1-h} B(k) + C = \frac{A_2 - 1}{A_1} + C > 0$$

$$A_1 = \frac{|e_h(v_a)|}{|f_h(v_a)|}, \quad A_2 = \frac{1}{\tilde{P}_R(f_h(v_a) \to e_h(v_a))}, \quad C \geq 0. \tag{13}$$

*Proof.* We simplify symbols $\mathcal{N}_a^{(h)}, N_a^{(h)}, e_h(v_a), f_h(v_a)$ as $\mathcal{N}_h, N_h, e_h, f_h$ for short.

$$\overrightarrow{N}_h = \#(\bigcup_{i=h}^{\min\{\mathcal{E}(v_a), h+k-1\}} \{v_j | v_j \in \mathcal{N}_a^{(i)}\})$$

$$\overleftarrow{N}_h = \#(\bigcup_{i=\max\{0, h-k+1\}}^{h} \{v_j | v_j \in \mathcal{N}_a^{(i)}\}) \tag{14}$$

where $\#(\cdot)$ is a counting function indicating the number of occurrence times of specified nodes, i.e., the cardinality of a sampled set.

According to the definition in Theorem 2, we have $N_h = |e_h| + |f_h|$. Since only nodes in $e_h$ could travel from $\mathcal{N}_h$ to $\mathcal{N}_{h+1}$, We firstly consider $\tilde{P}_R(e_h \to e_{h+1}|e_h \to \mathcal{N}_{h+1})$ and $\tilde{P}_B(e_h \to e_{h+1}|e_h \to \mathcal{N}_{h+1})$.

For general random walks, the choice of destination is based on uniform sampling, thus causing

$$\tilde{P}_R(e_h \to e_{h+1}|e_h \to \mathcal{N}_{h+1}) = \frac{|e_{h+1}|}{N_{h+1}} \tag{15}$$

For *BC-based random walk*, we need calculate BC value of nodes of $e_{h+1}$ and $f_{h+1}$ for starters. Let $BC(e_{h+1})$ and $BC(f_{h+1})$ represent the BC value of nodes in $e_{h+1}$ and $f_{h+1}$ respectively, and the correspond legal SPs counts come from 4 sources as $\{\overleftarrow{\mathcal{N}}_h \to \overrightarrow{\mathcal{N}}_{h+2}\}$, $\{\overleftarrow{\mathcal{N}}_h \to \mathcal{N}_{h+1}\}$, $\{\mathcal{N}_{h+1} \to \overrightarrow{\mathcal{N}}_{h+2}\}$ and $\{\mathcal{N}_{h+1} \to \mathcal{N}_{h+1}\}$. And we use $BC(\{\cdot \to \cdot\})$ as the BC gain from the specified source, then

$$BC(\{\overleftarrow{\mathcal{N}}_h \to \overrightarrow{\mathcal{N}}_{h+2}\}) = \overleftarrow{N}_h \overrightarrow{N}_{h+2}$$
$$BC(\{\overleftarrow{\mathcal{N}}_h \to \mathcal{N}_{h+1}\}) = \overleftarrow{N}_h(|f_{h+1}| \cdot 0 + |e_{h+1}|\beta_e^{(1)})$$
$$BC(\{\mathcal{N}_{h+1} \to \overrightarrow{\mathcal{N}}_{h+2}\}) = \overrightarrow{N}_{h+2}(|f_{h+1}| \cdot 1 + |e_{h+1}|\beta_e^{(1)})$$
$$BC(\{\mathcal{N}_{h+1} \to \mathcal{N}_{h+1}\}) = N_{h+1}^2 \beta_e^{(2)} \tag{16}$$

where $\beta_e^{(1)}$ means average BC gain between nodes in $e_{h+1}$ and $\overrightarrow{\mathcal{N}}_{h+2}$, and $\beta_e^{(2)}$ means average BC gain between nodes both in $e_{h+1}$, which are constantly related with $G$. Therefore, we have

$$BC(e_{h+1}) = \overleftarrow{N}_h \overrightarrow{N}_{h+2} + (\overleftarrow{N}_h + \overrightarrow{N}_{h+2})|e_{h+1}|\beta_e^{(1)} + \overrightarrow{N}_{h+2}|f_{h+1}| + N_{h+1}^2 \beta_e^{(2)} \tag{17}$$

Likewise, we calculate

$$BC(f_{h+1}) = \overleftarrow{N}_h \overrightarrow{N}_{h+2} \cdot 0 + \overleftarrow{N}_h(|f_{h+1}|\beta_f^{(1)} + |e_{h+1}| \cdot 0) + \overrightarrow{N}_{h+2}(|f_{h+1}|\beta_f^{(1)} + e_{h+1} \cdot 0) + N_{h+1}^2 \beta_e^{(2)}$$
$$= (\overleftarrow{N}_h + \overrightarrow{N}_{h+2})\beta_f^{(1)} + N_{h+1}^2 \beta_f^{(2)} \tag{18}$$

To compare the above $BC(e_{h+1})$ and $BC(f_{h+1})$,

$$\frac{BC(f_{h+1})}{BC(e_{h+1})} = \frac{(\overleftarrow{N}_h + \overrightarrow{N}_{h+2})\beta_f^{(1)} + N_{h+1}^2 \beta_f^{(2)}}{\overleftarrow{N}_h \overrightarrow{N}_{h+2} + (\overleftarrow{N}_h + \overrightarrow{N}_{h+2})|e_{h+1}|\beta_e^{(1)} + \overrightarrow{N}_{h+2}|f_{h+1}| + N_{h+1}^2 \beta_e^{(2)}} \tag{19}$$

note that for any $N_j$ and $N_{(x,y)} = \vec{N}_0 - \overleftarrow{N}_x - \vec{N}_y$ where $j, x, y \in \{0, \mathcal{E}(v_a)\}$,

$$\lim_{k \to \mathcal{E}(v_a)-1-h} \frac{N_j}{N_{(x,y)}} = \lim_{k \to \mathcal{E}(v_a)-1-h} \epsilon(k) = 0 \tag{20}$$

Equation 18 is reduced to

$$\frac{\text{BC}(f_{h+1})}{\text{BC}(e_{h+1})} = \frac{2\epsilon(k)\beta_f^{(1)} + \epsilon(k^2)\beta_f^{(2)}}{1 + 2\epsilon(k)n|e_{h+1}|\beta_e^{(1)} + \epsilon(k)|f_{h+1}| + \epsilon(k^2)\beta_e^{(2)}} = 2\beta_f^{(1)}\epsilon(k) \tag{21}$$

Then, we perform weighted random sampling based on BC and get

$$\tilde{P}_B(e_h \to e_{h+1}|e_h \to \mathcal{N}_{h+1}) = \frac{|e_{h+1}|}{|e_{h+1}| + 2|f_{h+1}|\beta_f^{(1)}\epsilon(k)} \tag{22}$$

Now, we consider the relation between $\tilde{P}_R(\mathcal{N}_h \to e_{h+1}|\mathcal{N}_h \to \mathcal{N}_{h+1})$ and $\tilde{P}_B(\mathcal{N}_h \to e_{h+1}|\mathcal{N}_h \to \mathcal{N}_{h+1})$.

$$\begin{aligned}
\frac{\tilde{P}_B(\mathcal{N}_h \to \mathcal{N}_{h+1})}{\tilde{P}_R(\mathcal{N}_h \to \mathcal{N}_{h+1})} &= \frac{\tilde{P}_B(e_h)\tilde{P}_B(e_h \to \mathcal{N}_{h+1})}{\tilde{P}_R(e_h)\tilde{P}_R(e_h \to \mathcal{N}_{h+1})} \\
&= \frac{\tilde{P}_B(\mathcal{N}_{h-1} \to e_h) + \tilde{P}_B(\mathcal{N}_{h-1} \to f_h)\tilde{P}_B(f_h \to e_h)}{\tilde{P}_R(\mathcal{N}_{h-1} \to e_h) + \tilde{P}_R(\mathcal{N}_{h-1} \to f_h)\tilde{P}_R(f_h \to e_h)} \\
&= 1 + \frac{\left[\tilde{P}_B(\mathcal{N}_{h-1} \to e_h) - \tilde{P}_R(\mathcal{N}_{h-1} \to e_h)\right]\left[1 - \tilde{P}_R(f_h \to e_h)\right]}{\tilde{P}_R(\mathcal{N}_{h-1} \to e_h) + \tilde{P}_R(\mathcal{N}_{h-1} \to f_h)\tilde{P}_R(f_h \to e_h)} \\
&\quad + \frac{\tilde{P}_R(f_h \to e_h)\epsilon}{\tilde{P}_R(\mathcal{N}_{h-1} \to e_h) + \tilde{P}_R(\mathcal{N}_{h-1} \to f_h)\tilde{P}_R(f_h \to e_h)} \\
&= 1 + \frac{|f_h|[1 - 2|f_h|\beta_f^{(1)}\epsilon(k)][1 - \tilde{P}_R(f_h \to e_h)]}{(1 + \frac{|f_h|}{|e_h|})[|e_h| + 2|f_h|\beta_f^{(1)}\epsilon(k)]\tilde{P}_R(f_h \to e_h)} \\
&\quad + \frac{\tilde{P}_R(f_h \to e_h)\epsilon}{\tilde{P}_R(\mathcal{N}_{h-1} \to e_h) + \tilde{P}_R(\mathcal{N}_{h-1} \to f_h)\tilde{P}_R(f_h \to e_h)}
\end{aligned} \tag{23}$$

Let $C = \frac{\tilde{P}_R(f_h \to e_h)\epsilon}{\tilde{P}_R(\mathcal{N}_{h-1} \to e_h) + \tilde{P}_R(\mathcal{N}_{h-1} \to f_h)\tilde{P}_R(f_h \to e_h)}$, $B(k) = \frac{|f_h|[1 - 2|f_h|\beta_f^{(1)}\epsilon(k)][1 - \tilde{P}_R(f_h \to e_h)]}{(1 + \frac{|f_h|}{|e_h|})[|e_h| + 2|f_h|\beta_f^{(1)}\epsilon(k)]\tilde{P}_R(f_h \to e_h)}$. Since $C$ is a non-negative value independent of $k$, finally we get

$$\frac{\tilde{P}_B(\mathcal{N}_h \to \mathcal{N}_{h+1})}{\tilde{P}_R(\mathcal{N}_h \to \mathcal{N}_{h+1})} = 1 + B(k) + C \tag{24}$$

where

$$\lim_{k \to \mathcal{E}(v_a)-1-h} B(k) + C = \frac{A_2 - 1}{A_1} + C. \tag{25}$$

$\square$

**Remark 1.** *Equations 12 and 13 in Theorem 2 give definite conditions under which* BC-*based random walk travels further than random walk. Since nodes in $v_a$'s h-order neighborhood $\mathcal{N}_a^{(h)}$ could always be divided into $e_h(v_a)$ and $f_h(v_a)$ like Figure 3a,* BC-*based random walk improves exploration distance beyond local loops and dead ends in contrast with naive random walk by two aspects. On one side, even if $\mathcal{N}_a^{(h)}$ comprises a larger number of nodes in $f_h(v_a)$ than $e_h(v_a)$ and thus $A_1$ decreases,* BC-*based random walk tends to transit in $e_h(v_a)$ to get near with the next desired set $\mathcal{N}_a^{(h+1)}$. On the other side, even if nodes in $f_h(v_a)$ have fewer links to those in $e_h(v_a)$ and thus $A_2$ increases,* BC-*based random walk tends to transit between $f_h(v_a)$ and $e_h(v_a)$ instead of looping within $f_h(v_a)$ as well as $\mathcal{N}_a^{(h-1)}$.*

A.4 PROOF OF PROPOSITION 1

*Proof.* In Proposition 1, we optimize an approximate objective $\hat{\mathcal{L}}_n$ instead of $\mathcal{L}_n$, i.e.,

$$\hat{\mathcal{L}}_n(Z) = \sum_{v_i \in V} \tilde{P}(v_i) \left\{ \mathbb{E}_{v_j \sim \tilde{Q}_{W_i}(V)} [\log \hat{\sigma}(\mathbf{Z}_i \mathbf{Z}_j^T)] + \lambda \mathbb{E}_{v_k \sim \tilde{P}_n(V)} [\log \hat{\sigma}(-\mathbf{Z}_i \mathbf{Z}_k^T)] \right\} \tag{26}$$

Here, we rewrite the negative sampling item of Equation 26 as

$$\mathbb{E}_{v_k \sim \tilde{P}_n(W_i)} [\log \hat{\sigma}(-\mathbf{Z}_i \mathbf{Z}_k^T)]) = \sum_{v_k \in W_i} \tilde{P}_n(v_k|v_i) [\log \hat{\sigma}(-\mathbf{Z}_i \mathbf{Z}_k^T)]$$
$$= \tilde{P}_n(v_j|v_i) [\log \hat{\sigma}(-\mathbf{Z}_i \mathbf{Z}_j^T)] + \sum_{v_k \in W_i \setminus \{v_j\}} \tilde{P}_n(v_k|v_i) [\log \hat{\sigma}(-\mathbf{Z}_i \mathbf{Z}_k^T)] \tag{27}$$

Then, for each pair of $v_i \in V$ and $v_j \in W_i$, we get independent objective by combing similar items from the total likelihood $\hat{\mathcal{L}}_n$, and reach

$$\hat{\mathcal{L}}_n(Z) = \sum_{v_i \in V} \sum_{v_j \in W_i} \mathcal{L}'(\mathbf{Z}_i, \mathbf{Z}_j)$$
$$\mathcal{L}'(\mathbf{Z}_i, \mathbf{Z}_j) = \tilde{P}(v_i, v_j) \log \hat{\sigma}(-\mathbf{Z}_i \mathbf{Z}_j^T) + \lambda \tilde{P}(v_i) \tilde{P}_n(v_j|v_i) \log \hat{\sigma}(-\mathbf{Z}_i \mathbf{Z}_j^T) \tag{28}$$

Let $\hat{D} = ZZ^T$, for each node pair $(v_a, v_b)$ with SP distance $\hat{D}_{ab}$, consider

$$\hat{D}_{ab} = \mathbf{Z}_a \mathbf{Z}_b^T = \arg \max_{\mathbf{Z}_a, \mathbf{Z}_b} \mathcal{L}'(\mathbf{Z}_a, \mathbf{Z}_b) \tag{29}$$

Remember that $\tilde{Q}_{W_i}$ is a distribution resampled from $\tilde{P}_{W_i}$, as an efficient approximation to $\tilde{P}_{\mathring{W}_i}$, i.e.,

$$\tilde{Q}_{W_i}(v_j) = \frac{\alpha^{D_{ij}} \mathrm{BC}(v_j)}{\sum_{v_k \in W_i} \alpha^{D_{ik}} \mathrm{BC}(v_k)} \tag{30}$$

Denote $\mathrm{BC}(v_b)$ by $\gamma_b$, according to Equation 30, the joint distribution of each pair $(v_a, v_b)$ could be formulated as

$$\tilde{P}(v_a, v_b) = \tilde{P}(v_a) \cdot \tilde{P}(v_b|v_a) = \tilde{P}(v_a) \cdot \alpha^{D_{ab}} \gamma_b \tag{31}$$

Then, consider the formualtion of $\tilde{P}_n(v_b|v_a)$. Recall that the negative sampling is a uniform distribution on $W_a$ which are simulated by *BC-based random walk*, and the probability of $v_b$'s occurrence relies on $\gamma_b$ and $\mathcal{W}_a$. Thus there holds

$$\tilde{P}_n(v_b|v_a) = \frac{\gamma_b}{\kappa(v_a)} \tag{32}$$

where $\kappa(v_a)$ is in proportion with the number of nodes covered by $\mathcal{W}_a$ on the graph.

Furthermore, Equation 29 could be rewritten as

$$\hat{D}_{ab} = \arg \max_{\hat{D}_{ab}} \tilde{P}(v_a) \alpha^{D_{ab}} \gamma_b \log \hat{\sigma}(\hat{D}_{ab}) + \frac{\lambda}{\kappa(v_a)} \tilde{P}(v_a) \gamma_b \log \hat{\sigma}(-\hat{D}_{ab}) \tag{33}$$

Solve the above problem by just let $\frac{\partial \mathcal{L}(v_a, v_b)}{\partial \hat{D}_{ab}}$ be equal to zero, i.e.,

$$\frac{\partial \mathcal{L}(v_a, v_b)}{\partial \hat{D}_{ab}} = \tilde{P}(v_a) \alpha^n \gamma_b \hat{\sigma}(1 - \hat{D}_{ab}) - \frac{\lambda}{\kappa(v_a)} p(v_a) \gamma_b \hat{\sigma}(1 + \hat{D}_{ab}) = 0 \tag{34}$$

After some simplification, we get

$$\hat{D}_{ab} = D_{ab} \log \alpha - \log \frac{\lambda}{\kappa(v_a)} \tag{35}$$

Let $\epsilon = \lambda \kappa^{-1}(v_a)$ and there holds

$$\hat{D}_{ab} = -\log \epsilon + D_{ab} \log \alpha \tag{36}$$

$\square$

## A.5 REMARK FOR THE RELATIONSHIP BETWEEN DISTANCE MATRIX & DISTANCE-BASED SIMILARITY MATRIX

**Remark 2.** *Equation 7 in Proposition 1 reveals the linear projections between elements in $\hat{D}$ and $D$. Thereinto, $-\log \epsilon$ is a big positive constant with respect to $|log|W_a|| - |\log \lambda|$, and $D_{ab} \log \alpha$ is a negative value that decreases linearly with SP distance $D_{ab}$. It indicates that there exists a finite distance range $n \in \mathbb{N}_+$, for each node $v_b \in \{v_x | D_{ax} \leq n\}$, the distance relation between $v_a$ and $v_b$ could be well-optimized by converging $Z_a Z_b^T \to \hat{D}_{ab} > 0$. It also reveals that the significance of* distance resampling *is to preserve the SP distance relations between nodes which are well-observed on given arbitrary walk paths. Besides, although the similarity matrix $\hat{D}$ could not directly tell the absolute distance, it also shares similar properties with $D$, i.e., if we fix $v_a$ as the source node in a path, the comparison between the similarities of $(v_a, v_b)$ and $(v_a, v_c)$ just reflects the SP distance relations between them. This practical property is discussed in Theorem 1.*

## A.6 PROOF OF THEOREM 1

*Proof.* In terms of node pair $(v_a, v_b)$, as proved in Proposition 1, their similarity $\hat{D}_{ab}$ in the embedding space varies linear with respect to the SP distance $D_{ab}$ on the graph, i.e.,

$$\hat{D}_{ab} = -\log \epsilon + D_{ab} \log \alpha \tag{37}$$

Likewise, for $(v_a, v_c)$, there holds

$$|\hat{D}_{ac}| = -\log \epsilon + D_{ac} \log \alpha \tag{38}$$

where $0 < \alpha < 1$ and $\epsilon$ relies on $W_a$ which is independent of $v_b$ and $v_c$.

Then, consider the distance relation of $v_a$, $v_b$ and $v_c$, there holds

$$(D_{ab} - D_{ac})(\hat{D}_{ab} - \hat{D}_{bc}) = (D_{ab} - D_{ac})((D_{ab} - D_{ac})) \log \alpha = \log \alpha \cdot (D_{ab} - D_{ac})^2 \leq 0. \tag{39}$$

$\square$

## A.7 MOTIVATION OF BCDR PROCEDURE AGAINST DIRECTLY SAMPLING SPs

We further clarify in this section the motivation for leveraging BCDR instead of directly sampling SPs. As a prerequisite, it should be acknowledged that we need sampled SPs as *observation* to optimize node embeddings $Z$. However, to perform sufficient *observation* on all pairs of shortest paths is time-consuming, which takes at least $O(N^2)$ time on sparse unweighted graphs. Towards this, an intuitive idea is to sample a limited number of paths that starts only at a few nodes (landmarks). But it will introduce strong bias on the landmarks and ignore many shortest paths far away from them. To alleviate this bias, in BCDR, we hope to observe shortest paths rooted at all nodes on the graph (instead of the landmarks only). Therefore, we need some strategies to overcome the huge complexity of directly sampling these paths (since it requires performing BFS on all nodes). The proposed strategy is *BC-based random walk* where we intend to equip 'random walk' with the awareness of high-order SP structure and make the sampled walk paths much more likely to be certain shortest paths. This strategy is comparatively efficient since the sampling complexity is proportional to its path length $l$. Then the subsequent module DR further resampled from these paths for implicitly preserving SP distance relations on $Z$.

According to the above discussion, a brief procedure of BCDR with its motivation could be summarized as follows.

- estimate BC just by BFS from only a few nodes (landmarks).
  **motivation**: determine which node is prone to trigger high-order explorations of SP distances.

- perform *BC-based random walk*.
  **motivation**: observe the potential shortest paths rooted at each node sufficiently.

- leverage DR for resampling approximate random shortest paths.
  **motivation**: implicitly preserving distance relations on observed paths.

- optimize $Z$ from the observation of the resampled paths.
  **motivation**: reflect the SP structure on the graph instead of arbitrary linkage.

Each step above possesses linear complexity with respect to $N$ (number of nodes in the graph).

Besides, we are convinced of the necessity of BCDR procedure and would like to explain it carefully from both technical and empirical perspectives.

From a technical perspective, directly leveraging shortest paths as observation to optimize $Z$ has a few shortcomings.

- **Prohibitive Complexity of Sufficient Observation**. Observing all pairs of SP distance requires at least $O(N^2)$ time for sparse unweighted graphs. Alternatively, an insufficient observation with linear complexity will cause a loss in accuracy (see experimental results below).

- **Inflexible Path Length for Optimization**. Since we leverage the skip-gram algorithm for optimizing $Z$, it should be clear how long the sliding window size is, serving to reconstruct the distance relations between nodes. But shortest paths rooted at a certain node factually possess significantly divergent path lengths, which causes difficulty in determining proper sliding window size on different paths, i.e., a longer window helps to capture long-distance correlation but causes indistinguishable in shorter paths and vice versa. Alternatively, if we only select shortest paths of a certain fixed length, paths shorter than this length will be ignored, thus impairing the performance.

Correspondingly, the BCDR procedure overcomes the above shortcomings as follows.

- **Linear Complexity of Such Observation**. Instead of directly simulating shortest paths, we sample paths by *BC-based random walk* and transform the paths into approximate random shortest paths by DR. Both of these operation share linear time complexity. Also, the optimization on such resampled paths is proved to share similar properties with that on real shortest paths by Proposition 1 and Theorem 2.

- **Flexible Path Length for Optimization**. Since the paths are resampled from random paths, the number and length of them (i.e., $w_{out}, l_{out}$) could be customized. We are thus able to fix them at a certain proper length for subsequent optimization.

From an empirical perspective, we further construct and evaluate 6 competitive baselines which have the same architecture and hyper-parameters as BCDR, but adopt different intuitive strategies to directly optimize on shortest paths. The basic description of these baselines is stated as follows.

- **Shortest Paths on Landmarks only (SPoL)** Since we need anyway perform BFS from landmarks to acquire distance triplet for learning distance predictor, we intuitively retrieve the shortest paths starting from the landmarks. This operation introduces little extra time cost. The size of landmark set is the same as BCDR (i.e., $|L| = 80$)

- **Shortest Paths on Landmarks only with Fixed Length (SPoL-F)** This is similar to SPoL but restricts the output walk length at a certain level (the same as BCDR, i.e., $l_{out} = 10$).

- **Shortest Paths on All Nodes (SPoN)** In BCDR, we perform BC random walk on each node $v_a$ to locate its position on the graph. Here, we directly sample shortest paths from $v_a$ to any other nodes instead. Specifically, for each source node $v_a$, we take a uniform sampling over $V$ to acquire the destination nodes, and retrieve the shortest paths between them. The number and max length of shortest paths on each node is the same as BCDR (i.e., $w_{out} = 40, l_{out} = 10$).

- **Shortest Paths on All Nodes with Fixed Length (SPoN-F)** This is similar to SPoN but restricts the output walk length (the same as BCDR, i.e., $l_{out} = 10$).

- **Shortest Paths on Arbitrary Node Pair (SPoANP)** We randomly select a group of node pairs $(v_s, v_t)$ and retrieve one of the shortest paths between them by BFS. The number of paths is the same as the total number of walk paths on all nodes in BCDR (i.e., $N \times w_{out}$).

- **Shortest Paths on Arbitrary Node Pair (SPoANP-F)** This is similar to SPoANP but restricts the output walk length (the same as BCDR, i.e., $l_{out} = 10$).

All of the above baselines are evaluated on GrQc dataset, and the experimental results are presented in Table 3.

Table 3: Comparisons between BCDR and directly sampling SPs by different intuitive strategies. PT: pre-processing time, ST: time of sampling paths, mAE: mean of Absolute Error, mRE: mean of Relative Error.

| Model | PT | ST | mAE | mRE |
|---|---|---|---|---|
| SPoL | 60.71 s | **15.36s** | 0.9703 | 0.1641 |
| SPoL-F | **60.46s** | 15.63 s | 1.2874 | 0.1961 |
| SPoN | 283.0 s | 235.5 s | 1.0411 | 0.1645 |
| SPoN-F | 2,702 s | 2,656 s | 1.2961 | 0.1969 |
| SPoANP | 282.1 s | 234.4 s | 1.3564 | 0.2047 |
| SPoANP-F | 6,341 s | 6,290 s | 1.3294 | 0.2003 |
| BCDR | 69.61 s | 27.66 s | **0.7043** | **0.1274** |

We see from the table that BCDR outperforms all the baselines in approximation quality (i.e., mAE and mRE) within proper time. Specifically, SPoL possesses desirable pre-processing time since only the shortest paths rooted at landmarks are considered. But they are plagued with insufficient observation of other shortest paths that do not pass through landmarks. SPoN and SPoANP suffer huge complexity when retrieving shortest paths on the whole graph, and perform even worse due to the uncertainty of reasonable sliding window size. From SPoL-F, SPoN-F, and SPoANP, we see that even if the path length is fixed, some uncaptured shorter paths will also cause a loss in accuracy.

## A.8 DATASETS

To thoroughly evaluate our proposed method, we conduct experiments on real-world graphs and synthetic graphs with divergent properties on sizes, structures, diameters, etc. Thereinto, real-world graphs are extracted from Stanford Large Network Dataset Collection (Leskovec & Krevl, 2014), and synthetic graphs are simulated according to specific rules described in A.8.6. The visualization results of each graph are illustrated in Figure 6, and the corresponding statistics are presented in Table 4. In the experiments, we show the efficiency and scalability of BCDR on real-world graphs of different sizes and test on smaller synthetic graphs with different structures for further analysis of exploration range and distance preservation. Here are brief descriptions of these graphs:

### A.8.1 CORA

This is a graph that describes the citation relationship of papers, which contains 2708 nodes and 10556 directed edges among them. Each node also has a predefined feature with 1433 dimensions.

### A.8.2 FACEBOOK

This is a graph that describes the relationship among Facebook users by their social circles (or friend lists), which is collected from a group of test users. Facebook has also encoded each user with a reindexed user ID to protect their privacy.

### A.8.3 GRQC

This is a graph recorded from the e-print arXiv in the period from January 1993 to April 2003, which represents co-author relationships based on their submission. Each undirected edge $(v_i, v_j)$ represents that an author $v_i$ is co-authored a paper with another author $v_j$. If one paper is owned by $k$ authors, a complete subgraph of $k$ nodes is generated correspondingly.

### A.8.4 DBLP

This is a graph collected as a computer science bibliography that provides a comprehensive list of research papers in computer science. As an undirected collaboration network, each edge reflects the corresponding two authors who have at least one paper together.

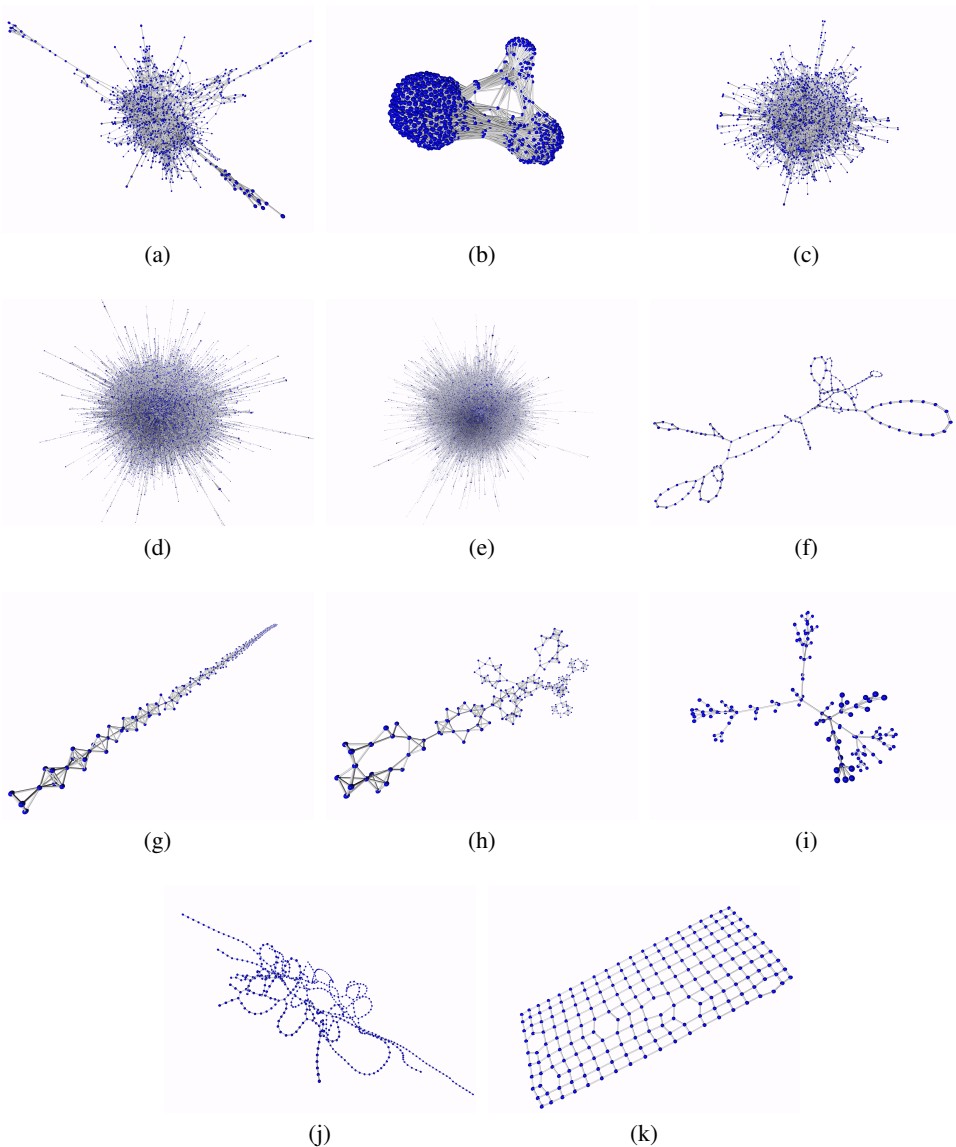

Figure 6: Visualization results of the graphs used for evaluation. **(a):** Cora. **(b):** Facebook. **(c):** GrQc. **(d):** DBLP. **(e):** YouTube. **(f):** CG. **(g):** TG. **(h):** TCG. **(i):** TRG. **(j):** SG. **(k):** NG.

### A.8.5 YOUTUBE

This is a graph constructed from users' social relations on a video-sharing website Youtube. Each node represents a user, and each edge indicates a friendship between two users.

### A.8.6 SYNTHETIC GRAPHS

We also construct some smaller graphs reflecting one or some of the typical sub-structures which are frequently occurred in complex graphs. The simulation rules of each graph are listed as follows.

- Circle Graph (CG): this is a graph that contains several circles of different sizes. The simulation of circle graphs takes an iterative process where for each newly introduced circle, there are a limited number of nodes (called exit nodes) connected to the previous circles.

Table 4: Statistics of the graphs used for evaluation. $N$ denotes the number of nodes, $M$ denotes the number of edges, RoBC denotes the range of BC, mBC denotes the average BC on nodes, $\mathcal{D}$ denotes the diameter of a graph, BFS denotes the average processing time of breadth-first search on all nodes. Each graph is considered an undirected and unweighted graph in the SP representation problem.

|  | $N$ | $M$ | $M/N$ | $\mathcal{D}$ | RoBC | mBC | BFS |
|---|---|---|---|---|---|---|---|
| Cora | $2,708$ | $10,787$ | $3.9834$ | $21$ | $375.20$ | $2.7174$ | $0.0074$ s |
| Facebook | $4,039$ | $176,437$ | $21.846$ | $8$ | $1,306.9$ | $0.8993$ | $0.0191$ s |
| GrQc | $5,242$ | $30,042$ | $5.7310$ | $17$ | $148.43$ | $2.7219$ | $0.0148$ s |
| DBLP | $317,080$ | $1,049,866$ | $3.3110$ | $21$ | $1,140.9$ | $2.0795$ | $46.0907$ s |
| YouTube | $1,134,890$ | $5,975,248$ | $5.2650$ | $20$ | - | - | $691.0986$ s |
| CG | $197$ | $216$ | $1.0964$ | $46$ | $0.6058$ | $0.1032$ | - |
| TG | $384$ | $984$ | $2.5625$ | $100$ | $0.4987$ | $0.0928$ | - |
| TCG | $242$ | $594$ | $2.4545$ | $31$ | $0.4891$ | $0.0551$ | - |
| TRG | $134$ | $133$ | $0.9925$ | $12$ | $0.6416$ | $0.0630$ | - |
| SG | $550$ | $585$ | $1.0636$ | $85$ | $0.3068$ | $0.0528$ | - |
| NG | $200$ | $364$ | $1.8200$ | $28$ | $0.2914$ | $0.1215$ | - |

- Triangle Graph (TG): this is a graph possessing several cliques which are linearly connected mutually.

- Tri-circle Graph (TCG): this is a graph that combines the properties of circle graphs and triangle graphs. Here, each circle is simulated by connecting triangle sub-graphs end to end.

- Tree Graph (TRG): this is a graph that is generated from one root to several leaves recursively. There is no cycle in tree graphs. To control the tree structure, we define a splitting probability that is decayed exponentially with current depth.

- Spiral Graph (SG): this is a graph shaped like a spiral line. We first simulate a line graph and add edges between nodes with exponentially increased distances by their indices on the line.

- Net Graph (NG): this is a graph containing grid-like connections between nodes. We define a small probability of dropping those edges stochastically.

## A.9 Baseline & Parameter Setup

The parameter setups of each baseline are listed as follows. For the oracle-based method, $\alpha_0$ is set to 2 for the best accuracy, as discussed in the previous work. For the landmark-based method, we choose a sufficient size of the landmark set as $|L| = 128$ and take the constrained strategy, i.e., for each landmark selected, nodes within two hops are discarded from consideration. For learning-based methods, the embedding size $d$ is fixed at 16. In addition, the number of selected landmarks in learning-based methods is up to 80 for small graphs (i.e., Cora, Facebook, and GrQc) and 24 for large graphs (i.e., DBLP and Youtube). Other hyper-parameters of each model follow the default configurations discussed in their works. For the baselines proposed in road networks, the coordinate-related features are omitted in their models, since there's no coordinate assumption in our graph datasets. For general GRL methods, all of the baselines follow the default configurations and are further trained by linear regression to extract the distance between any two nodes.

For our proposed method, we simulate $w_{in} = 20$ walks on each node, and each walk is truncated at a length of $l_{in} = 40$. Each landmark is selected randomly up to the size of a landmark set $|L| = 80$. The number of negative samples $n$ is set to 1. The process of distance resampling outputs $w_{out} = 40$ walks with each walk at a length of $l_{out} = 10$. The decay coefficients of BC and distance are fixed as $\zeta = 10$, $\alpha = 0.35$. We train the distance predictor using a two-layer perceptron with a learning rate $\epsilon_r = 0.01$ for 15 epochs and train the CatBoost regressors with a grid search for their best parameters at the offline stage. For large graphs (i.e., DBLP and YouTube), we adjust the above parameters by $|L| = 5, w_{in} = 2$, $\zeta = 1.0$. For BCDR-FC, the number of walks is reduced by a half for every graph. For BCDR-FQ, we take the raw outputs of the distance predictor without searching

Table 5: Parameter settings of evaluation on 5 real-world graphs. Smaller graphs include Cora, FaceBook, and GrQc. Larger Graphs include DBLP and YouTube.

| Parameters | for Smaller Graphs | for Larger Graphs |
|---|---|---|
| $d$ | 16 | 16 |
| $|L|$ | 80 | 5 |
| $w_{in}$ | 20 | 2 |
| $l_{in}$ | 40 | 40 |
| $w_{out}$ | 40 | 40 |
| $l_{out}$ | 10 | 10 |
| $\zeta$ | 10 | 1 |
| $\alpha$ | 0.35 | 0.35 |
| epochs | 15 | 15 |

first-order neighborhoods on the graph (i.e., set $\tau = $ True). The boosting module is only utilized in BCDR and disabled in BCDR-FQ and BCDR-FC (i.e. set $\chi = $ False). We summarized the critical parameter setting to reproduce results in Table 2 as follows.

A.10   EXTENDED COMPARISONS WITH GRL MODELS ON APPROXIMATION QUALITY

We present in Table 6 the experimental results of comparisons to general GRL models. Here, only the approximation quality (i.e., mAE and mRE) is evaluated, and the metrics are exerted directly on the representations without quantizing the outputs to integers or checking adjacency matrices. We see from Table 6 that although general embeddings by GRL methods could preserve some local SP structures, our proposed method with explicit SP constraint possesses better approximation quality for the SP distance queries.

Table 6: Extended comparison to general GRL models on approximation quality

| Model | Cora | | Facebook | | GrQc | |
|---|---|---|---|---|---|---|
| | mAE | mRE | mAE | mRE | mAE | mRE |
| LLE (Roweis & Saul, 2000) | 5.6265 | 0.8445 | 1.9921 | 0.6841 | 4.8849 | 0.7105 |
| LE (Roweis & Saul, 2000) | 5.6393 | 0.8455 | 2.0312 | 0.6998 | 5.0046 | 0.7366 |
| GF (Ahmed et al., 2013) | 5.6249 | 0.8440 | 1.8743 | 0.6383 | 4.8562 | 0.7125 |
| DeepWalk (Perozzi et al., 2014) | 1.5183 | 0.2425 | 0.9323 | 0.3289 | 2.8002 | 0.4169 |
| GraRep (Shaosheng et al., 2015) | 2.6206 | 0.3830 | 2.8702 | 1.0479 | 4.2445 | 0.6292 |
| Node2Vec (Grover & Leskovec, 2016) | 1.3072 | 0.2115 | 0.8541 | 0.2993 | 1.5156 | 0.2278 |
| NetMF (Qiu et al., 2018) | 4.1736 | 0.6025 | 1.6982 | 0.6163 | 3.8799 | 0.5779 |
| VERSE (Tsitsulin et al., 2018) | 2.8895 | 0.4049 | 1.1092 | 0.3729 | 3.3436 | 0.4689 |
| LPCA (Chanpuriya et al., 2020) | 2.2813 | 0.3337 | 2.1373 | 0.8611 | 2.4526 | 0.3475 |
| BCDR (ours.) | **0.9768** | **0.1605** | **0.4804** | **0.1770** | **1.0490** | **0.1684** |

A.11   FURTHER INVESTIGATION ON BCDR FRAMEWORK

A.11.1   ABLATION STUDY OF BCDR FRAMEWORK ON APPROXIMATION QUALITY

We discuss here the impact on approximation quality of different components in BCDR framework. In addition to those plausible post-processing operations described in Algorithm 1 (i.e., enable $\tau$, $\chi$ or not), we also explore other operations that influences approximation quality when pre-processing graphs. The modifications to BCDR are stated as follows, and the corresponding results evaluated on Facebook and DBLP are shown in Table 7.

- *no checks on adjacency*. The outputs of BCDR are accepted as predictions of SP distance without checking if there is any immediate edge between each node pair (i.e., set $\tau = $ True).
- *no global features*. SP distances are solely predicted by the two-layer neural network, and the boosting module based on global distances to landmarks is omitted (i.e., set $\chi = $ False).

- *no local features*. SP distances are solely predicted by the boosting module, and the learning process of the two-layer neural network on local features is omitted.
- *no BC*. Node representations are learned without *BC-based random walk*. For any nodes, each transition in simulating walks considers its first-order neighbors equally, ignoring their BC values.
- *no DR*. Node representations are learned without *distance resampling*. The resampling rule (i.e. Equation 5) is replaced by a uniform sampling.
- *degree selection*. In simulation of distance triplets and BC values, the landmarks are selected in descending order of degree, instead of random selection.

We see from the table that the full approach of BCDR achieves the best performance on approximation quality. It also reveals that all of the components significantly improve the prediction results. Specifically, for the checks on adjacency, it is notable that learning-based methods on SP representation show much difficulty in catching distance to first-order neighbors. Checking adjacency of input node pairs is necessary for accurate SP prediction. For global and local features, SP distance predicted from global features performs better than that from local features. It means leveraging global distances to each landmark helps a lot in locating a node on the graph. Furthermore, BCDR combines both global and local features for prediction and shows superior performance compared to either of them. For representing local features (i.e., *BC* and *DR*), we see that both *BC-based random walk* and *distance resampling* help to enhance the node representations with high-order SP structures, making it easier to extract distances to remote nodes. For landmark selection, we find that random selection of landmarks is more necessary for BCDR than other existing strategies. This is because we need to estimate BC values by performing BFS from these landmarks, and any assumption on landmark distribution will lead to unfair numerical estimation. If only landmarks with large degrees are selected, the BC value of nodes located in dense regions will be over-estimated, which impairs the efficiency of *BC-based random walk*.

Table 7: Ablation study of BCDR framework on approximation quality

| Model | Facebook | | DBLP | |
|---|---|---|---|---|
| | mAE | mRE | mAE | mRE |
| BCDR - *no checks on adjacency* | 0.0253 | 0.0180 | 0.5677 | 0.0907 |
| BCDR - *no global features* | 0.1138 | 0.0378 | 1.0070 | 0.1484 |
| BCDR - *no local features* | 0.0453 | 0.0171 | 0.5385 | 0.0855 |
| BCDR - *no BC* | 0.0210 | 0.0086 | 0.5437 | 0.0839 |
| BCDR - *no DR* | 0.0285 | 0.0142 | 0.5093 | 0.0808 |
| BCDR - *degree selection* | 0.3820 | 0.1139 | 1.1524 | 0.1611 |
| **BCDR - *full approach*** | **0.0106** | **0.0044** | **0.4923** | **0.0798** |

### A.11.2 FURTHER INVESTIGATION ON CRITICAL PARAMETER SETTING OF BCDR

Then, we further investigate the parameter settings of BCDR and discuss 9 critical parameters for their impacts on performance. Notably, although we describe rather detailed settings of parameters in Appendix A.9, the proposed method BCDR is factually robust and effective, and its performance does not sensitively rely on any one of them. Here, we show the impacts of these parameters on related metrics and how to easily tune them in any unweighted graphs, both conceptually and practically. The next discussion and evaluation of each parameter follow its order in Table 5.

$d$: **the dimension of node-level embeddings (i.e., $Z$).** In our experiment, $d$ is not a fine-tuned parameter but fixed at a certain value (i.e., $d = 16$) among different models to fairly evaluate their performance. This parameter could improve the performance on accuracy since a large size of embeddings could dump more valuable information about SP structures at the expense of higher storage cost and deficiency in query speed. To verify this, We test BCDR with different $d = \{2, 4, 16, 64, 128, 256\}$ on Facebook and GrQc to evaluate their performance under these metrics.

From the Table 8 and 9, we see that the accuracy loss could be cut down by increasing $d$, but it will lead to significant deterioration in storage cost and query speed. As we discuss a low-dimensional

and accurate SP representation in this paper, the results also reveal that even at a rather lower dimension of embeddings (like $d = 4$), the distance relations on the graph could be well-preserved.

Table 8: Impacts of $d$ on the performance of BCDR evaluated on FaceBook

| d | 2 | 4 | 16 | 64 | 128 | 256 |
|---|---|---|---|---|---|---|
| Storage | **0.1307 MB** | 0.1924 MB | 1.210 MB | 3.160 MB | 5.944 MB | 8.579 MB |
| mAE | 0.0902 | 0.0150 | 0.0202 | 0.0499 | 0.0310 | **0.0130** |
| mRE | 0.0347 | 0.0075 | 0.0091 | 0.0212 | 0.0122 | **0.0064** |
| Query Time | **4,089 ns** | 4,664 ns | 8,334 ns | 22,430 ns | 41,142 ns | 81,188 ns |

Table 9: Impacts of $d$ on the performance of BCDR evaluated on GrQc

| d | 2 | 4 | 16 | 64 | 128 | 256 |
|---|---|---|---|---|---|---|
| Storage | **0.2240 MB** | 0.3023 MB | 0.7916 MB | 2.750 MB | 5.415 MB | 10.86 MB |
| mAE | 0.8719 | 0.8954 | 0.7089 | 0.6867 | **0.6746** | 0.677 |
| mRE | 0.1548 | 0.1555 | 0.1259 | 0.1227 | **0.1201** | 0.1209 |
| Query Time | **2,421 ns** | 3,011 ns | 6, 570 ns | 20,837 ns | 39, 479 ns | 79,023 ns |

$|L|$: **the number of landmarks for constructing distance triplet and estimating BC.** This parameter mainly affects accuracy and pre-processing time since involving more landmarks helps to alleviate harmful inductive bias on a certain part of the graph but suffers higher computing overhead. It is also observed in the previous works (Rizi et al., 2018) that for large graphs with strong centrality on a few nodes, the number of landmarks could be reduced without much loss of accuracy. We evaluate BCDR with a group of landmarks ($|L| = \{10, 20, 40, 80, 160\}$) on Facebook and GrQc, to see their impacts on the two metrics.

Table 10: Impacts of $|L|$ on the performance of BCDR evaluated on FaceBook

| $|L|$ | 10 | 20 | 40 | 80 | 160 |
|---|---|---|---|---|---|
| Pre-processing Time | **127.8 s** | 134.3 s | 142.5 s | 157.5 s | 187.5 s |
| mAE | 0.0342 | 0.0297 | 0.0148 | 0.0193 | **0.0124** |
| mRE | 0.0134 | 0.0108 | 0.0063 | 0.0096 | **0.0062** |

Table 11: Impacts of $|L|$ on the performance of BCDR evaluated on GrQc

| $|L|$ | 10 | 20 | 40 | 80 | 160 |
|---|---|---|---|---|---|
| Pre-processing Time | **47.47 s** | 52.95 s | 64.35 s | 83.75 s | 123.7 s |
| mAE | 0.9922 | **0.6837** | 0.7383 | 0.7112 | 0.7065 |
| mRE | 0.1591 | **0.1185** | 0.1266 | 0.1217 | 0.1231 |

The results in Table 10 and 11 show that the pre-processing time on graphs increases linearly with $|L|$ since performing BFS from the added landmarks needs extra traversal on the whole graph for $O(N + M)$ time. It is also interesting to see that the number of landmarks large enough for the best performance diverges for dense and sparse graphs, i.e., it generally takes more than $40$ landmarks for Facebook but only $20$ landmarks necessary for GrQc. Specifically, for relatively dense graphs (i.e., Facebook), each node shares weaker centrality due to the enriched links, which means we need to observe more landmarks to cover more SPs on the graphs (according to the hub-labeling theory in (Cohen et al., 2003)). But for sparse graphs (i.e., GrQc), as long as several nodes with strong centrality are well-observed, SP distance between most node pairs could be preserved, resulting in tolerance of reduced landmarks.

$w_{in}, l_{in}$: **the number and length of sampled BC walks on each node.** These parameters affect the accuracy and pre-processing time. When we simluate BC walks rooted at a certain node, a large $w_{in}$ makes it sufficient to observe the local structure of each node (like BFS), while a large

$l_{in}$ allows wider exploration on the graph to let the distance with remote nodes be seen (like DFS). Like the previous evaluation, we test $w_{in} = \{5, 10, 20, 30, 40\}$ and $l_{in} = \{5, 10, 20, 40, 60, 80\}$ to investigate their impacts, respectively.

Table 12: Impacts of $w_{in}$ on the performance of BCDR evaluated on FaceBook

| $w_{in}$ | 5 | 10 | 20 | 30 | 40 |
|---|---|---|---|---|---|
| Pre-processing Time | **128.5 s** | 143.7 s | 186.3 s | 228.2 s | 273.5 s |
| mAE | 0.0136 | 0.0454 | 0.0113 | 0.0188 | **0.0062** |
| mRE | 0.0061 | 0.0182 | 0.0056 | 0.0093 | **0.0027** |

Table 13: Impacts of $l_{in}$ on the performance of BCDR evaluated on FaceBook

| $l_{in}$ | 5 | 10 | 20 | 40 | 60 | 80 |
|---|---|---|---|---|---|---|
| Pre-processing Time | 126.9 s | 131.6 s | 145.9 s | 182.8 s | 215.4 s | **114.0 s** |
| mAE | 0.0133 | 0.0145 | 0.0725 | **0.0081** | 0.0320 | 0.0370 |
| mRE | 0.0065 | 0.0067 | 0.0304 | **0.0037** | 0.0132 | 0.0184 |

Table 14: Impacts of $w_{in}$ on the performance of BCDR evaluated on GrQc

| $w_{in}$ | 5 | 10 | 20 | 30 | 40 |
|---|---|---|---|---|---|
| Pre-processing Time | **112.2 s** | 114.5 s | 118.8 s | 125.1 s | 130.2 s |
| mAE | 0.7227 | 0.6671 | **0.6581** | 0.6930 | 0.7115 |
| mRE | 0.1250 | 0.1195 | **0.1166** | 0.1245 | 0.1259 |

Table 15: Impacts of $l_{in}$ on the performance of BCDR evaluated on GrQc

| $l_{in}$ | 5 | 10 | 20 | 40 | 60 | 80 |
|---|---|---|---|---|---|---|
| Pre-processing Time | **114.0 s** | 115.7 s | 117.8 s | 119.3 s | 119.7 s | 120.8 s |
| mAE | 0.7460 | **0.6765** | 0.7074 | 0.6643 | 0.6895 | 0.6926 |
| mRE | 0.1280 | **0.1146** | 0.1217 | 0.1171 | 0.1227 | 0.1230 |

The experimental results from Table 12 to 15 show the accuracy of BCDR is not sensitive to these parameters, owing much to the efficiency of BC walk and well-preserved distance relations by DR. Intuitively, we recommend setting $l_{in}$ proportional to the diameter of the graph, which makes the whole graph observed from any nodes. Also, $w_{in}$ could be reduced when the connectivity on the graph is relatively weak since the local structures are quite simple to explore.

$w_{out}, l_{out}$**: the number and length of resampled paths (by DR) on each node.** These parameters control the shape of output node sequences to subsequently optimize $Z$ under a skip-gram procedure. To avoid much loss of information and preserve the correlation in BC walks, we intend to keep the scale of outputs similar to that of inputs, i.e., $w_{out}l_{out} = \Omega(w_{in}l_{in})$. To accelerate the optimization process, we could further shorten $l_{out}$ and keep this scale (by correspondingly expanding $w_{out}$). Note that this reshaping operation does not apparently change the locality nor impair the performance since DR resamples nodes from high-order neighborhoods with respect to their distance from the root, thus resulting in well-defined convergence, as shown in Prop. 1. In the experiment, we fix the scale of output node sequences as half of the scale of BC walks (i.e., $w_{out}l_{out} = w_{in}l_{in}/2 = 400$), and test different combinations of their settings as $(w_{out}, l_{in}) = \{(200, 2), (100, 4), (50, 8), (40, 10), (25, 16), (16, 25), (10, 40), (8, 50), (4, 100), (2, 200)\}$.

The results in Table 16 and 17 reveal that the pre-processing time dramatically increases along with $l_{out}$. This is because we utilize the whole sequence to optimize co-occurrence likelihood between the root and nodes in this sequence, which requires joint training with a large number of node embeddings proportional to $l_{out}$. It is also shown that the accuracy does not significantly fluctuate as pre-processing time, indicating a relatively small $l_{out}$ will help to reduce the off-line time cost.

Table 16: Impacts of $(w_{out}, l_{out})$ on the performance of BCDR evaluated on FaceBook

| $(w_{out}, l_{out})$ | (200,2) | (100,4) | (50,8) | (40,10) | (25,16) | (16,25) | (10,40) | (8,50) | (4,100) | (2,200) |
|---|---|---|---|---|---|---|---|---|---|---|
| Pre-processing Time | 176.3 s | 154.1 s | **148.6 s** | 150.6 s | 156.8 s | 170.5 s | 206.3 s | 236.8 s | 501.0 s | 1,521 s |
| mAE | 0.0237 | 0.0217 | 0.0249 | 0.0258 | 0.0303 | 0.0128 | 0.0127 | 0.0360 | **0.0083** | 0.0237 |
| mRE | 0.0097 | 0.0093 | 0.0107 | 0.0108 | 0.0136 | 0.0059 | 0.0043 | 0.0145 | **0.0041** | 0.0098 |

Table 17: Impacts of $(w_{out}, l_{out})$ on the performance of BCDR evaluated on GrQc

| $(w_{out}, l_{out})$ | (200,2) | (100,4) | (50,8) | (40,10) | (25,16) | (16,25) | (10,40) | (8,50) | (4,100) | (2,200) |
|---|---|---|---|---|---|---|---|---|---|---|
| Pre-processing Time | **73.00 s** | 75.65 s | 77.78 s | 80.03 s | 88.09 s | 106.2 s | 151.1 s | 191.4 s | 512.2 s | 1,709 s |
| mAE | 0.7829 | 0.7107 | 0.6811 | 0.6780 | 0.7170 | 0.7129 | 0.6970 | **0.6751** | 0.7340 | 0.6986 |
| mRE | 0.1389 | 0.1225 | 0.1209 | 0.1225 | 0.1274 | **0.1204** | 0.1240 | 0.1212 | 0.1295 | 0.1263 |

$\zeta, \alpha$: **the decay coefficient of BC values and distance weights.** These parameters mainly affect the performance on accuracy by dominating the intrinsic behaviors of BC walk and DR, respectively. Thereinto, $\zeta$ determines how frequently a node could be enrolled in the current BC walk, which helps to diverge the direction of different walks from one root. Likewise, $\alpha$ determines how frequently a node with more hops from the root could be selected into resampled paths, which helps to distinguish neighbors of different orders. Like the previous evaluation, we test BCDR with $\zeta = \{-1, 0, 1, 2, 4, 10, 20\}$ and $\alpha = \{0.1, 0.2, 0.3, 0.4, 0.5, 0.9, 0.98\}$ to show their impacts.

Table 18: Impacts of $\zeta$ and $\alpha$ on the performance of BCDR evaluated on FaceBook

| $\zeta$ | -1 | 0 | 1 | 2 | 4 | 10 | 20 |
|---|---|---|---|---|---|---|---|
| mAE | 0.0522 | 0.0146 | **0.0061** | 0.0243 | 0.0137 | 0.0143 | 0.0131 |
| mRE | 0.0186 | 0.0069 | **0.0026** | 0.0099 | 0.0053 | 0.0056 | 0.0052 |

| $\alpha$ | 0.1 | 0.2 | 0.3 | 0.4 | 0.5 | 0.9 | 0.98 |
|---|---|---|---|---|---|---|---|
| mAE | 0.0104 | 0.0418 | 0.0506 | **0.0096** | 0.0252 | 0.0197 | 0.0341 |
| mRE | **0.0046** | 0.0204 | 0.0178 | **0.0046** | 0.0125 | 0.0096 | 0.0159 |

Table 19: Impacts of $\zeta$ and $\alpha$ on the performance of BCDR evaluated on GrQc

| $\zeta$ | -1 | 0 | 1 | 2 | 4 | 10 | 20 |
|---|---|---|---|---|---|---|---|
| mAE | **0.6419** | 0.6865 | 0.6844 | 0.6717 | 0.7219 | 0.6734 | 0.6879 |
| mRE | **0.1146** | 0.1234 | 0.1209 | 0.1214 | 0.1235 | 0.1175 | 0.1209 |

| $\alpha$ | 0.1 | 0.2 | 0.3 | 0.4 | 0.5 | 0.9 | 0.98 |
|---|---|---|---|---|---|---|---|
| mAE | 0.7216 | **0.6780** | 0.7036 | 0.7441 | 0.7006 | 0.7281 | 0.7258 |
| mRE | 0.1237 | **0.1202** | 0.1239 | 0.1295 | 0.1234 | 0.1267 | 0.1290 |

From the Table 18 and 19, we see the accuracy of BCDR is not sensitive to these parameters, but a fine-tuning process could improve the performance on specific graphs.

For choices of $\zeta$, it depends on the fluctuation of centrality on neighbor nodes. Specifically, for relatively dense graphs (like Facebook) with flattened centrality on neighbors, a larger $\zeta$ resists the frequency decaying of most preferred walk paths, leading to efficient exploration for high-order distance relations. On the contrary, a quick BC decaying (smaller $\zeta$) makes the priority of neighbor nodes indistinguishable, dragging down the performance like a naive random walk, since many neighbors possess similar centrality on such graphs.

For choices of $\alpha$, as discussed in Remark 2, it reflects a trade-off between quality (i.e., preserves accurate distance relations) and quantity (i.e., embeds more relations with a widened range of nodes). In detail, a smaller $\alpha$ slows down the process $\hat{D}_{ab} \to 0$, allowing relations between node pairs with larger distance $D_{ab}$ to converge, i.e., $\mathbf{Z}_a \mathbf{Z}_b^T \to \hat{D}_{ab} > 0$, but it causes nodes possessing similar distance from the root indistinguishable due to the noise in the embedding space, and vice versa.

**Number of epochs.** The number of epochs determines if it is sufficient to learn a NN distance predictor. To produce the results of Table 2, we just leverage the empirical value as discussed in (Rizi et al., 2018). Here, we evaluate its impact on accuracy loss and pre-processing time.

Table 20: Impacts of the number of epochs on the performance of BCDR evaluated on FaceBook

| Num. of epochs | 1 | 2 | 5 | 10 | 15 | 20 | 40 |
|---|---|---|---|---|---|---|---|
| Pre-processing Time | **121.2 s** | 125.3 s | 130.7 s | 139.7 s | 150.1 s | 160.0 s | 194.7 s |
| mAE | 0.0174 | 0.0136 | 0.0167 | 0.0121 | **0.0107** | 0.0176 | 0.0259 |
| mRE | 0.0087 | 0.0067 | 0.0083 | 0.0057 | **0.0048** | 0.0070 | 0.0104 |

Table 21: Impacts of the number of epochs on the performance of BCDR evaluated on GrQc

| Num. of epochs | 1 | 2 | 5 | 10 | 15 | 20 | 40 |
|---|---|---|---|---|---|---|---|
| Pre-processing Time | **44.67 s** | 47.92 s | 54.69 s | 68.03 s | 79.85 s | 92.09 s | 141.5 s |
| mAE | 0.6888 | 0.6913 | 0.7071 | **0.6786** | 0.6803 | 0.6884 | 0.7047 |
| mRE | 0.1225 | 0.1257 | 0.1263 | **0.1158** | 0.1197 | 0.1194 | 0.1225 |

The results in Table 20 and 21 show that learning with 15 epochs is generally appropriate for many real-world graphs. It also reflects that training the distance predictor with more iterations may cause an over-fitting problem since the training data (distance triplets) are extracted from a few landmarks, which induces harmful inductive bias on a certain part of the graph.

## A.12 EXTENDED RESULTS ON EXPLORATION RANGE OF DISTANCE

The extended results on all synthetic graphs are shown in Figure 7. We analyze the significance of utilizing BC-RW for a wider range of exploration on different structures as follows.

- For CGs and TCGs, BC-RW tends to choose the exit nodes of each circle since they provide a large BC gain by splitting all SPs between inner nodes and outer nodes regarding the current circle.

- For TGs, transitions on every triangle clique tend to move forward along the trunk road since the number of nodes beyond the current clique is often larger than that of inner nodes, contributing to more SPs.

- For TRGs, each transition from the root to leaves appears to be biased since subtrees with more descendants contribute to more SPs and possess larger BC values.

- For SGs, there are many shortcuts that link some nodes on the trunk road, and the BC values of shortcut nodes and other nodes are usually on par. BC-RW possesses a slight advantage by keeping a relatively good balance on these nodes.

- For NGs, most of the nodes are passed through by SPs with similar probabilities, and BC-RW is hard to tell the proper direction for deeper exploration like other walk strategies.

## A.13 EXTENDED RESULTS ON PRESERVATION OF DISTANCE RELATIONS

The extended results of distance preservation are shown in Figure 8 and 9. These figures confirm that our model is much more satisfactory in preserving distance relation than existing methods except for TGs. For most graphs, BC walk paths provide sufficient observation on each node by locating many remote nodes with a sequence of center nodes on a graph, and thus the resampling process based on such observation could preserve distance relations in the exploration range. For TGs, however, there are many final nodes (i.e., leaves) possessing trivial significance on BC walks which are insufficiently observed for calibrating their distance relations well.

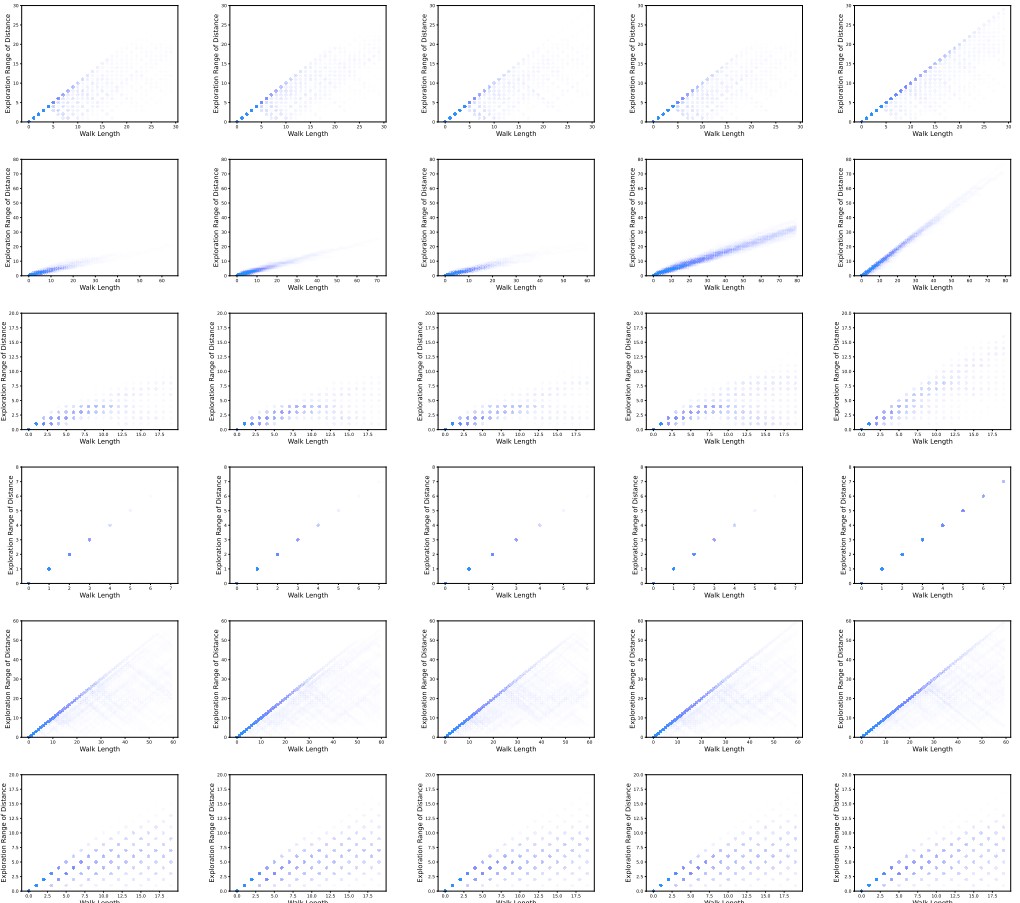

Figure 7: Exploration range of distance when taking different walk strategies tested on six synthetic graphs. **Row from top to bottom:** different synthetic graphs including CG, TG, TCG, TRG, SG, and NG, respectively. **Column from left to right:** different walk strategies including NRW, SORW, RS, DFS-RW, BC-RW (ours.), respectively.

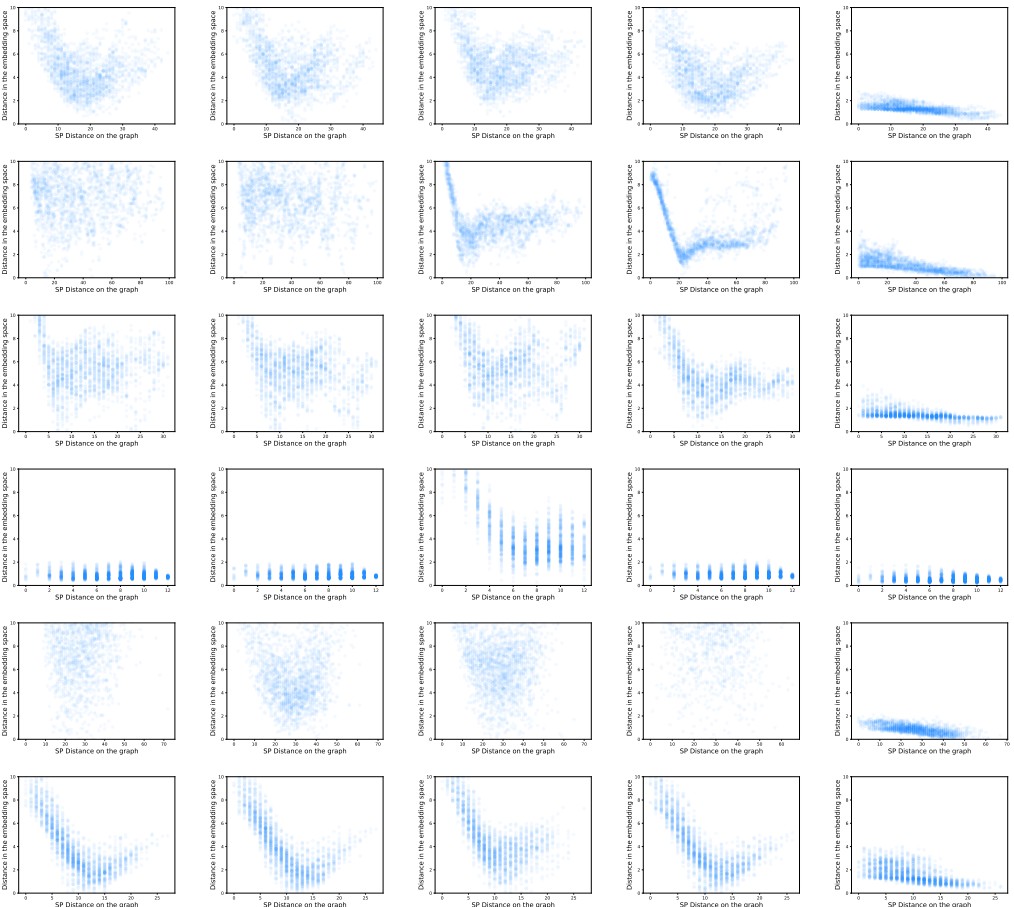

Figure 8: Measured distance from the embedding space and the original graph. **Row from top to bottom:** different graphs including CG, TG, TCG, TRG, SG, and NG, respectively. **Column from left to right:** embeddings learned by different walk strategies,i.e., NRW, SORW, RS, DFS-RW, and BC-RW (ours.), respectively. For ours, walk paths are further simulated by *distance resampling*.

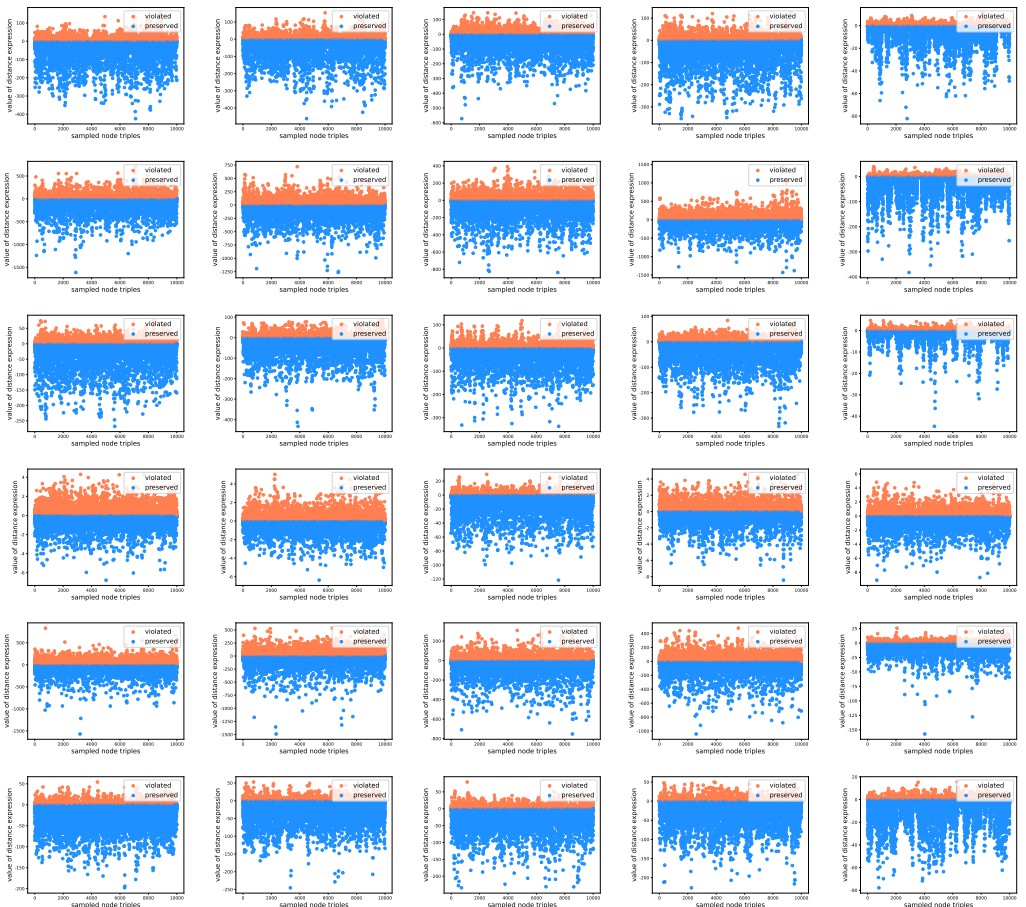

Figure 9: Distance preservation among node triplets in the embedding spaces. **Row from top to bottom:** different graphs including CG, TG, TCG, TRG, SG, and NG, respectively. **Column from left to right:** embeddings learned by different walk strategies,i.e., NRW, SORW, RS, DFS-RW, and BC-RW (ours.), respectively. For ours, walk paths are further simulated by *distance resampling*.

