# OpenReview forum: "Learn Low-dimensional Shortest-path Representation of Large-scale and Complex Graphs"
_ICLR.cc/2023/Conference — Submitted to ICLR 2023_

### Official Review · Reviewer_87wC · 2022-10-21

**Confidence:** 2
**Correctness:** 2
**Technical Novelty And Significance:** 2
**Empirical Novelty And Significance:** 3
**Recommendation:** 5

**Clarity, Quality, Novelty And Reproducibility:**

My main critique with the current work is that I don't understand the necessity of first sampling random walks based on sampled betweenness-centrality and then extracting shortest paths from the random walks. If shortest paths are sampled anyways to estimate BC, why can't these paths directly be used as contexts for the training of the model.
Moreover, why not use shortest paths of certain max length as training data?
The authors should either argue more clearly on the necessity of the approach or empirically show the drawbacks of the 'natural' approach.


Minor Issues:
- p1: runtime of $k$ sp-computations on an unweighted graph should be $O(kN)$, not $O(kN^2)$.
- p2: Table 1 reports asymptotic runtime and space requirements. What are the last three colums doing there? To which graph are you referring and how does this relate to the asymptotic results?
- p5: what is a 'random SP walk'?

**Strength And Weaknesses:**

Strengths:
- the empirical results look promising

Weaknesses
- it is unclear how the training paths ('random sp walk') are constructed from the random walks
- it remains unclear how the present work relates to path2vec both conceptually and practically



**Summary Of The Paper:**

The authors propose a method to compute embeddings suitable for fast and exact prediction of shortest path distances in a large graph.

**Summary Of The Review:**

I recommend to reject the paper at the current point in time.

---

> ### Author Response · Authors · 2022-11-13
> **Author Response to Reviewer 87wC (1/3)**
>
> We sincerely thank Reviewer 87wC for in-depth feedback and interesting questions on the proposed method, and we would like to carefully address the concerns as follows.
>
> ***
>
> ### For the main critique
>
> > **Comment 1** I don't understand the necessity of ﬁrst sampling random walks based on sampled betweenness centrality and then extracting shortest paths from the random walks.
> >
> > If shortest paths are sampled anyways to estimate BC, why can't these paths directly be used as contexts for the training of the model. Moreover, why not use shortest paths of certain max length as training data?
>
> **Response 1** Initially, we further clarify the motivation for utilizing BCDR instead of directly sampling shortest paths. As a prerequisite, it should be acknowledged that we need sampled shortest paths as *observation* to optimize node embeddings $Z$. However, to perform a sufficient *observation* on all pairs of shortest paths is time-consuming, which takes at least $O(N^2)$ time on sparse unweighted graphs. Towards this, an intuitive idea is to sample a limited number of shortest paths that starts only at a few nodes (landmarks). But it will introduce strong bias on the landmarks and ignore many shortest paths far away from them. To alleviate this bias, in BCDR, we hope to observe shortest paths rooted at all nodes on the graph (instead of the landmarks only). Therefore, we need some strategies to overcome the huge complexity of directly sampling these paths (since it requires performing BFS on all nodes). The proposed strategy is *BC-based random walk* where we intend to equip 'random walk' with the awareness of high-order SP structure and make the sampled walk paths much more likely to be certain shortest paths. This strategy is comparatively efficient since the sampling complexity is proportional to its path length $l$. Then the subsequent module DR further resampled from these paths for implicitly preserving SP distance relations on $Z$.
>
> According to the above discussion, a brief procedure of BCDR with its motivation could be summarized as follows.
>
> 1. estimate BC just by BFS from only a few nodes (landmarks)
> - **motivation**: determine which node is prone to trigger high-order explorations of SP distances.
> 2. perform *BC-based random walk*
>   - **motivation**: observe the potential shortest paths rooted at each node sufficiently.
> 3. leverage DR for resampling approximate random shortest paths
>   - **motivation**: implicitly preserving distance relations on observed paths.
> 4. optimize $Z$ from the observation of the resampled paths
>   - **motivation**: reflect the SP structure on the graph instead of arbitrary linkage.
>
> Each step above possesses linear complexity with respect to $N$ (number of nodes in the graph).
>
> Besides, we are convinced of the necessity of BCDR procedure and would like to explain it carefully from **both technical and empirical perspectives**.
>
> **From a technical perspective**, directly leveraging shortest paths as observation to optimize $Z$ has a few shortcomings.
>
> 1. Prohibitive Complexity of Sufficient Observation.
>    - Observing all pairs of SP distance requires at least $O(N^2)$ time for sparse unweighted graphs.
>    - Alternatively, an insufficient observation with linear complexity will cause a loss in accuracy (see experimental results below)
> 2. Inflexible Path Length for Optimization.
>    - Since we leverage the skip-gram algorithm for optimizing $Z$, it should be clear how long the sliding window size is, serving to reconstruct the distance relations between nodes. But shortest paths rooted at a certain node factually possess significantly divergent path lengths, which causes difficulty in determining proper sliding window size on different paths, i.e., a longer window helps to capture long-distance correlation but causes indistinguishable in shorter paths and vice versa.
>    - Alternatively, if we only select shortest paths of a certain fixed length, paths shorter than this length will be ignored, thus impairing the performance.
>
> Correspondingly, the BCDR procedure overcomes the above shortcomings as follows.
>
> 1. Linear Complexity of Such Observation.
>    - Instead of directly simulating shortest paths, we sample paths by *BC-based random walk* and transform the paths into approximate random shortest paths by DR. Both of these operation share linear time complexity. Also, the optimization on such resampled paths is proved to share similar properties with that on real shortest paths by Prop. 1 and Thm. 1.
> 2. Flexible Path Length for Optimization.
>    - Since the paths are resampled from random paths, the number and length of them (i.e., $w_{out},l_{out}$) could be customized. We are thus able to fix them at a certain proper length for subsequent optimization.

---

> > ### Author Response · Authors · 2022-11-13
> > **Author Response to Reviewer 87wC (2/3)**
> >
> > **From an empirical perspective**, we further construct and evaluate 6 competitive baselines which have the same architecture and hyper-parameters as BCDR, but adopt different intuitive strategies to directly optimize on shortest paths.
> >
> >  The basic description of these baselines is stated as follows.
> >
> > 1. Shortest Paths on Landmarks only (SPoL)
> >    - Since we need anyway perform BFS from landmarks to acquire distance triplet for learning distance predictor, we intuitively retrieve the shortest paths starting from the landmarks. This operation introduces little extra time cost.
> >    - The size of landmark set is the same as BCDR (i.e., $|L|=80$)
> > 2. Shortest Paths on Landmarks only with Fixed Length (SPoL-F)
> >    - This is similar to SPoL but restricts the output walk length at a certain level (the same as BCDR, i.e., $l_{out}=10$)
> > 3. Shortest Paths on All Nodes (SPoN)
> >    - In BCDR, we perform BC random walk on each node $v_a$ to locate its position on the graph. Here, we directly sample shortest paths from $v_a$ to any other nodes instead.
> >    - Specifically, for each source node $v_a$, we take a uniform sampling over $V$ to acquire the destination nodes, and retrieve the shortest paths between them. The number and max length of shortest paths on each node is the same as BCDR (i.e., $w_{out}=40,l_{out}=10$)
> > 4. Shortest Paths on All Nodes with Fixed Length (SPoN-F)
> >    - This is similar to SPoN but restricts the output walk length (the same as BCDR, i.e., $l_{out}=10$)
> > 5. Shortest Paths on Arbitrary Node Pair (SPoANP)
> >    - We randomly select a group of node pairs $(v_s, v_t)$ and retrieve one of the shortest paths between them by BFS. The number of paths is the same as the total number of walk paths on all nodes in BCDR (i.e., $N \times w_{out}$)
> > 6. Shortest Paths on Arbitrary Node Pair (SPoANP-F)
> >    - This is similar to SPoANP but restricts the output walk length (the same as BCDR, i.e., $l_{out}=10$)
> >
> > All of the above baselines are evaluated on GrQc dataset, and the experimental results are presented as follows.
> >
> > |  Model   |   PT   |   ST   |  mAE   |  mRE   |
> > | :------: | :----: | :----: | :----: | :----: |
> > |   SPoL   | 60.71s | 15.36s | 0.9703 | 0.1641 |
> > |  SPoL-F  | 60.46s | 15.63s | 1.2874 | 0.1961 |
> > |   SPoN   | 283.0s | 235.5s | 1.0411 | 0.1645 |
> > |  SPoN-F  | 2,702s | 2,656s | 1.2961 | 0.1969 |
> > |  SPoANP  | 282.1s | 234.4s | 1.3564 | 0.2047 |
> > | SPoANP-F | 6,341s | 6,290s | 1.3294 | 0.2003 |
> > |   BCDR   | 69.61s | 27.66s | 0.7043 | 0.1274 |
> >
> > PT: processing time, ST: time for sampling paths, mAE: mean of Absolute Error, mRE: mean of Relative Error
> >
> > We see from the table that BCDR outperforms all the baselines in approximation quality (i.e., mAE and mRE) with proper processing time. Specifically, SPoL possesses desirable processing time since only the shortest paths rooted at landmarks are considered. But they are plagued with insufficient observation of other shortest paths that do not pass through landmarks. SPoN and SPoANP suffer huge complexity when retrieving shortest paths on the whole graph, and perform even worse due to the uncertainty of reasonable sliding window size. From SPoL-F, SPoN-F, and SPoANP, we see that even if the path length is fixed, some uncaptured shorter paths will also cause a loss in accuracy.

---

> > > ### Author Response · Authors · 2022-11-13
> > > **Author Response to Reviewer 87wC (3/3)**
> > >
> > > ### For other concerns,
> > >
> > > > **Comment 2** runtime of sp-computations on an unweighted graph should be $O(kN)$, not $O(kN^2)$.
> > >
> > > **Response 2** For unweighted graphs, the time complexity of computing shortest paths between any node pair is bounded by performing BFS on the graph, i.e., $O(N+M)$, where $N$ and $M$ are the number of nodes and edges, respectively. Since we assume in the sentence that the graph is dense, where $M\approx N^2$, thus we can say the time complexity is $O(kN^2)$ for $k$ queries.
> > >
> > > Despite the correctness of this assertion, we consider it necessary to change its expression to '$O(kN)$ for sparse unweighted graphs'. As the graphs addressed in the rest of this paper are relatively sparse, discussing an upper bound for dense graphs here is somewhat inappropriate. We thanks the Reviewer for pointing out this issue.
> > >
> > > ***
> > >
> > > > **Comment 3** Table 1 reports asymptotic runtime and space requirements. What are the last three colums doing there? To which graph are you referring and how does this relate to the asymptotic results?
> > >
> > > **Response 3** As already stated in the caption of Table 1, we aim to show the comparison of each approach when evaluated on the DBLP graph dataset (see details in A.7.4). Columns 3-5 show the asymptotic complexity on such an unweighted graph, and Column 6-8 show the actual performance on answering SP distance queries.
> > >
> > > ***
> > >
> > > > **Comment 4** what is a 'random SP walk'?
> > >
> > > **Response 4** For a specific node $v_a$, we refer to 'random SP walk rooted at $v_a$' as a walking pattern whose transition reflects the probability of each shortest path passing through $v_a$. Specifically, each transition tends to select the nodes that are likely to be a hub on most shortest paths from $v_a$ to others. It also means that paths sampled from *random SP walk* are prone to be a shortest path rooted at $v_a$. Notably, *random SP walk* is different from *BC-based random walk* since the latter also considers other shortest paths passing through the current node (may avoid $v_a$). As mentioned above, we proposed DR to retrieve proper resampled paths from BC walk paths, which share similar properties with *random sp walk*.
> > >
> > > ***
> > >
> > > > **Comment 5** it remains unclear how the present work relates to path2vec both conceptually and practically
> > >
> > > **Response 5** Path2Vec aims to learn general node embedding, which reflects user-defined pair-wise distance measures. In terms of shortest-path representation, we would like to discuss some limitations of Path2Vec in contrast with BCDR.
> > >
> > > **Conceptually**, Path2Vec ignores the local structure of each node and directly optimizes on the mutual distance of node pair, even though both Path2Vec and BCDR leverage skip-gram algorithm to optimize node embeddings on distance relations. In fact, Path2Vec takes a simple strategy to jointly maximize the co-occurrence likelihood between the first-order neighbors of the source and destination. However,  the resulting behaviors of the nodes in these neighborhoods remain unclear and totally unconstrained. In BCDR, we optimize the relation between the current node and its observed high-order neighborhoods (computed by BC walk), and leverage DR to restrict the distance relations among these neighborhoods, ensuring the resulting embeddings (**not only of source and destination but also their high-order neighbors**) for preserving desirable properties on SP distance.
> > >
> > > **Practically**, we have shown the performance of Path2Vec in Table 2. The result reveals that Path2Vec takes a long time to finis since most of the nodes have a large number of first-order neighbors on test graphs. The accuracy of Path2Vec is also plagued by its loss in local structural observation.
> > >
> > >
> > > Most of the above discussion and analysis are already updated in our latest manuscript, and we would appreciate it if these revisions and supplementary material are considered to be informative for the Reviewer's assessment.

---

> ### Author Response · Authors · 2022-12-10
> **Thank you! Looking forward to your feedback.**
>
> Dear Reviewer 87wC,
>
> Thanks again for your constructive and insightful review of our work! We have carefully addressed all your concerns and revised our manuscript from many aspects in the past weeks (see **General Response**). As Discussion Phase 2 is **ending in two days**, we would like to know if our responses have cleared up your questions and if any additional issues could be further clarified. We value and expect the opportunity to discuss this paper with each reviewer and are willing to make every effort to meticulously improve our work.
>
> Many thanks for your precious time.
>
>
> Best wishes,
>
> Anonymous Authors.

---

### Official Review · Reviewer_VC9e · 2022-10-24

**Confidence:** 3
**Correctness:** 3
**Technical Novelty And Significance:** 3
**Empirical Novelty And Significance:** Not applicable
**Recommendation:** 6

**Clarity, Quality, Novelty And Reproducibility:**

The paper is in general well written and clear in its presentation. The central idea using betweenness centrality seems rather straightforward and not particularly novel, however, the approach has merits as demonstrated in the experimentation when compared to existing procedures.  The reproducibility of the results are somewhat unclear relying on tuning of \alpha and the dimensionality of Z which is not discussed and carefully investigated in terms of impact. This hampers reproducibility as these parameter-settings seem critical whereas additional parameter tuning such as path lengths etc. also need to be set. The manuscript thus needs to much more carefully discuss the settings and impacts of the multiple critical parameters needed for the BCRD as also highlighted in Algorithm 1.

**Strength And Weaknesses:**

Strengths:
The proposed BCDR provides good performance that performs well contrasted alternative procedures quantifying SP distances.
The approach utilizing betweenness centrality is sound and the mathematical derivations and procedure proposed carefully designed.
The embedding procedure used to quantify Z is efficient and performs well evaluating all shortest paths approximately

Weaknesses:
The approach appears rather straightforward and the importance quantifying accurately SP should be better motivated. Clearly, SP have important applications, however, it would be interesting to highlight the utility quantifying SP accurately in downstream tasks as SP is not a particularly good predictor for several graph representation learning tasks such as link-prediction, node classification, community detection etc in comparison to state-of-the-art GRL approaches. It would therefore strengthen the papers to further motivate the importance of explicitly characterizing accurately SP in terms of important downstream tasks relying on such accurate quantification.

The approach relies on the tuning of a weight decay term \alpha. It should be discussed what the impact of this parameter is and its tuning experimentally. The approach further relies on the embedding dimensionality of Z this is not discussed nor is the impact of dimensionality accessed, this needs to be clarified. It is very unclear how these parameters should be set and their impact on the results. How were they tuned for the presented results in Table 2?


**Summary Of The Paper:**

The authors propose an efficient procedure for evaluating and characterizing shortest path distances by exploring a random walk procedure biased by betweenness centrality thereby better exploring important connections between communities in the random walk process used to access shortest paths (SP). Using this biased sampling strategy they infer a graph embedding Z based on the procedure based on the negative sampling approach of Milokov et al. augmented to account for betweenness centrality and they find that their procedure more accurately establish the SP distances when compared to a range of current approaches quantifying SP considering six networks with favorable processing usage.

**Summary Of The Review:**

A well written paper based on a simple idea of exploring betweenness centrality when quantifying SP using random walks. The central idea using betweenness centrality is not particularly innovative but the approach has merits as demonstrated in the experimentation when compared to existing procedures.

Whereas the results are compelling demonstrating enhanced recovery of SP distances in graphs at favorable computational demands, the importance of such quantification in downstream tasks in which graph representation learning approaches (GRL) are typically considered state-of-the-art is unclear.

The BCDR relies on the setting of multiple hyperparameters but lacks careful investigation and justification of the settings of these parameters.

---

> ### Author Response · Authors · 2022-11-13
> **Author Response to Reviewer VC9e (1/8)**
>
> We sincerely thank Reviewer VC9e for valuable comments and suggestions to improve this manuscript. We would like to carefully address the concerns as follows.
>
> > **Comment 1** The BCDR relies on the setting of multiple hyperparameters but lacks careful investigation and justiﬁcation of the settings of these parameters.
> >
> > The manuscript thus needs to much more carefully discuss the settings and impacts of the multiple critical parameters needed for the BCRD as also highlighted in Algorithm 1.
> >
> > It is very unclear how these parameters should be set and their impact on the results. How were they tuned for the presented results in Table 2?
>
> **Response 1** To address this issue, we further investigate the parameter settings of the proposed method and discuss several critical parameters for their impacts on performance in the updated manuscript.
>
> Notably, although we describe rather detailed settings of parameters in Appendix A.9 and Alg. 1, the proposed method BCDR is factually robust and effective, and its performance does not sensitively rely on any one of them. For reproduction of the presented results in Tab. 2, we have presented in Appendix A.8 a group of proper parameters, also summarized as follows.
>
> |                                         | for small graphs (≤ 10k nodes) | for large graphs (over 100k nodes) |
> | :-------------------------------------: | :----------------------------: | :--------------------------------: |
> |     **dimension of embeddings $d$**     |               16               |                 16                 |
> |       **num. of landmarks $\|L\|$**       |               80               |                 5                  |
> |      **num. of BC walk $w_{in}$**       |               20               |                 2                  |
> |     **length of BC walk $l_{in}$**      |               40               |                 40                 |
> |  **num. of resampled walk $w_{out}$**   |               40               |                 40                 |
> | **length of resampled walk $l_{out}$**  |               10               |                 10                 |
> |    **BC decay coefficient $\zeta$**     |               10               |                 1                  |
> | **distance decay coefficient $\alpha$** |              0.35              |                0.35                |
> |           **num. of epochs**            |               15               |                 15                 |
>
> Then, we would like to show the impacts of these parameters on related metrics and how to easily tune them in any unweighted graphs, both conceptually and practically. The next discussion and evaluation of each parameter follow its order in the above table.
>
> **1. $d$: the dimension of node-level embeddings (i.e., $Z$)**
>
> In our experiment, $d$ is not a fine-tuned parameter but fixed at a certain value (i.e., $d = 16$) among different models to fairly evaluate their performance. This parameter could improve the performance on accuracy since a large size of embeddings could dump more valuable information about SP structures at the expense of higher storage cost and deficiency in query speed. To verify this, We test BCDR with different $d = ${$2,4,16,64,128,256$} on Facebook and GrQc to evaluate their performance under these metrics.
>
> - On Facebook,
>
> |      $d$       |       2       |     4     |    16    |    64     |    128    |    256     |
> | :------------: | :-----------: | :-------: | :------: | :-------: | :-------: | :--------: |
> |  **Storage**   | **0.1307 MB** | 0.1924 MB | 1.210 MB | 3.160 MB  | 5.944 MB  |  8.579 MB  |
> |    **mAE**     |    0.0902     |  0.0150   |  0.0202  |  0.0499   |  0.0310   | **0.0130** |
> |    **mRE**     |    0.0347     |  0.0075   |  0.0091  |  0.0212   |  0.0122   | **0.0064** |
> | **Query Time** | **4,089 ns**  | 4, 664 ns | 8,334 ns | 22,430 ns | 41,142 ns | 81,188 ns  |
>
> - On GrQc,
>
> |      $d$       |       2       |     4     |    16     |    64     |    128     |    256    |
> | :------------: | :-----------: | :-------: | :-------: | :-------: | :--------: | :-------: |
> |  **Storage**   | **0.2240 MB** | 0.3023 MB | 0.7916 MB | 2.750 MB  |  5.415 MB  | 10.86 MB  |
> |    **mAE**     |    0.8719     |  0.8954   |  0.7089   |  0.6867   | **0.6746** |  0.6770   |
> |    **mRE**     |    0.1548     |  0.1555   |  0.1259   |  0.1227   | **0.1201** |  0.1209   |
> | **Query Time** | **2,421 ns**  | 3,011 ns  | 6, 570 ns | 20,837 ns | 39, 479 ns | 79,023 ns |
>
> From the table, we see that the accuracy loss could be cut down by increasing $d$, but it will lead to significant deterioration in storage cost and query speed. As we discuss a low-dimensional and accurate SP representation in this paper, the results also reveal that even at a rather lower dimension of embeddings (like $d=4$), the distance relations on the graph could be well-preserved.

---

> > ### Author Response · Authors · 2022-11-13
> > **Author Response to Reviewer VC9e (2/8)**
> >
> > **2. $|L|$: the number of landmarks for constructing distance triplet and estimating BC.**
> >
> > This parameter mainly affects accuracy and pre-processing time since involving more landmarks helps to alleviate harmful inductive bias on a certain part of the graph but suffers higher computing overhead. It is also observed in the previous works [1] that for large graphs with strong centrality on a few nodes, the number of landmarks could be reduced without much loss of accuracy. We evaluate BCDR with a group of landmarks ($\|L\| = ${$10,20,40,80,160$}) on Facebook and GrQc, to see their impacts on the two metrics.
> >
> >
> > - On Facebook,
> >
> > |          $\|L\|$          |     10      |   20    |   40    |   80    |    160     |
> > | :---------------------: | :---------: | :-----: | :-----: | :-----: | :--------: |
> > | **Pre-processing Time** | **127.8 s** | 134.3 s | 142.5 s | 157.5 s |  187.5 s   |
> > |         **mAE**         |   0.0342    | 0.0297  | 0.0148  | 0.0193  | **0.0124** |
> > |         **mRE**         |   0.0134    | 0.0108  | 0.0063  | 0.0096  | **0.0062** |
> >
> > - On GrQc,
> >
> > |          $\|L\|$          |     10      |     20     |   40    |   80    |   160   |
> > | :---------------------: | :---------: | :--------: | :-----: | :-----: | :-----: |
> > | **Pre-processing Time** | **47.47 s** |  52.95 s   | 64.35 s | 83.75 s | 123.7 s |
> > |         **mAE**         |   0.9922    | **0.6837** | 0.7383  | 0.7112  | 0.7065  |
> > |         **mRE**         |   0.1591    | **0.1185** | 0.1266  | 0.1217  | 0.1231  |
> >
> > The results show that the pre-processing time on graphs increases linearly with $|L|$ since performing BFS from the added landmarks needs extra traversal on the whole graph for $O(N+M)$ time. It is also interesting to see that the number of landmarks large enough for the best performance diverges for dense and sparse graphs, i.e., it generally takes more than $40$ landmarks for Facebook but only $20$ landmarks necessary for GrQc. Specifically, for relatively dense graphs (i.e., Facebook), each node shares weaker centrality due to the enriched links, which means we need to observe more landmarks to cover more SPs on the graphs (according to the hub-labeling theory in [2]). But for sparse graphs (i.e., GrQc), as long as several nodes with strong centrality are well-observed, SP distance between most node pairs could be preserved, resulting in tolerance of reduced landmarks.

---

> > > ### Author Response · Authors · 2022-11-13
> > > **Author Response to Reviewer VC9e (3/8)**
> > >
> > > **3. $w_{in}, l_{in}$: the number and length of sampled BC walks on each node**
> > > These parameters affect the accuracy and pre-processing time. When we simluate BC walks rooted at a certain node, a large $w_{in}$ makes it sufficient to observe the local structure of each node (like BFS), while a large $l_{in}$ allows wider exploration on the graph to let the distance with remote nodes be seen (like DFS). Like the previous evaluation, we test $w_{in}=${$5,10,20,30,40$} and $l_{in}=${$5,10,20,40,60,80$} to investigate their impacts, respectively.
> > >
> > > - On Facebook,
> > >
> > > |        $w_{in}$         |      5      |   10    |   20    |   30    |     40     |
> > > | :---------------------: | :---------: | :-----: | :-----: | :-----: | :--------: |
> > > | **Pre-processing Time** | **128.5 s** | 143.7 s | 186.3 s | 228.2 s |  273.5 s   |
> > > |         **mAE**         |   0.0136    | 0.0454  | 0.0113  | 0.0188  | **0.0062** |
> > > |         **mRE**         |   0.0061    | 0.0182  | 0.0056  | 0.0093  | **0.0027** |
> > >
> > > |        $l_{in}$         |      5      |   10    |   20    |     40     |   60    |   80    |
> > > | :---------------------: | :---------: | :-----: | :-----: | :--------: | :-----: | :-----: |
> > > | **Pre-processing Time** | **126.9 s** | 131.6 s | 145.9 s |  182.8 s   | 215.4 s | 114.0 s |
> > > |         **mAE**         |   0.0133    | 0.0145  | 0.0725  | **0.0081** | 0.0320  | 0.0370  |
> > > |         **mRE**         |   0.0065    | 0.0067  | 0.0304  | **0.0037** | 0.0132  | 0.0184  |
> > >
> > >
> > > - On GrQc,
> > >
> > > |        $w_{in}$         |      5      |   10    |     20     |   30    |   40    |
> > > | :---------------------: | :---------: | :-----: | :--------: | :-----: | :-----: |
> > > | **Pre-processing Time** | **112.2 s** | 114.5 s |  118.8 s   | 125.1 s | 130.2 s |
> > > |         **mAE**         |   0.7227    | 0.6671  | **0.6581** | 0.6930  | 0.7115  |
> > > |         **mRE**         |   0.1250    | 0.1195  | **0.1166** | 0.1245  | 0.1259  |
> > >
> > > |        $l_{in}$         |      5      |     10     |   20    |     40     |   60    |   80    |
> > > | :---------------------: | :---------: | :--------: | :-----: | :--------: | :-----: | :-----: |
> > > | **Pre-processing Time** | **114.0 s** |  115.7 s   | 117.8 s |  119.3 s   | 119.7 s | 120.8 s |
> > > |         **mAE**         |   0.7460    |   0.6765   | 0.7074  | **0.6643** | 0.6895  | 0.6926  |
> > > |         **mRE**         |   0.1280    | **0.1146** | 0.1217  |   0.1171   | 0.1227  | 0.1230  |
> > >
> > >
> > > The experimental results show the accuracy of BCDR is not sensitive to these parameters, owing much to the efficiency of BC walk and well-preserved distance relations by DR. Intuitively, we recommend setting $l_{in}$ proportional to the diameter of the graph, which makes the whole graph observed from any nodes. Also, $w_{in}$ could be reduced when the connectivity on the graph is relatively weak since the local structures are quite simple to explore.

---

> > > > ### Author Response · Authors · 2022-11-13
> > > > **Author Response to Reviewer VC9e (4/8)**
> > > >
> > > > **4. $w_{out},l_{out}$: the number and length of resampled paths (by DR) on each node**
> > > >
> > > > These parameters control the shape of output node sequences to subsequently optimize $Z$ under a skip-gram procedure. To avoid much loss of information and preserve the correlation in BC walks, we intend to keep the scale of outputs similar to that of inputs, i.e., $w_{out}l_{out} = \Omega(w_{in}
> > > >  l_{in})$. To accelerate the optimization process, we could further shorten $l_{out}$ and keep this scale (by correspondingly expanding $w_{out}$). Note that this reshaping operation does not apparently change the locality nor impair the performance since DR resamples nodes from high-order neighborhoods with respect to their distance from the root, thus resulting in well-defined convergence, as shown in Prop. 1.
> > > >
> > > > In the experiment, we fix the scale of output node sequences as half of the scale of BC walks (i.e., $w_{out}l_{out} = w_{in}l_{in}/2 = 400$), and test different combinations of their settings as $(w_{out},l_{in})=${$(200,2),(100,4),(50,8),(40,10),(25,16),(16,25),(10,40),(8,50),(4,100),(2,200)$}.
> > > >
> > > > - On Facebook,
> > > >
> > > > |   $(w_{out},l_{out})$   | (200,2) | (100,4) | (50,8)  | (40,10) | (25,16) | (16,25) | (10,40) | (8,50)  |  (4,100)   | (2,200) |
> > > > | :---------------------: | :-----: | :-----: | :-----: | :-----: | :-----: | :-----: | :-----: | :-----: | :--------: | :-----: |
> > > > | **Pre-processing Time** | 176.3 s | 154.1 s | 148.6 s | 150.6 s | 156.8 s | 170.5 s | 206.3 s | 236.8 s |  501.0 s   | 1,521 s |
> > > > |         **mAE**         | 0.0237  | 0.0217  | 0.0249  | 0.0258  | 0.0303  | 0.0128  | 0.0127  | 0.0360  | **0.0083** | 0.0237  |
> > > > |         **mRE**         | 0.0097  | 0.0093  | 0.0107  | 0.0108  | 0.0136  | 0.0059  | 0.0043  | 0.0145  | **0.0041** | 0.0098  |
> > > >
> > > > - On GrQc,
> > > >
> > > > |   $(w_{out},l_{out})$   | (200,2) | (100,4) | (50,8)  | (40,10) | (25,16) |  (16,25)   | (10,40) |   (8,50)   | (4,100) | (2,200) |
> > > > | :---------------------: | :-----: | :-----: | :-----: | :-----: | :-----: | :--------: | :-----: | :--------: | :-----: | :-----: |
> > > > | **Pre-processing Time** | 73.00 s | 75.65 s | 77.78 s | 80.03 s | 88.09 s |  106.2 s   | 151.1 s |  191.4 s   | 512.2 s | 1,709 s |
> > > > |         **mAE**         | 0.7829  | 0.7107  | 0.6811  | 0.6780  | 0.7170  |   0.7129   | 0.6970  | **0.6751** | 0.7340  | 0.6986  |
> > > > |         **mRE**         | 0.1389  | 0.1225  | 0.1209  | 0.1225  | 0.1274  | **0.1204** | 0.1240  |   0.1212   | 0.1295  | 0.1263  |
> > > >
> > > > The results reveal that the pre-processing time dramatically increases along with $l_{out}$. This is because we utilize the whole sequence to optimize co-occurrence likelihood between the root and nodes in this sequence, which requires joint training with a large number of node embeddings proportional to $l_{out}$. It is also shown that the accuracy does not significantly fluctuate as pre-processing time, indicating a relatively small $l_{out}$ will help to reduce the off-line time cost.

---

> > > > > ### Author Response · Authors · 2022-11-13
> > > > > **Author Response to Reviewer VC9e (5/8)**
> > > > >
> > > > > **5. $\zeta,\alpha$: the decay coefficient of BC values and distance weights**
> > > > >
> > > > > These parameters mainly affect the performance on accuracy by dominating the intrinsic behaviors of BC walk and DR, respectively. Thereinto, $\zeta$ determines how frequently a node could be enrolled in the current BC walk, which helps to diverge the direction of different walks from one root. Likewise, $\alpha$ determines how frequently a node with more hops from the root could be selected into resampled paths, which helps to distinguish neighbors of different orders. Like the previous evaluation, we test BCDR with $\zeta=${$-1,0,1,2,4,10,20$} and $\alpha=${$0.1,0.2,0.3,0.4,0.5,0.9,0.98$} to show their impacts.
> > > > >
> > > > > - On Facebook,
> > > > >
> > > > > | $\zeta$ |   -1   |   0    |     1      |   2    |   4    |   10   |   20   |
> > > > > | :-----: | :----: | :----: | :--------: | :----: | :----: | :----: | :----: |
> > > > > | **mAE** | 0.0522 | 0.0146 | **0.0061** | 0.0243 | 0.0137 | 0.0143 | 0.0131 |
> > > > > | **mRE** | 0.0186 | 0.0069 |   0.0026   | 0.0099 | 0.0053 | 0.0056 | 0.0052 |
> > > > >
> > > > > | $\alpha$ |    0.1     |  0.2   |  0.3   |    0.4     |  0.5   |  0.9   |  0.98  |
> > > > > | :------: | :--------: | :----: | :----: | :--------: | :----: | :----: | :----: |
> > > > > | **mAE**  |   0.0104   | 0.0418 | 0.0506 | **0.0096** | 0.0252 | 0.0197 | 0.0341 |
> > > > > | **mRE**  | **0.0046** | 0.0204 | 0.0178 | **0.0046** | 0.0125 | 0.0096 | 0.0159 |
> > > > >
> > > > > - On GrQc,
> > > > >
> > > > > | $\zeta$ |     -1     |   0    |   1    |   2    |   4    |   10   |   20   |
> > > > > | :-----: | :--------: | :----: | :----: | :----: | :----: | :----: | :----: |
> > > > > | **mAE** | **0.6419** | 0.6865 | 0.6844 | 0.6717 | 0.7219 | 0.6734 | 0.6879 |
> > > > > | **mRE** | **0.1146** | 0.1234 | 0.1209 | 0.1214 | 0.1235 | 0.1175 | 0.1209 |
> > > > >
> > > > > | $\alpha$ |  0.1   |    0.2     |  0.3   |  0.4   |  0.5   |  0.9   |  0.98  |
> > > > > | :------: | :----: | :--------: | :----: | :----: | :----: | :----: | :----: |
> > > > > | **mAE**  | 0.7216 | **0.6780** | 0.7036 | 0.7441 | 0.7006 | 0.7281 | 0.7258 |
> > > > > | **mRE**  | 0.1237 | **0.1202** | 0.1239 | 0.1295 | 0.1234 | 0.1267 | 0.1290 |
> > > > >
> > > > > From the above tables, we see the accuracy of BCDR is not sensitive to these parameters, but a fine-tuning process could improve the performance on specific graphs.
> > > > >
> > > > > For choices of $\zeta$, it depends on the fluctuation of centrality on neighbor nodes. Specifically, for relatively dense graphs (like Facebook) with flattened centrality on neighbors, a larger $\zeta$ resists the frequency decaying of most preferred walk paths, leading to efficient exploration for high-order distance relations. On the contrary, a quick BC decaying (smaller $\zeta$) makes the priority of neighbor nodes indistinguishable, dragging down the performance like a naive random walk, since many neighbors possess similar centrality on such graphs.
> > > > >
> > > > > For choices of $\alpha$, as discussed in Remark 2, it reflects a trade-off between quality (i.e., preserves accurate distance relations) and quantity (i.e., embeds more relations with a widened range of nodes). In detail, a smaller $\alpha$ slows down the process $\hat D_{ab} \rightarrow 0$, allowing relations between node pairs with larger distance $D_{ab}$ to converge, i.e., $\mathbf Z_a\mathbf Z_b^T \rightarrow \hat D_{ab} > 0$, but it causes nodes possessing similar distance from the root indistinguishable due to the noise in the embedding space, and vice versa.

---

> > > > > > ### Author Response · Authors · 2022-11-13
> > > > > > **Author Response to Reviewer VC9e (6/8)**
> > > > > >
> > > > > > **6. Number of epochs**
> > > > > >
> > > > > > The number of epochs determines if it is sufficient to learn a NN distance predictor. To produce the results of Tab. 2 in the manuscript, we just leverage the empirical value as discussed in [1]. Here, we evaluate its impact on accuracy loss and pre-processing time.
> > > > > >
> > > > > > - On Facebook,
> > > > > >
> > > > > > |     num. of epochs      |      1      |    2    |    5    |   10    |     15     |   20    |   40    |
> > > > > > | :---------------------: | :---------: | :-----: | :-----: | :-----: | :--------: | :-----: | :-----: |
> > > > > > | **Pre-processing Time** | **121.2 s** | 125.3 s | 130.7 s | 139.7 s |  150.1 s   | 160.0 s | 194.7 s |
> > > > > > |         **mAE**         |   0.0174    | 0.0136  | 0.0167  | 0.0121  | **0.0107** | 0.0176  | 0.0259  |
> > > > > > |         **mRE**         |   0.0087    | 0.0067  | 0.0083  | 0.0057  | **0.0048** | 0.0070  | 0.0104  |
> > > > > >
> > > > > > - On GrQc,
> > > > > >
> > > > > > |     num. of epochs      |      1      |    2    |    5    |     10     |   15    |   20    |   40    |
> > > > > > | :---------------------: | :---------: | :-----: | :-----: | :--------: | :-----: | :-----: | :-----: |
> > > > > > | **Pre-processing Time** | **44.67 s** | 47.92 s | 54.69 s |  68.03 s   | 79.85 s | 92.09 s | 141.5 s |
> > > > > > |         **mAE**         |   0.6888    | 0.6913  | 0.7071  | **0.6786** | 0.6803  | 0.6884  | 0.7047  |
> > > > > > |         **mRE**         |   0.1225    | 0.1257  | 0.1263  | **0.1158** | 0.1197  | 0.1194  | 0.1225  |
> > > > > >
> > > > > > The results show that learning with $15$ epochs is generally appropriate for many real-world graphs. It also reflects that training the distance predictor with more iterations may cause an over-fitting problem since the training data (distance triplets) are extracted from a few landmarks, which induces harmful inductive bias on a certain part of the graph.

---

> > > > > > > ### Author Response · Authors · 2022-11-13
> > > > > > > **Author Response to Reviewer VC9e (7/8)**
> > > > > > >
> > > > > > > > **Comment 2** The approach appears rather straightforward and the importance quantifying accurately SP should be better motivated. it would be interesting to highlight the utility quantifying SP accurately in downstream tasks as SP is not a particularly good predictor for several graph representation learning tasks such as link-prediction, node classiﬁcation, community detection etc in comparison to state-of-the-art GRL approaches.
> > > > > > >
> > > > > > > **Response 2** We thank the Reviewer for this constructive suggestion. We would like to clarify the conceptual gaps (or possible misconceptions) between SP representation learning (SPRL) and GRL approaches, and strengthen our motivation for learning to answer accurate SP distance.
> > > > > > >
> > > > > > > It should be initially emphasized that SPRL and GRL are motivated by totally different objectives, which means we cannot expect SPRL to be as versatile as GRL methods in many downstream tasks such as node classification and link prediction, etc.
> > > > > > >
> > > > > > > Specifically, for GRL methods [3, 4], they learn representations reflecting ambiguous similarity measurement among massive nodes. Clearly, by these representations, we could tell whether two nodes are similar or not, but the reason to explain their similarity is agnostic, which limits the direct applications. To better motivate learning such representations, GRL methods should be thus applied to potential downstream tasks to further evaluate their performance.
> > > > > > >
> > > > > > > Furthermore, for SPRL methods [1, 5], they learn to map nodes on the graph into vectors in a metric space where the SP distance among them is well-preserved. Unlike an ambiguous measurement in GRL, the SP measurement among nodes is definite and familiar, thus directly contributing to enormous applications related to it (see detailed discussion below). Since the representations themselves are strongly connected with applications, a further evaluation of potential downstream tasks is not necessary. It could also be seen in the literature of SPRL that the performance of such a model is mainly evaluated on mAE, mRE, and query time, and few of the related work considers evaluation on any other tasks.

---

> > > > > > > > ### Author Response · Authors · 2022-11-13
> > > > > > > > **Author Response to Reviewer VC9e (8/8)**
> > > > > > > >
> > > > > > > > Based on the above discussion, we would like to strengthen our motivation for learning to answer accurate SP distance by investigating its direct relations with several downstream real-world applications in different fields.
> > > > > > > >
> > > > > > > > **Case 1: find nearest points of interest (POI) in road and social networks (most important and intuitive).**
> > > > > > > >
> > > > > > > > Points of interest (POI) [6] are specific point locations that someone may find useful or interesting, e.g., hotels, campsites, fuel stations, etc. A real road network may contain millions of nodes, while thousands of users may issue SP distance queries simultaneously for searching the nearest POI from their location, like 'finding restaurants within 5 km distance' or 'ranking restaurant search results by distance'. To achieve such demands, learning to accurately and fast answer SP distance with limited computing resources is of high significance. Specifically, utilizing limited computing resources means the algorithm should be space- and time-efficient. Thereinto, less storage overhead enables the representations to be stored in users' mobile devices instead of centrally computing SP on the server. And less query time ensures that the computation of SP distance can be processed in real-time (since some POIs may change their positions frequently over time).
> > > > > > > >
> > > > > > > > **Case 2: construct skeleton graph from mesh for 3D animation.**
> > > > > > > >
> > > > > > > > In the literature of 3D animation, animating an articulated character requires constructing a skeleton graph to control the movement of the surface, i.e., place the skeleton joints inside the character and specify which parts of the surface are attached to which bone. A critical technique [7, 8] to automatically embed a skeleton into a character relies on computing a harmonic function under the SP metric on mesh graphs. This requires finding a group of nodes that locally maximize SP distance with the user-defined node. Since the mesh of a delicate-described character may have tens or hundreds of vertices, estimating and finding such nodes with the longest SP distance accurately and fast are also well-motivated.
> > > > > > > >
> > > > > > > > **Case 3: estimate latencies in communication networks.**
> > > > > > > >
> > > > > > > > In large-scale communication networks, the latencies between Internet hosts are defined as a round-trip measurement from one to another (i.e., SP distance), which is utilized for performance optimization in many network applications such as content distribution networks [9], multicast systems [10], distributed ﬁle systems [11], etc.
> > > > > > > >
> > > > > > > >
> > > > > > > >
> > > > > > > > [1] Rizi, F. S., Schloetterer, J., & Granitzer, M. (2018, August). Shortest path distance approximation using deep learning techniques. In *2018 IEEE/ACM International Conference on Advances in Social Networks Analysis and Mining (ASONAM)*(pp. 1007-1014). IEEE.
> > > > > > > >
> > > > > > > > [2] Cohen, E., Halperin, E., Kaplan, H., & Zwick, U. (2003). Reachability and distance queries via 2-hop labels. *SIAM Journal on Computing*, *32*(5), 1338-1355.
> > > > > > > >
> > > > > > > > [3] Perozzi, B., Al-Rfou, R., & Skiena, S. (2014, August). Deepwalk: Online learning of social representations. In *Proceedings of the 20th ACM SIGKDD international conference on Knowledge discovery and data mining* (pp. 701-710).
> > > > > > > >
> > > > > > > > [4] Grover, A., & Leskovec, J. (2016, August). node2vec: Scalable feature learning for networks. In *Proceedings of the 22nd ACM SIGKDD international conference on Knowledge discovery and data mining* (pp. 855-864).
> > > > > > > >
> > > > > > > > [5] Jiang, L., Lai, Y., Chen, Q., Zeng, W., Yang, F., & Yi, F. (2021, September). Shortest Path Distance Prediction Based on CatBoost. In *International Conference on Web Information Systems and Applications* (pp. 133-143). Springer, Cham.
> > > > > > > >
> > > > > > > > [6] Chen, M., Wang, N., Lin, G., & Shang, J. S. (2021). Network-Based Trajectory Search over Time Intervals. *Big Data Research*, *25*, 100221.
> > > > > > > >
> > > > > > > > [7] Aujay, G., Hétroy, F., Lazarus, F., & Depraz, C. (2007, August). Harmonic skeleton for realistic character animation. In *SCA 2007-ACM-SIGGRAPH/Eurographics Symposium on Computer Animation* (pp. 151-160). Eurographics Association.
> > > > > > > >
> > > > > > > > [8] Poirier, M., & Paquette, E. (2009, May). Rig retargeting for 3D animation. In *Graphics interface* (pp. 103-110).
> > > > > > > >
> > > > > > > > [9] Ratnasamy, S., Handley, M., Karp, R., & Shenker, S. (2002, June). Topologically-aware overlay construction and server selection. In *Proceedings. Twenty-First Annual Joint Conference of the IEEE Computer and Communications Societies* (Vol. 3, pp. 1190-1199). IEEE.
> > > > > > > >
> > > > > > > > [10] Nogueira, J. (2014). A large-scale and decentralised applicationlevel multicast infrastructure.
> > > > > > > >
> > > > > > > > [11] Rhea, S. C., Eaton, P. R., Geels, D., Weatherspoon, H., Zhao, B. Y., & Kubiatowicz, J. (2003, March). Pond: The OceanStore Prototype. In *FAST* (Vol. 3, pp. 1-14).

---

> > > > > > > > > ### Comment · Reviewer_VC9e · 2022-12-05
> > > > > > > > > **I thank the authors for carefully addressing my concerns**
> > > > > > > > >
> > > > > > > > > I appreciate the authors’ efforts to systematically assessing and clarifying hyper-parameter impact, which I found unclear and missing in the original paper.
> > > > > > > > >
> > > > > > > > > I also found it helpful to clarify the importance of SPRL as this is crucial for the motivation and impact of the proposed approach. This has helped clarify the impact of the proposed research.
> > > > > > > > >
> > > > > > > > > All in all, I find that the rebuttal has helped clarify the manuscript and my concerns and raise my score finding the manuscript marginally above the acceptance threshold (6).

---

> > > > > > > > > > ### Author Response · Authors · 2022-12-05
> > > > > > > > > > **Thank you!**
> > > > > > > > > >
> > > > > > > > > > We would like to sincerely thank Reviewer VC9e for carefully reading our manuscript and response, providing constructive comments to improve this manuscript as well as updating the recommendation score.

---

### Official Review · Reviewer_WYyv · 2022-10-27

**Confidence:** 3
**Correctness:** 3
**Technical Novelty And Significance:** 4
**Empirical Novelty And Significance:** 3
**Recommendation:** 6

**Clarity, Quality, Novelty And Reproducibility:**

The idea of using betweenness centrality to improve SP random walks is intuitively appealing, and appears to be both novel and useful.

Although the paper is reasonably well-organized, there are significant presentational issues: frequent errors of grammar and idiom, very terse explanations in parts, undefined notation (particularly in Eq. 4), overloaded notation (sigma), and many hidden details.  Despite the complexity of the material, the paper could be much more clear and self-contained than it currently is.

Even with the presentational issues in the main body of the paper, the supplementary material does provide much useful information, including full proofs of the theoretical claims (as well as new claims not appearing in the main body), and important details regarding the data sets and implementation. Much of this should be at least referred to within the main paper.


**Strength And Weaknesses:**

Strengths:

1) The proposal of BC-based walks for this problem is well motivated through a discussion of the drawbacks of uniform random walks, with excellent illustrations on example graphs.  The authors argue that although truncated random walks are relied upon as a heuristic to generate a set of representative SPs from which paths from a test pair of nodes can be estimated, such truncated random walks have only a limited exploration range due to their tendency to remain in the vicinity of the start node.  The authors address this by augmenting random walk by reweighting the probability of selection of a node according to its betweenness centrality - the number of shortest paths that contain the node as a proportion of all shortest paths in the graph (excluding paths that start or end at the node in question).

2) The authors provide a theoretical justification for their distance resampling method, showing that as the number of SP observations increases, distances in the learned embedding tend to a linear relation with the original graph distance.  The distance resampling method adapts an existing strategy for learning node-level embeddings of arbitrary paths so as to preserve point-wise mutual information. In order to encourage embeddings with more faithful distance preservation and better coverage, the authors use shortest paths instead of arbitrary paths, generated using BC-augmented walks.  Although the learning framework is essentially the same, the authors use a negative sampling regularization upon which their theoretical guarantees depend.

3) An extensive empirical evaluation against 8 competing algorithms, including most if not all of the recent state-of-the-art methods, for 5 large real-world graphs. The results are impressive, with the authors' 3 proposed variants consistently and substantially outperforming these benchmarks in terms of accuracy (mAE and mRE). Of the benchmark methods, only CatBoost was able to remain competitive.

Weaknesses (and questions):

1) Very significant presentational issues exist in the paper (see below).

2) Close variants of SP representation learning are not considered. Can the method be applied to unweighted graphs?

3) The BC-based node selection reweighting proposed in Eq. 3 seems not to take into account the proportion of shortest paths through a neighbor of node v that also go through v itself (or equivalently, avoid v). Could such information be leveraged in improving the characteristics of the SP search, and the paths produced thereby?


**Summary Of The Paper:**

This paper proposes an approach for overcoming some of the deficiencies of random-walk graph exploration in learning a low-dimensional shortest path (SP) representation of graphs.  After identifying the drawbacks of generating SP representations via random walks, they propose an alternative representation based on betweenness centrality (BC), and argue that BC allows a wider range of exploration as well as better prospects for identifying high-quality shortest paths between test points at runtime.  In order to improve the fidelity of distance relations within the learned SP representation graph, they adapt a distance resampling learning strategy originally used with random walks, and then give theoretical guarantees on the relationship between learned pairwise distances and the original graph distances.  They then provide an empirical evaluation showing substantial improvement in performance over a large range of competing methods.


**Summary Of The Review:**

The idea of using betweenness centrality to improve SP random walks seems to be new, and the empirical performance is impressive and bear out the theoretical guarantees provided. However, the paper is dragged down by presentational issues and a lack of clarity and completeness in exposition. With sufficient attention to exposition, greater precision in the notation, and the inclusion of proof sketches in the main body, this could be an excellent paper.

---

> ### Author Response · Authors · 2022-11-13
> **Author Response to Reviewer WYyv (1/2)**
>
> We sincerely thank Reviewer WYyv for in-depth suggestions of the proposed method and constructive comments on the manuscript. We would like to carefully address the concerns as follows.
>
> ***
>
> > **Comment 1** Very signiﬁcant presentational issues exist in the paper.
> >
> > frequent errors of grammar and idiom, very terse explanations in parts, undeﬁned notation (particularly in Eq. 4), overloaded notation (sigma), and many hidden details.
> >
> > Much of supplementary material should be at least referred to within the main paper.
>
> **Response 1** We agree that the main body contains presentational issues and revise it from several aspects as follows. Also, we will further improve the writing quality and organization after several main concerns of this paper have been addressed.
>
> For terse explanations and hidden details,  we find several concepts and explanations which should be further discussed in detail (also raised by other reviewers). We revise some of them in the updated manuscript (both in the main paper and supplementary material), including:
>
> - better motivation for estimating accurate SP distance by investigating several important real-world downstream applications.
> - explain carefully why we need a BCDR procedure instead of directly sampling SPs to optimize $Z$ both conceptually and empirically.
> - a detailed investigation of parameter settings of Alg. 1 and discuss 9 critical parameters for their impacts both conceptually and empirically.
>
> For undeﬁned and overloaded notations, we correct the existing mistakes and unify the use of notations mentioned by the Reviewer.
>
> For the supplementary materials, We have linked each part of the materials with several corresponding parts of the main body.
>
> For frequent errors of grammar and idiom, we have tried our best to check and fix each existing grammar error we find and would appreciate it if detailed information towards this issue could be further provided.
>
> ***
>
> > **Comment 2** close variants of SP representation learning are not considered. Can the method be applied to unweighted graphs?
>
> **Response 2** The proposed method BCDR has been applied to 5 real-world unweighted graphs (see details in Tab. 4) and possesses superior performance both on accuracy and computational efficiency. We also evaluate BCDR on 6 unweighted graphs simulated with divergent topology, which shows BCDR could perform well in any other unweighted graphs. Besides, it would be interesting to see the application in weighted graphs, which requires several trivial but necessary modifications (listed as follows).
>
> - replace each BFS operation with the Dijkstra algorithm for estimating BC on weight graphs (line 6 in Alg. 1).
> - distance retrieved from BC walk should be updated by the edge weight instead of $1$, i.e., in Alg. 1, replace line 24 with
> $D_i[v_j]=\min ${$ D_i[v_j], D_i[v_c] + D_{cj}$}, and line 26 with $D_i[v_j]=D_i[v_c]+D_{cj}$. Note that $D_{cj}$ could be directly retrieved from $E$, since $v_j \in \mathcal N_c$ (line 21).
> - remove the quantizing process discussed in Sec. 4.1, since the outputs of models belong to $\mathbb R_+$ instead of $\mathbb N_+$.

---

> > ### Author Response · Authors · 2022-11-13
> > **Author Response to Reviewer WYyv (2/2) (modified)**
> >
> > > **Comment 3** The BC-based node selection reweighting proposed in Eq. 3 seems not to take into account the proportion of shortest paths through a neighbor of node v that also go through v itself (or equivalently, avoid v). Could such information be leveraged in improving the characteristics of the SP search, and the paths produced thereby?
> >
> > **Response 3** We appreciate this interesting and in-depth question for improving the performance of BC-based walk. We would like to explain the advantages and disadvantages of this modification from two aspects.
> >
> > **First**, the modified probability distribution of node transition sounds more desirable for wider exploration than BC-based walk, but suffers an extra expense of storing transition-dependent weights. In contrast with BC-based walk defined by Eq. 3, this transition could be formulated as
> >
> > $\qquad \tilde P(\mathcal W^j_a = v_m | \mathcal W^{j-1}_a = v_n) = \dfrac{ {\rm BC}(v_m|v_a)} { \sum\_{v_k \in \mathcal N_n} {\rm BC}(v_k|v_a) } ,\ v_m \in \mathcal N_n$
> >
> > where $v_a$ is the root of a BC-based walk and $\rm BC(v_m|v_a)$ is defined as a part of $\rm BC(v_m)$ that calculates SPs passing through both $v_m$ and  $v_a$, i.e.,
> >
> > $\qquad {\rm BC} (v_m|v_a) := \sum_{s \neq m \neq t \wedge s \neq a \neq t} \dfrac{\sigma_{st}(v_a,v_m)}{\sigma_{st}(v_a)}$
> >
> > Here, $\sigma_{st} (v_a,v_m)$ means the number of SPs passing through $v_a$ and $v_m$. This transition avoids some nodes $v_m$ that indeed possess a high probability to be a part of most SPs but might not be that of SPs passing through $v_a$. Although it sounds much more plausible, it introduces at least $O(N^2)$ space to store such *conditional BC* (or even worse time complexity for directly computing on each transition). Instead, the proposed BC-based walk possesses linear complexity both in time and space, which is more practical for exploring high-order SP structures.
> >
> > **Second**, although such inconsistent bias exists in BC-based walks, it has been corrected by the subsequent DR process. As stated in Eq. 5 and further concluded in Prop. 1, the resampling distribution $\tilde Q_{W_i}(v_j)$ reflects the accurate SP-based locality of $v_i$ by introducing $\alpha^{D_{ij}}$ (which indicates the distance between any $v_j$ to the root $v_i$ instead of arbitrary nodes defined in BC), and the inconsistent bias on BC is eliminated by constructing negative sampling distribution also from the multiset of nodes on BC-based walk paths (i.e., $W_i$). A more mathematical explanation of this fact can be found in the proof of Prop. 1.
> >
> > In conclusion, avoiding such inconsistent bias directly when performing explorations of SP distance on graphs is still an attractive direction worthy of further study, despite its currently prohibitive complexity in time and space. We would like to study this problem in the future.

---

> > > ### Comment · Reviewer_WYyv · 2022-12-07
> > > **Response to authors**
> > >
> > > I have read the authors' responses. First, let me apologize for the typo in W2 - "unweighted" should be "weighted"! Since only the unweighted case was actually considered, it would have been interesting to see the performance of the proposed approach on edge-weighted graphs.
> > >
> > > For W3, I was mainly concerned with immediate neighbors (adjacent nodes) - the authors' argument that prohibitive storage would be needed is true for large neighborhoods. However, for 1-neighborhoods, the transition-dependent information could be stored with the graph edges so as to avoiding the quadratic storage costs mentioned by the authors in their reply.
> > >
> > > I am still concerned with the presentational issues, and do hope that the authors put more effort into this, regardless of whether the paper is ultimately accepted.

---

> > > > ### Author Response · Authors · 2022-12-08
> > > > **Author Response to Reviewer WYyv**
> > > >
> > > > We thank Reviewer WYyv for kind feedback and further clarification on the concerns. We do understand the reviewer‘s concern about presentation issues and will undoubtedly make every effort to meticulously revise this manuscript for better exposition and further improve the readability in later versions.
> > > > ***
> > > > Besides, we appreciate the added clarification on *W3* and would like to share further thoughts regarding this point. We must first apologize for a confusing mistake (which has been corrected now) made in the previous **response 3** that '$v_n$' should be 'the root node $v_a$' since we aim to find a node possessing a longer SP distance **relative to the root** instead of the previous sampled node (i.e., $v_n$). The computation of such *conditional BC* indeed requires $\Theta (N^2)$ space and even worse time complexity. However, if only immediate neighborhoods (i.e., $\mathcal N_m$) are considered, i.e.,
> > > >
> > > > $\qquad \tilde P(\mathcal W^j_a = v_m | \mathcal W^{j-1}_a = v_n) = \dfrac{ {\rm BC}(v_m|v_n)} { \sum\_{v_k \in \mathcal N_n} {\rm BC}(v_k|v_n) } ,\ v_m \in \mathcal N_n$
> > > >
> > > > , as the reviewer stated, the transition-dependent information could be definitely reduced to $O(M)$ (where $M$ is the number of edges).
> > > >
> > > > However, we empirically find that such local information gives rise to little improvement in (even harmful to) the exploration range of SP distance. In contrast with *BC-based walk*, we further construct a similar variant where each transition tends to select nodes with larger $BC(v_m|v_n)$ instead of $BC(v_m)$, called the *first-neighborhood-conditioned BC-based walk* (*1st-CBC walk*).
> > > >
> > > > Also, we evaluate *1st-CBC walk* for its exploration range of distance by each transition step on the six simulated graphs defined in Appendix A.8.6. The experimental setting is the same as Sec. 4.2,  but we present the results in the following tables (since uploading figures is not supported on the OpenReview website).
> > > >
> > > > | SP Distance on CG | [0,5]      | (5,10]     | (10,15]    | (15,20]    | (20,25]   | (25,30]   |
> > > > | ----------------- | ---------- | ---------- | ---------- | ---------- | --------- | --------- |
> > > > | BC-based walk     | 32.32%     | 26.25%     | **19.46%** | **13.56%** | **6.90%** | **1.52%** |
> > > > | 1st-CBC walk      | **43.59%** | **30.86%** | 16.26%     | 7.26%      | 1.73%     | 0.29%     |
> > > >
> > > > | SP Distance on TG | [0,14]     | (14,27]    | (27,40]    | (40,54]   | (54,67]   | (67,80]   |
> > > > | ----------------- | ---------- | ---------- | ---------- | --------- | --------- | --------- |
> > > > | BC-based walk     | 57.94%     | **23.42%** | **11.62%** | **5.43%** | **1.36%** | **0.23%** |
> > > > | 1st-CBC walk      | **98.35%** | 1.65%      | 0.00%      | 0.00%     | 0.00%     | 0.00%     |
> > > >
> > > > | SP Distance on TCG | [0,4]      | (4,7]      | (7,10]     | (10,14]   | (14,17]   | (17,20]   |
> > > > | ------------------ | ---------- | ---------- | ---------- | --------- | --------- | --------- |
> > > > | BC-based walk      | 57.41%     | **21.73%** | **14.77%** | **6.02%** | **0.08%** | **0.00%** |
> > > > | 1st-CBC walk       | **84.52%** | 11.62%     | 3.42%      | 0.44%     | 0.00%     | **0.00%** |
> > > >
> > > >
> > > > | SP Distance on TRG | [0,2]      | (2,3]      | (3,4]      | (4,6]      | (6,7]     | (7,8] |
> > > > | ------------------ | ---------- | ---------- | ---------- | ---------- | --------- | ----- |
> > > > | BC-based walk      | 48.05%     | **13.72%** | **11.77%** | **19.56%** | **6.90%** | 0.00% |
> > > > | 1st-CBC walk       | **72.47%** | 12.99%     | 8.18%      | 5.71%      | 0.65%     | 0.00% |
> > > >
> > > > | SP Distance on NG | [0,4]      | (4,7]      | (7,10]     | (10,14]   | (14,17]   | (17,20]   |
> > > > | ----------------- | ---------- | ---------- | ---------- | --------- | --------- | --------- |
> > > > | BC-based walk     | 56.90%     | **29.42%** | 10.67%     | 2.77%     | **0.24%** | **0.00%** |
> > > > | 1st-CBC walk      | **57.58%** | 26.97%     | **12.14%** | **3.18%** | 0.13%     | **0.00%** |
> > > >
> > > > | SP Distance on SG | [0,10]     | (10,20]    | (20,30]    | (30,40]   | (40,50]   | (50,60]   |
> > > > | ----------------- | ---------- | ---------- | ---------- | --------- | --------- | --------- |
> > > > | BC-based walk     | 30.94%     | 31.76%     | **22.54%** | **9.53%** | **4.04%** | **1.19%** |
> > > > | 1st-CBC walk      | **36.25%** | **32.47%** | 20.59%     | 7.17%     | 2.78%     | 0.75%     |
> > > >
> > > >
> > > > where each percentage means the proportion of nodes possessing different SP distances that occurred in walk paths.
> > > >
> > > > The results show that a heuristic division of BC on local neighborhoods impacts the ability to find high-order neighbors of the root. Clearly, $BC(v_m)$ involves several SPs that do not pass through $v_n$, but we can not assume these SPs also do not contain $v_a$ nor simply prune them without a proper global prior. Under such an insufficient assumption of locality, 1st-CBC walk appears to degenerate like naive random walks.

---

### Official Review · Reviewer_MJqQ · 2022-11-29

**Confidence:** 3
**Correctness:** 4
**Technical Novelty And Significance:** 4
**Empirical Novelty And Significance:** 4
**Recommendation:** 8

**Clarity, Quality, Novelty And Reproducibility:**

The paper is well-written and the ideas are presented clearly. The organization is good. Based on the related work presented in the paper and on my limited literature search, I believe that the idea of using between centrality and distance resampling for the task of shortest path based representation learning is novel.

**Strength And Weaknesses:**

Strengths:
- The paper addresses an intellectually challenging and important problem
- The proposed method is novel and is supported by theoretical arguments and empirical evaluation

Weaknesses:
- It is not clear if the implementation will be made public which could impact the reproducibility of the results


**Summary Of The Paper:**

The paper presents an algorithm to learn shortest-path (SP) representation of nodes in a graph. Existing strategies to learn SP are based on random-walks that, based on the structure of the graph, have limitations in terms of performance as well as distance preservation. The authors propose a method called Between Centrality-based Distance Resampling or BCDR that accomplishes two things. First, by using a centrality-based random walk approach, it ensures a better exploration of the possible paths and thus yield better distance preservation. Second, it uses a distance resampling strategy to improve the performance of the learning task. Authors provide theoretical proofs to support the algorithmic choices in the BCDR algorithm. A comparison with several existing methods on several benchmark data sets is provided to show that BCDR allows for improvements, both in terms of distance preservation accuracy and performance (speed).

**Summary Of The Review:**

Overall, the paper seems to make significant contributions and presents and easy to understand paper that will be of interest to the conference audience and the community.

---

> ### Author Response · Authors · 2022-11-30
> **Author Response to Reviewer MJqQ**
>
> We sincerely thank Reviewer MJqQ for the constructive and encouraging review of our work.
>
> We are willing to **make the full implementation codes of BCDR public** and **provide an efficient open-source tool** for SP distance querying after the acceptance. Furthermore, we believe the results of BCDR algorithm are easy to re-produce, since our implementation is totally based on four existing and widely used machine learning or graph computing toolkits, i.e., NetworkX [1], GenSim [2], PyTorch [3] and CatBoost [4]. Thereinto,
>
> - NetworkX==2.8.5: for performing BFS on undirected graphs to find distance triplets and estimate BC;
> - GenSim==4.2.0: for training the BC walk paths to acquire node-level embeddings;
> - PyTorch==1.12.1: for constructing a two-layer fully connected network to predict SP distance.
> - CatBoost==1.0.6: for directly boosting the accuracy performance by combining local and global features.
>
> Based on these toolkits, one could implement BCDR according to a detailed pseudo-code description in Appendix A.2 (a link to our GitHub Repository will be also available in the camera-ready paper).
>
> Besides, we also would like to further clarify any reproducible issues regarding our experimental results, please do not hesitate to post additional comments and questions on them, thanks!
>
> [1] https://networkx.org/
>
> [2] https://radimrehurek.com/gensim/
>
> [3] https://pytorch.org/
>
> [4] https://catboost.ai/

---

### Author Response · Authors · 2022-11-19
**General Response at the end of Discussion Phase 1**

We sincerely thank all reviewers for their efforts and constructive suggestions for improving this manuscript, despite no further additional questions regarding our responses received during Discussion Phase 1. We understand reviewers are busy reviewing several long papers in a limited time and are willing to briefly summarize here major concerns and corresponding revisions in the updated manuscript. We hope the following summary will help reviewers / AC save time and be able to roughly grasp our revisions in the coming Discussion Phase 2.
***

**1. Motivation issues.**

> The significance of estimating SP distance accurately

We clarify that fast and accurate estimation of SP distance is not an indirect theoretical or computational problem, and it does directly play significant roles in many real-world downstream applications. We investigate several real-world scenarios in Appendix A.1.1 for its significance.

> Why utilize BCDR procedure instead of directly optimizing $Z$ on sampled SPs?

We clarify the motivation of each step in BCDR and compare BCDR with directly optimizing $Z$ on SPs both technically and empirically in Appendix A.7.  Further experimental results comparing BCDR with 6 intuitive strategies to directly optimize on SPs are provided.

***

**2. Methodological issues.**

> What is a 'random SP walk'?

We carefully explain in Sec. 3.2 this abstract concept and its relationship with BC-based walks. We hope the enriched explanation of DR process will improve the readability.

***

**3. Experimental issues.**

>  BCDR relies on the setting of multiple hyperparameters but careful investigation is missing.

We further carefully investigate the impacts of 9 critical parameters on multiple metrics both conceptually and empirically in Appendix A.11.2, and a summary of parameter setting to reproduce the results in Tab. 2 is also provided in Appendix A.9.

***

**4. General Presentational issues.**

In the latest manuscript, we correct all mistakes we find, including overloaded / undefined notations and grammar errors. We add relatively brief proof sketches of Prop. 1 and Thm. 2 in the main paper. We make sure each part of the appendix (including the newly added parts) is well linked with several corresponding parts of the main body.

***

We look forward to further comments regarding our responses and will be more than happy to address them. Thanks.

---

### Decision · Program_Chairs · 2023-01-20

**Decision:**

Reject

**Justification For Why Not Higher Score:**

The problem studied in the paper is not well-motivated and the quality of writing should be improved before acceptance.

**Justification For Why Not Lower Score:**

N/A

**Metareview: Summary, Strengths And Weaknesses:**

The authors introduce a new technique to compute shortest path embeddings of a graph. Classic approaches to compute shortest path embedding are based on truncated random walks. The authors first describe the drawbacks of such approaches and then present a new algorithm that uses random walk biassed by node centrality to obtain higher accuracy estimators.

The paper presents some interesting ideas and the experimental results on recovering the shortest path distance are nice.

Although the paper has some fundamental shortcoming that should be addressed before publication. First, the writing of the paper should be improved. The revised version is in a better state compared with the submitted one but the paper would still benefit by an in-depth rewriting. Second, the authors should motivate the problem of distance embedding more clearly. The reviewers were thankful for adding A.1.1, although all the reported examples are on planar graphs for which better distance oracles exist.

Overall, the paper presents some nice interesting ideas but it is not yet ready for publication at this stage.